# A SUPERPERSUASIVE AUTONOMOUS POLICY DEBATING SYSTEM

## ABSTRACT

The capacity for complex, evidence-grounded, and strategically adaptive persuasion remains a formidable grand challenge for artificial intelligence. Prior work, like IBM Project Debater, focused on generating isolated persuasive speeches in highly simplified and shortened debate formats for lay audiences. We introduce a novel autonomous system capable of participating in and winning a full, unmodified two-team competitive policy debate. Our system employs a hierarchical architecture of specialized multi-agent workflows, where teams of LLM-powered agents collaborate and critique one another to perform discrete argumentative tasks. Each workflow utilizes iterative retrieval, synthesis, and self-correction using an existing massive corpus of policy debate evidence: OpenDebateEvidence (Roush et al., 2024), mirroring the creative and strategic processes of elite human debate teams at real debate tournaments. We demonstrate through a **continuously running *live* spectacle debate performance** that our agents can autonomously construct logically sound, evidence-backed cases and engage in multi-turn debates, generate full speech transcripts and intelligently cross-examine each other. In preliminary evaluations against human-authored cases, our system produces qualitatively superior argumentative components and consistently wins simulated rounds as adjudicated by an independent autonomous judge. We also find that expert human debate coaches consistently prefer the arguments, evidence, and cases constructed by our system vs human debaters. Our system is easy to run, and has few dependencies. We release all of our code to the public.

## 1 INTRODUCTION

Generation of persuasive and strategically coherent argumentation is a long-standing goal in AI (Bench-Capon and Dunne, 2007; Dung, 1995). Existing research has largely studied competitive persuasive argumentation through evidence light, highly simplified variants of debate aimed at a lay audience (Slonim et al., 2021). We contend this approach avoids the strategic, game-theoretic, and iterative nature of real-world, competitive debate.

To address this, we turn to the uniquely suitable domain of American-style after-school competitive policy debate [1]. This format serves as an idealized crucible for AI research for many reasons: it is a popular extracurricular activity (McGrath, 2020), lengthy but strictly time-constrained, grounded in a vast body of high quality evidence, and possesses a complex, regimented, and formal structure that demands both long-term strategic planning and second by second tactical decision making. (NSDA, 2025). Human competitors have evolved sophisticated techniques [2] and tools to manage the immense cognitive load which makes it a premier environment for benchmarking and developing advanced autonomous reasoning agents.

We present an autonomous system that can create a complete affirmative case from a topic, research and construct a multi-pronged negative strategy, and then execute a full, eight-speech debate, including simulated cross-examinations. Our core contribution is a novel multi-agent architecture where complex creative and strategic tasks are decomposed into a pipeline of specialized workflows. Within each workflow, a team of Large Language Model agents collaborate and critique each other's outputs.

---

[1] A good introduction to the activity can be found here
[2] Including "Spreading" a portmanteau for "Speed Reading"

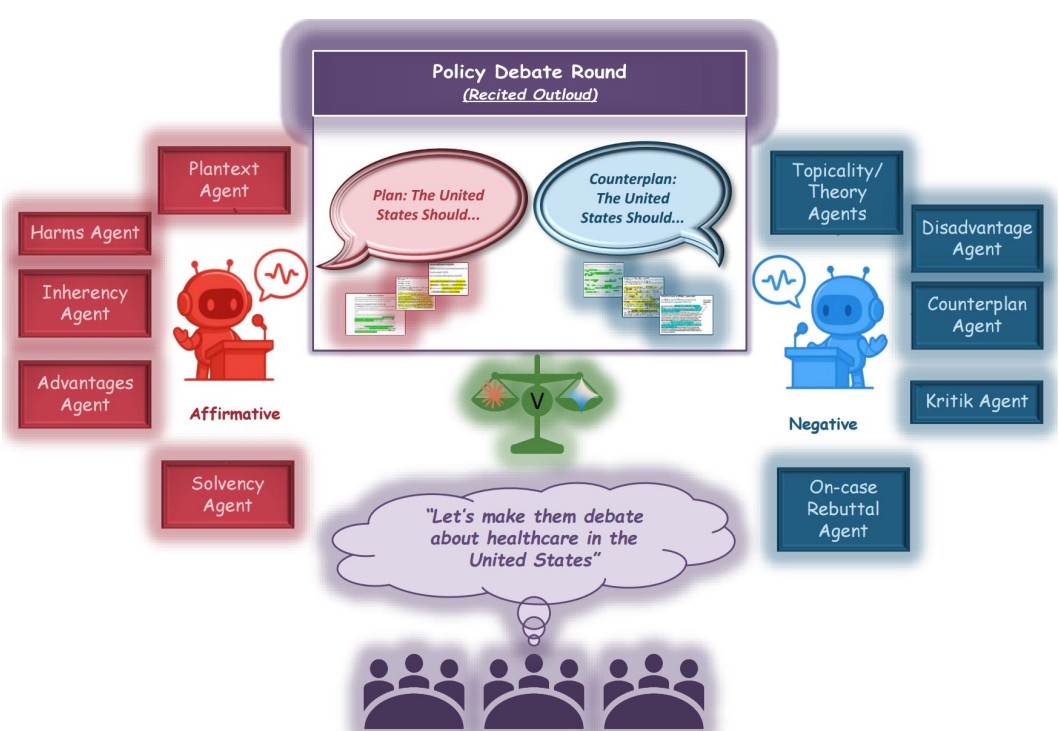

Figure 1: **Proposed Creative Demonstration:** After the audience picks a resolution, two onscreen systems—Team Affirmative (red) and Team Negative (blue)—launch specialist agents (Affirmative: Plan-text, Harms, Inherency, Advantages, Solvency; Negative: Topicality/Theory, Disadvantage, Counterplan, Kritik, On-case Rebuttal). Each team uses gpt-4-mini + OpenDebateEvidence indexed in DuckDB, with a live UI streaming the AG2 agent chats, searches, and evidence as arguments are drafted. Completed speeches are voiced through ElevenLabs text-to-speech[3] while the other side prepares its reply, cycling through the full debate round. Independent Judge agents (green) powered by Claude or Gemini will judge the round at the end of the speeches. Brave audience volunteers may step in for any speech or to propose a new topic (triggering a new debate)

Our contribution are threefold:

1. We introduce a hierarchical multi-agent framework for end-to-end generation of complex, evidence-grounded argumentation, modeling the **entire lifecycle** of a competitive policy debate.

2. We show that by decomposing the creative process into discrete, role-based agent work-flows, our system can **master** the intricate structure and esoteric strategies of an expert argumentative domain.

3. Through empirical comparisons and human expert evaluations, we show that our system produces argumentative artifacts of superior quality, faithfulness, and strategic coherence compared to strong human baselines, and can consistently win simulated debates.

A complete description of our **Creative System Demonstration** is given in **Figure 1**

## 2 BACKGROUND: THE CRUCIBLE OF POLICY DEBATE

American competitive policy debate is a popular afterschool extracurricular team-based activity where two teams, the Affirmative and the Negative, argue over a resolution. The Affirmative presents a specific plan to enact the resolution and argues that it will result in desirable outcomes (Advantages). The Negative's objective is to refute the Affirmative case and argue that the plan (and sometimes the whole resolution) is a bad idea.

The structure of policy debate is rigidly formalized, comprising constructive speeches, cross-examinations, and rebuttal speeches[4]. The foundation of modern policy debate is evidence, colloqui-ally known as "cards." A card consists of a direct quotation from a published source (e.g., academic journals, government reports, news articles), a full citation, and a "tag" (a short, synthesized claim the evidence is meant to support). Cases are built by chaining these cards together alongside natural language arguments to establish a logical sequence from problem to solution, or from "solution" to other problem.

## 3 RELATED WORK

The pursuit of computational argumentation has a rich history. Early work focused on argument min-ing and logical formalism (Bench-Capon, 2002; Rinott et al., 2015; Murakami and Raymond, 2010). More recently, Large Language Models have demonstrated a remarkable capacity for generating fluent and coherent argumentative text (Al Khatib et al., 2021).

The most prominent predecessor to our work is IBM's Project Debater (Slonim et al., 2021; Bar-Haim et al., 2021b). Project Debater was a landmark achievement, showcasing an AI system that could engage in a limited and highly simplified, live, public debate with a human. It excelled at mining a massive corpus for relevant claims, clustering them into themes, and delivering a single, persuasive speech for a lay audience. While Project Debater focused on the rhetorical task of persuading an audience in a one-off speech, our system engages with the strategic, evidence-based, and multi-turn *game* of competitive policy debate. The distinctions are critical:

- **Domain and Audience:** The debate format used by Project Debater was non standard (i.e. there are no tournaments in that format, compared to hundreds per year in Policy Debate), extremely short, and was designed for ease of development and creation of the spectacle. Project Debater targets a lay audience with appeals to pathos and ethos. Our system operates in a real world, expert domain where persuasion is governed by a complex set of rules, norms, and a reliance on "cards" evaluated by a specialist judge.

- **Task Complexity:** Project Debater produces a single, monolithic speech with very limited reference to evidence. Our system generates an entire ecosystem of structured arguments and evidence for an eight-speech, two-team debate, including multi-pronged negative strategies, cross-examinations, and full rebuttal speeches that must respond to every argument made in the preceding speech (Orbach et al., 2020; Lavee et al., 2019). Our system is fully interactive and can allow humans to stand in for one of the teams.

---

[4]A summary of the rules and speaker order can be found here

- **Argumentation Style:** Project Debater synthesizes arguments from a broad corpus of non fact-checked evidence to make general claims. Our system uses a vast, human created and curated, high-quality evidence dataset to construct intricate, interlocking chains of specific factual and grounded claims. (Bilu et al., 2019; Singh et al., 2018).

Our work also builds upon recent advances in multi-agent systems (Wu et al., 2023; Park et al., 2023; Bar-Haim et al., 2021a). Frameworks like AutoGen/AG2[5], smolagents (Roucher et al., 2025), crewai[6], and similar have shown that ensembles of LLM agents with specified roles can solve complex tasks more effectively than a single model. We extend this paradigm by creating a hierarchical structure of these multi-agent teams, pipelining their outputs to tackle the multi-stage creative process of building and executing a debate case. This structured collaboration is the key to managing the immense complexity of the task.

In parallel, a rapidly growing body literature examines AI persuasion, and specifically its emerging capabilities, risks, and governance. Empirically, controlled trials show GPT-4 class models already outperform humans in extremely simplified live, structured debates when given minimal personal information (Salvi et al., 2025), a result amplified by media coverage (Dolan, 2025; EPFL News, 2023). Surveys and normative analyses frame these trends as "superpersuasion" or "hyperpersuasion and map mitigations (Rogiers et al., 2024; Floridi, 2024a;b; Barnes, 2021; Yale Institution for Social and Policy Studies, 2024). Some argue present systems aren't yet "dangerously" persuasive while others warn of rapid scaling and targeting (Ars Technica, 2025; Castelvecchi, 2025). Commentary puts a political lens on potential weaponization (Molloy, 2025) and scenario work explores broader societal dynamics (Kokotajlo et al., 2025; Wade, 2022). Popular media and industry voices (i.e Sam Altman) have foregrounded the notion of "superhuman persuasion," shaping the discourse (Nosta, 2023; Harnessing Hybrid Intelligence Blog, 2025; Altman, 2023; Dupré, 2023; medtigo News, 2023).

## 4 SYSTEM ARCHITECTURE

Our autonomous debating system is a modular, pipelined framework built upon a series of specialized multi-agent workflows. At its core are two main components: the evidence base and the multi-agent conversational architecture. A **complete system diagram** is given in **appendix B 3**

### 4.1 EVIDENCE CORPUS AND RETRIEVAL

The system is grounded in the OpenDebateEvidence (Roush et al., 2024) dataset, a large-scale, human created corpus of over 3 million "cards" used in actual high school and college tournaments, indexed into a DuckDB (Mühleisen and Raasveldt, 2025) database. We utilize BM25 keyword search via the ducksearch (Sourty, 2024) library for efficient retrieval. This evidence grounding is paramount because every substantive claim made by the system must be directly traceable to a specific piece of evidence in the database. Agents generate queries to find evidence and then reason over the retrieved documents before using structured generation to respond.

### 4.2 MULTI-AGENT WORKFLOW PATTERN

The fundamental building block of our system is a repeatable multi-agent workflow pattern used by most of the debate agents. We decompose the task into a collaborative and rigidly structured optimization loop between specialized agents powered by gpt-4.1-mini via the AG2/Autogen framework. **Figure 2** shows a typical pattern used at the heart of each workflow

### 4.3 THE DEBATE GENERATION PIPELINE

The system simulates a full debate by executing a pipeline of multi-agent workflows in sequence, with the output of each stage being appended to the iteratively drafted debate round document, and the document serving as the context for the next agent/speech.

---

[5]https://ag2.ai/
[6]https://www.crewai.com/

Figure 2: This loop of generation, retrieval, and critical review continues for a set number of iterations or until the Reviewer agent is satisfied. Structured outputs, enforced via Pydantic (Colvin et al., 2025) models and structured/constrained generation, guarantee that agent messages are machine-readable and conform to the expected format for each task.

Policy Debate Cases have a **rigid** format that **must be used** or else **winning is difficult**. Our workflows perfectly match this format. The iterative drafting pipeline is detailed below, in sequential order:

**1. 1AC (First Affirmative Constructive) Generation:** The system begins by constructing the Affirmative case. This is performed in three phases.

- *Plantext Generation:* The first workflow brainstorms and researches a viable plan of action that affirms the resolution.

- *Stock Issue Workflows:* A series of dedicated workflows then build out the case. Separate agent teams are tasked with finding the best evidence for the **Harms** (the problem in the status quo), **Inherency** (why the problem is not being solved now), and **Solvency** (how the plan solves the problem).

- *Advantage Generation:* Additional workflows create distinct **Advantages**, each with its own internal evidence chain for **Uniqueness** (status quo baseline), **Link** (how the plan causes change), **Internal Link** (steps from change to impact), and **Impact**.

*Cross-Examination (CX) after 1AC:* A Negative speaker questions the Affirmative on clarity and vulnerabilities in the plantext, solvency mechanisms, and advantages. The purpose is to extract concessions, expose contradictions, or set up later negative arguments.

**2. 1NC (First Negative Constructive) Generation:** Given the completed 1AC as context, the system generates a multi-pronged negative strategy.

- *Strategy Generation:* A high-level workflow first generates a portfolio of diverse negative positions.

- *Off-Case Workflows:* Specialized agent teams are then deployed to fully construct each position, complete with evidence. These positions will be one or more of the following:

  - **Topicality/Theory:** Argues the Affirmative plan is not an example of the resolution or violated some rules of how debate should operate

- **Disadvantages (DAs):** Argues the plan causes a new, terrible problem. The reverse of an **Advantage**. Each has its own **Uniqueness**, **Link**, **Internal Link**, and **Impact** workflows.
- **Counterplans (CPs):** Proposes an alternative, non-topical action that is superior to the plan. Includes its own **Counterplan Text**.
- **Kritiks (Ks):** Challenges the philosophical or ethical underpinnings of the Affirmative's advocacy. Creates a formalized **Alternative** (counter advocacy) to the plan.

- *On-Case Rebuttals:* The system also finds evidence to directly attack the cards presented in the 1AC.

*Cross-Examination (CX) after 1NC:* The Affirmative asks targeted questions of the Negative, clarifying the scope of Disadvantages, testing Counterplan competitiveness, probing the logical consistency of Kritiks, and setting up permutations and solvency defense.

**3. Rebuttals, Cross-Examination, and Round Mechanics:** After the 1NC, the pipeline continues through the remaining speeches in strict sequence: **2AC (Second Affirmative Constructive)**, **2NC (Second Negative Constructive)**, **1NR (First Negative Rebuttal)**, **1AR (First Affirmative Rebuttal)**, **2NR (Second Negative Rebuttal)**, **2AR (Second Affirmative Rebuttal)**, with each speech agent receiving the *entire accumulated transcript* (flows, cards, concessions, and CX records) as context. The system enforces common policy-debate norms (line-by-line structure, extensions with warrants, weighing, "no new evidence" constraints in later rebuttals). Cross-examinations (CX) are simulated as targeted two-agent Q&A interludes attached to the relevant speech.

- *2AC (Second Affirmative Constructive):*
  - **Primary role:** Answer *every* 1NC position on a line-by-line basis and rebuild affirmative offense. The 2AC responds to all negative off case positions: Topicality/Theory, Disadvantage links/impact defense, Counterplan theory/competitiveness, solvency deficits, permutations (*do both*, *severance/intrinsicness* as permitted), and Kritik framework/alternative answers.
  - **Case work:** Extend 1AC Advantages with explicit **warranted extensions** (tag + author + reasoning), renew internal links, and add **impact calculus** (magnitude, timeframe, probability, reversibility, scope).

  *Cross-Examination (CX) after 2AC:* Negative speakers probe the thoroughness of 2AC answers, test permutations for logical flaws, and set up strategic collapse into the block.

- *The Negative Block (2NC and 1NR):* Taken together, these two back-to-back negative speeches form the "block," giving the Negative extended time to deepen select positions and induce *time pressure* on the 1AR. The "block" exists as a check-back for the advantages that the Affirmative's first and last speeches give it.
  - **2NC:** *Strategic collapse and deepening.* Chooses a focused path to the ballot (e.g., **Counterplan + Disadvantage**, **Theory/Topicality**, or a **Kritik**) and drives depth: extends key links/uniqueness, develops solvency deficits, articulates competition (functional/textual), answers 2AC permutations, and pushes **impact calculus** tailored to the chosen path. By default, the system disallows brand-new off-case positions after the 1NC (configurable), but permits new warranted analytics, evidence, and applications that *extend* existing positions.
  - **1NR:** *Coverage and cleanup.* Completes block coverage on remaining flows (e.g., additional Disadvantage answers, case takeouts/turns, Kritik links/alt solvency, Topicality reasonability vs. competing interpretations). The 1NR may **kick** (concede) a position with an explicit theory of *no risk/no offense* to reallocate time.

  *Cross-Examination (CX) after the Block:* The Affirmative uses this period to pressure the Negative's strategic collapse, testing solvency deficits, pointing out contradictions, and locking in concessions for 1AR exploitation.

- *1AR (First Affirmative Rebuttal):*
  - **Primary role:** Compress the round under severe time asymmetry. The 1AR must cleanly *answer the entire block*, preserve decisive Affirmative offense, and prosecute the Negative's weakest links (e.g., permutation drops, Counterplan solvency holes, Disadvantage uniqueness turns).

- **Techniques:** Grouping closely related arguments, **cross-applying** answers across similar links, and prioritizing **weighing** ("even if" framing). No new off-case *positions* or *evidence*.

- *2NR (Second Negative Rebuttal):*
    - **Primary role:** Present a single, coherent **ballot story**. The 2NR *collapses* to one or two winning routes (e.g., **Topicality**, **Counterplan + Net Benefit**, **Big Disadvantage**, or **Kritik**) and executes decisive **impact calculus** versus any remaining Affirmative offense, pressing **presumption** where applicable (status quo or Counterplan vs. plan).
    - **Constraints:** No brand-new positions or evidence allowed. Only extensions and crystallization of the block.

- *2AR (Second Affirmative Rebuttal):*
    - **Primary role:** Answer the 2NR's narrowed case and win the **framing**. The 2AR prioritizes decisive offense that the 2NR failed to answer, executes high-leverage **impact framing** (e.g., existential risk vs. economic downturn, short-term irreversible harms vs. long-term diffuse harms), and resolves theory/framework burdens.
    - **Constraints:** No brand-new 2AR arguments/evidence. Only extensions/responses to the 2NR's crystallized claims.

    *Judge Agent and RFD (Reason for Decision):* The Judge Agent gives a decision, issues a detailed **Reason for Decision** with speaker points, decision calculus, and cites to specific evidence IDs.

A **sample complete debate round transcript** appears in **Appendix A 8**.

## 5 EXPERIMENTS AND EVALUATION

### 5.1 EXPERIMENT 1: HUMAN EVALUATION ON COMPONENT-LEVEL QUALITY

We performed a human evaluation of individual argumentative components. We randomly selected 3 advantages generated by our system and 3 human-authored advantages from OpenDebateEvidence on the same topic. These were anonymized and presented to 5 expert debate coaches (all with over 10 years of experience and multiple championship teams coached) for evaluation on a 1-5 scale across three metrics:

- **Quality:** The overall strategic coherence and persuasiveness of the argument.
- **Factuality:** The accuracy of the claims made in the tags and analysis.
- **Faithfulness:** How well the tag accurately summarizes the accompanying evidence ("card").

We reran this experiment three times to create error bars.

Table 1: Mean scores of system-generated vs. human-authored claims as rated by expert judges (1-5 scale, higher is better). Values are shown as mean $\pm$ standard deviation.

| Metric | Our System | Human-Authored |
|---|---|---|
| Quality | $4.32 \pm 0.31$ | $3.65 \pm 0.52$ |
| Factuality | $4.45 \pm 0.25$ | $3.98 \pm 0.23$ |
| Faithfulness | $4.81 \pm 0.19$ | $4.05 \pm 0.48$ |

### 5.2 EXPERIMENT 2: SIMULATED ROUND PERFORMANCE

To assess holistic strategic performance, we conducted 20 simulated debates. In 10 rounds, we pitted a human-authored 1AC against our system's generated 1NC and subsequent rebuttals. In the other 10, our system's 1AC was pitted against a human-authored 1NC. The full debate transcripts were then fed to our autonomous Judge Agent (using Gemini) to determine a winner.

Table 2: Simulated Debate Round Performance vs. Human-Authored Strategies

| Scenario | Rounds Played | System Wins | Win Rate |
|----------|:-------------:|:-----------:|:--------:|
| System as Negative (vs. Human 1AC) | 10 | 9 | 90% |
| System as Affirmative (vs. Human 1NC) | 10 | 8 | 80% |
| **Overall Performance** | **20** | **17** | **85%** |

Our system achieved a 90 percent win rate when playing Negative against human 1ACs, and an 80 percent win rate when playing Affirmative against human negative strategies. The Judge Agent's RFDs frequently highlighted the system's superior evidence quality and density and its comprehensive, line-by-line refutation in rebuttals.

## 6 CONCLUSION

We have presented a novel autonomous system that demonstrates a high level of proficiency in the complex, creative, and strategic domain of extracurricular competitive policy debate. By employing a hierarchical architecture of specialized, collaborative multi-agent workflows, our system manages the immense complexity of researching, constructing, and executing a full debate round. Our preliminary results indicate that this approach produces coherent and well-supported arguments and achieves a level of strategic quality that exceeds strong human baselines.

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

# 7 ACKNOWLEDGMENTS

This research was supported by National Science Foundation Small Business Innovation Research (NSF-SBIR) Grant No. 2431521. The views expressed are those of the authors and do not necessarily reflect the views of the funding agency.

# 8 LIMITATIONS

Our goal is to advance research on structured, evidence-grounded argumentation, not to produce a turnkey persuasion system. While our results are promising, they come with important caveats and risks.

Our evaluations focus on U.S.-style policy debate using a specialized evidence paradigm and norms (e.g., time pressures, line-by-line refutation, and spreading). This domain is intentionally adversarial and rule-bound. Performance may degrade outside English, outside the policy debate topic areas, or when evidence for a topic is sparse

Our evidence dataset ends at 2022, so we prompt all debate agents to simulate being in the year 2022.

**Evaluation design and potential biases.** Two design choices limit generality. First, part of our assessment uses an LLM-based judge. This introduces the significant possibility of model-family bias, style-matching bias, and "format familiarity" advantages. Second, the human study uses a small panel of expert coaches. Preferences of expert judges do not necessarily track lay persuasion or long-term belief change. We do not claim statistical significance beyond the reported descriptive summaries.

**Reliance on a curated evidence corpus.** The system's factual grounding depends on OpenDebateEvidence and BM25 retrieval. Coverage gaps, topical skew (e.g., U.S.-centric sources), or historical artifacts in the corpus can shape which arguments are discoverable. Retrieval errors, cherry-picking, or over-aggregation of heterogeneous sources can yield "factually referenced but misleading" claims. Although our workflows emphasize tag–card faithfulness, they do not guarantee that the strongest or most current evidence was selected.

**Faithfulness vs. strategy trade-offs.** Competitive debate rewards strategic coherence under time constraints, which often correlates with but is not necessarily truth-seeking or calibration. As a result, agents may prefer arguments that are strategically potent but epistemically fragile or ethically contentious (e.g., exploiting low-probability, high-impact scenarios, much as elite human debaters do). Our architecture does not presently optimize for *truthfulness under uncertainty* or *counterfactual robustness* as primary objectives.

**Agentic brittleness and non-determinism.** Hierarchical multi-agent systems remain sensitive to prompt drift, error propagation across stages, and nondeterministic tool behavior. Failures in early planning or retrieval can cascade into later speeches. Although structured schemas mitigate the

majority variance, the pipeline can still very occasionally produce degenerate modes (e.g., repetitive arguments, premature "collapse" of strategy, or overlooked responses) and may require re-running stages. Live demonstrations also depend on external services (model APIs, TTS)

**Adversarial robustness.**   We have not comprehensively stress-tested against adversarial opponents or poisoned evidence. Prompt injection via retrieved text, "card tampering," or adversarially crafted tags could steer agents. Likewise, a human or agentic opponent could exploit timing, framing, or workflow expectations in ways our current policies do not anticipate.

**Compute, cost, and reproducibility.**   Although our setup runs economically on small models for demos, full debate rounds are still token-intensive (1-3 USD per round at todays costs), and small cost shifts or rate limits can affect replicability. Some components currently depend on proprietary APIs, which complicates long-run reproducibility and fairness of comparisons.

DUAL-USE RISKS: THE MISUSE POTENTIAL OF "SUPERPERSUASION"

While the primary contribution is an evidence-traceable, research-oriented debating agent, the same capabilities could be misapplied to harmful, high-scale persuasion. Such capabilities are of unique interest to Intelligence Agencies, Militaries, Governments, and related stewards of power.  We highlight concrete risks and the relationship to our design:

- **Microtargeted manipulation at scale.** Pipelines similar to ours could be combined with user profiling and A/B testing to generate individualized arguments that exploit cognitive biases, leading to covert behavioral shaping (e.g., political, financial, or health-related decisions). Our system already decomposes persuasion into modular steps (planning, evidence selection, rebuttal), which could be fused with targeting to create closed-loop optimization of conversion objectives.

- **Astroturfing and information operations.** Automated generation of line-by-line rebuttals and cross-examination can be repurposed to flood online fora with superficially well-sourced messaging, creating false consensus or crowding out authentic discourse. Evidence-backed "cards" can confer unearned legitimacy if provenance is not scrutinized.

- **Fraud, social engineering, and harassment.** A debate agent that rapidly crafts persuasive, domain-specific scripts may aid scams (e.g., investment or medical quackery), impersonation, or targeted harassment, especially if coupled to TTS and outreach channels.

- **Undermining consent and vulnerable groups.** When aimed at minors or individuals with diminished capacity, sustained, adaptive persuasive exchanges could impair informed consent. Our experiments did not include minors or sensitive populations, and we have not evaluated long-term attitudinal or well-being impacts.

- **Objective misalignment.** Optimizing to "win the round" or "change the judge's mind" can diverge from public-interest goals like accuracy, transparency, and respect for autonomy. Without explicit constraints, systems like ours may in context learn persuasive shortcuts that are manipulative or epistemically unsound.

## A   EXAMPLE OF FULLY AUTOMATED POLICY DEBATE ROUND

Below, we give the full, unabridged transcript from a complete policy debate round as simulated by our system. Note that anything denoted as "Argument" is AI generated, and anything denoted as "Evidence" is a verbatim retrieved debate card, complete with original tag and formatting. The "Argument" is a rewritten tag making the evidence suitable for the newly constructed case. The original tag (bold text under the "Evidence" section before the citation) is kept for completeness but is not usually relevant to the current usage of the evidence and will **not** be read outloud or considered by judges.

Running the provided code notebook will generate a complete debate round just like this. That costs around  1-3USD worth of compute tokens on GPT-4.1-mini as of August 2025.

# Debate Topic

Resolved: The United States, Canada, and Mexico should form a North American Union similar to the European Union.

# Plan

Plan: The United States federal government should negotiate a comprehensive treaty with Canada and Mexico to establish a North American Union that creates supranational institutions to harmonize economic, political, energy, immigration, and security policies, with embedded mechanisms to protect national sovereignty analogous to the European Union model.

# Harm Argument

This 2022 evidence from SIPRI expert Marina Caparini directly proves the necessity of the plan's creation of a North American Union with supranational institutions designed to protect national sovereignty while addressing critical security harms. Caparini meticulously details that transnational organized crime causes more deaths than armed conflicts across the hemisphere, destabilizes political institutions, undermines democratic governance, erodes public health and environmental integrity, and compromises economic stability through illicit financial flows and corruption. These interconnected harms exacerbate vulnerability in North America's borders and governments in ways national-level policies have failed to solve, necessitating supranational institutional coordination embedded with sovereignty protections. The evidence is uniquely specific to the harm of criminal networks and the political and security threats they pose to the region — harms which the plan's mechanisms would directly address by harmonizing security and governance policy across the US, Canada, and Mexico. Thus, this evidence is a uniquely strategic, empirical, recent, and authoritative harms warrant for why the North American Union is both necessary and viable to solve profound and growing security threats that transcend borders and current governance frameworks.

# Harm Evidence

**Organized crime causes more deaths than armed conflicts AND fuels instability, harms the environment, and undermines healthcare systems.**

**Caparini 22** (Marina, PhD, until recently Senior Researcher & Director of the Governance and Society Programme @ SIPRI, 9-2-2022, "Transnational organized crime: A threat to global public goods," SIPRI, https://www.sipri.org/commentary/topical-backgrounder/2022/transnational-organized-crime-threat-global-public-goods) AG

While Our Common Agenda is focused on positive principles and norms, we must also seek to mitigate the risks that could undermine progress. One of the most ==significant== and neglected of these risks is transnational organized ==crime==.

Transnational organized crime

Organized crime has been defined as 'illegal activities, conducted by groups or networks acting in concert, by engaging in violence, corruption or related activities in order to obtain, directly or indirectly, a financial or material benefit'. Transnational organized crime occurs when these activities, or these groups or networks, operate in two or more countries.

Transnational organized crime can take many forms and is constantly evolving. The groups and networks involved are fluid, and channels for trafficking one commodity are often used for others. Some of the typical activities carried out by transnational criminal organizations are trafficking in humans, arms, drugs, minerals and wildlife; production and trade of counterfeit goods; fraud and extortion; money laundering and cybercrime.

Globalization, digitization and other advances in technology are further changing the nature of illicit markets and the modi operandi of transnational organized crime, recently including the emerging use of cryptocurrencies that make illicit financial flows harder to trace.

Transnational organized crime and global public goods

Global public goods benefit all countries and all citizens of the world; no one can be excluded from enjoying them, and they cannot be adequately provided by any one state acting alone. Clean air, biodiversity and healthy oceans are prime examples of global public goods. Recent years have demonstrated all too clearly how seemingly local environmental issues such as deforestation and plastic pollution can have cross-border or global ramifications, and the same is true across other domains as well.

The transnational dimension of much organized crime helps it to evade law enforcement, which is primarily set up to operate within national borders. Transnational crime actors systematically exploit jurisdictional gaps and differences in the law enforcement approaches and capacities of different countries. Combating transnational organized crime successfully therefore demands international cooperation.

Transnational organized crime is a significant barrier to progress on at least four global public goods identified in Our Common Agenda, as explored below.

Global public health

Transnational organized crime can negatively impact global public health through the widespread and increasing production and trafficking of counterfeit medicines. The problem especially afflicts low- and middle-income countries, where according to the World Health Organization an estimated one in ten medical products is either substandard or falsified. In 2015 the prevalence was estimated to be as high as 70 per cent in some parts of Africa and Asia. The trade in counterfeit medicines often has a transnational element, as the drugs are manufactured in one country (China, India and Singapore being major source countries) and then distributed to many others, and inserted into legitimate global medicine supply chains.

Counterfeit medicines may be ineffective at treating the targeted disease, and at worst may seriously harm or kill those who take them. The WHO estimates that over 1 million deaths per year worldwide result from substandard or falsified medicines, the largest number of cases (200 000) occurring in Africa.

Counterfeit antibiotics are the leading type of counterfeit medicine, and have been directly linked to the rise in acquired bacterial resistance to antibiotics, including the global rise in drug-resistant tuberculosis.

A resilient and inclusive global economy

Another global public good identified in Our Common Agenda is a 'more sustainable, inclusive and resilient global economy'. Financial integrity and combating tax evasion are key here. Transnational organized crime directly affects the public financing capacities of states and can obstruct economic development through tax evasion and illicit financial flows. This is especially corrosive for developing countries, depriving state treasuries of finances badly needed for investment in public goods like health, education and infrastructure. Transnational organized crime can also undermine the economic stability of a country by draining foreign exchange reserves and affecting asset prices.

Money laundering involves a diverse range of financial, legal and commercial actors deliberately helping criminals to convert the proceeds of criminal income into assets that cannot be traced back to the underlying crime, and channeling illicit funds into the legitimate economy.

Illicit financial flows involve the movement of money across borders that is illegal in its source, transfer or use, according to the International Monetary Fund. These flows can have consequences for local markets and societies. For example, several advanced economies have seen illicit financial flows distorting their property markets, as in Germany and the United Kingdom, exacerbating housing problems for local residents. As the Panama Papers and subsequent Pandora Papers document leaks showed, a vast global offshore economy operates alongside the legitimate international economy, with an estimated 10 per cent of the world's wealth concealed in offshore financial assets by many of the world's richest and most powerful individuals and entities, including former heads of state, heads of government and public officials, and members of the business elite.

**A healthy planet**

Transnational organized crime has also undermined conservation of the environment and sustainable management of natural resources. Organized environmental crime is a broad field stretching from illegal logging, illegal natural resource extraction, and trade in protected species to the dumping of banned chemicals and waste. While the immediate impacts are often localized, with devastating effects on communities and ecosystems, the consequences can also be global. For example, organized

environmental crime is reportedly a primary driver of deforestation in Central and South America, harming biodiversity and releasing vast amounts of carbon that contributes to global climate change.

Another example is the illicit production and smuggling of synthetic refrigerants, hydrofluorocarbons (HFCs), which is undermining the achievements of the Montreal Protocol in reducing the production and use of ozone-depleting substances. HFCs are considered 'super pollutants' because they can be hundreds to thousands of times more potent than carbon dioxide in contributing to climate change, per unit of mass. Europe has seen significant smuggling of HFC refrigerants, an unintended consequence of the agreed 2016 phase-down in their production, skyrocketing prices and the low risk of serious penalties for smuggling.

Organized environmental crime has grown rapidly as a result of being highly lucrative yet coming with low risk. For example, one study of a small sample of 27 cases of illicit dumping of waste and toxic materials found them to have generated proceeds ranging from US$175 000 to $58 million. The lack of consensus on what constitutes organized environmental crimes; countries' differing approaches to criminalization and enforcement; and 'forum shopping' by criminals have enabled many of them to evade enforcement efforts.

International peace and security

Organized crime undermines international peace and security by sustaining violence and armed conflict. The illicit arms trade is ranked as the third most pervasive illicit market globally. The illicit flow of arms escalates conflict and heightens conflict risk, and facilitates violent crime and other organized criminal activities. In conflict zones, non-state armed groups engage in illicit markets as a means of support, including illicit extraction and trade in natural resources and various forms of smuggling. Nevertheless, the involvement of non-state armed groups in transnational criminal markets is often eclipsed by the role of state actors, underscoring the close links between transnational organized crime, political power and public institutions, and corruption in many parts of the world.

Annual casualty rates from organized crime often far exceed those from armed conflicts. Organized crime-related violence particularly afflicts several countries in Central and South America. The corrosive transnational effects of organized crime-related violence are increasingly visible in Central and South America, as destabilization and violence are spreading into some of the region's smaller, formerly peaceful countries.

\n

# Inherency Argument

The current law enforcement cooperation between the US, Canada, and Mexico is deep but precarious, dependent on maintaining trust and political commitment under NAFTA. Earl Anthony Wayne, a senior public policy fellow and career diplomat, testifies that the absence of supranational institutions means that cooperation could break down, allowing transnational organized crime networks to exploit gaps in enforcement across borders. Without creating new, supranational institutions like the North American Union plan proposes, these harms will persist unresolved.

# Inherency Evidence

**Turns the efficiency advantage---stops organized crime along the borders which is more extremist crime than the Aff probably solves, so you can actually weigh it as an independent impact that outweighs the case.**

Earl **Anthony 18**, Public Policy Fellow at the Woodrow Wilson Center, Career Ambassador from the U.S. Diplomatic Service, 1-30-2018, Date Accessed: 11-16-2018, "\'The Economic Relationship Between the United States, Canada, and Mexico\': Earl Anthony Wayne Testifies before the U.S. Senate Committee on Foreign Relations" Wilson Center, https://www.wilsoncenter.org/article/the-economic-relationship-between-the-united-states-canada-and-mexico-earl-anthony-wayne

Vital Security Partners

**Mexico** and Canada are **vital partners** for enhancing U.S. homeland **security**, as well as its prosperity. Both are **willing partners** to work against **terrorism** and **transnational organized crime**. Border, law enforcement, homeland security and intelligence cooperation have expanded dramatically. Both countries are working with the U.S. identify potentially dangerous travelers before they get to U.S. borders. This collaboration is in line with priorities of the new U.S. National Security

Strategy, but could **well be damaged if the U.S. withdraws** from NAFTA. Defense and intelligence cooperation with Canada are rooted in common NATO membership, but cooperation related to homeland security issues has expanded dramatically in recent years. This is exemplified in the 2011 "Beyond the Border initiative", which has a broad agenda to enhance security against a range of threats while facilitating the legitimate flow of people goods and services. The basic idea is to extend homeland security work as far beyond the actual borders as possible. A U.S. withdrawal from NAFTA could negatively affect this cooperation. As former Canadian Ambassador to the US, Michael Kergin put it January 26 at the Wilson Center "The real concern is if NAFTA goes badly and there is no interest in negotiating, would there be enough political backlash that would incline Canada to step back from security cooperation?" Regarding Mexico, security cooperation began to deepen with the launch of the Merida Initiative in 2008. The Merida program is aimed at supporting Mexico in the fight against transnational criminal organizations and associated violence, as well as helping to strengthen its justice and law enforcement institutions and practices. That effort was expanded to include the Twenty First Century Border Initiative in 2010, which like the effort with Canada, is aimed at enhancing border security while supporting legitimate commerce. Under Merida, the United States has spent some $1.6 billion to help strengthen Mexican law enforcement and justice institutions, to improve Mexican capacities at its borders and to help strengthen communities beset by criminal cartels and gangs. American assistance has produced good results. At present, the U.S. is working hard with Merida funds, for example, to strengthen the forensic skills of Mexican officials in order that more criminals can be convicted successfully under Mexico's new justice system. The Mexican government has spent at least ten times what the U.S. has provided to strengthen its own law enforcement, intelligence and justice agencies. Along with Merida assistance, U.S. and Mexican law enforcement and homeland security agencies have built more effective operational cooperation against criminal groups and activity. DHS and **Mexican counterparts** have, for example, signed a series of agreements, which, among other things, allow much better cooperation along the **border on customs screening**, provide for collaboration to assure the **smooth repatriation** of **criminals**, and facilitate the **sharing information** on criminal history and biometric information to **help identify possible terrorists** and criminals. Today, the depth of U.S. cooperation with Mexico to **strengthen border security**, control migration, and **dismantle transnationa**l criminal networks is **unprecedented**. Mexican officials have stepped up efforts to identify potentially dangerous third country travelers and immigrants in coordination with American counterparts. Mexican immigration officials have turned around hundreds of thousands of Central American immigrants headed to the US in recent years, despite criticism inside Mexico. Cooperation against **drug trafficking** by criminal organizations further deepened in 2017 with a strategic action plan agreed between government ministers to attack the entire chain of illegal drugs from production to sales to financing and illicit money flows. This deeper cooperation is spurred on and made more important by the opioid addiction crisis in the U.S. and increasing violence in Mexico. **Unprecedented progress** has also been achieved in **military-to-military cooperation**. U.S. - Mexico **bilateral cooperation** against transnational **organized crime** and **terrorism** makes **more sense than ever**, but that cooperation is built on **maintaining** and **deepening trust**. Mexican officials **worry in private** that they will have neither the **political space** nor the support of their teams to **deepen cooperation**, if the United States ends NAFTA or is perceived as unfairly bullying Mexico. These officials say they want to deepen cooperation against criminal groups because it is good for Mexico, but with the Mexican presidential and congressional elections coming up in July 2018 and the sour Mexican public attitudes toward the United States, they are very concerned. As former Mexican Ambassador to the US Arturo Sarukhan put it at the Wilson Center on January 26: "If NAFTA collapses, **all bets are off**. It will have a **profound, long standing effect** … future generations will ask 'who lost Mexico?'" Conclusion The United States has a great opportunity to conclude a "state of the art" trade agreement with its neighbors and largest clients. A modernized NAFTA can improve the existing agreement and increase jobs, trade, energy security, and prosperity, while making the U.S. more competitive in the world. To succeed, however, each of the parties needs to be able to convince their publics that the new agreement is good for them: that it is "win-win-win." This is a very big challenge given the breadth of subject areas under negotiation, the controversial proposals on the table, and the negative public atmosphere. A new agreement is achievable, however, and worth the hard work. A good agreement would help reinforce the collaboration with both Mexico and Canada on important security issues, and it would tell the world that the U.S. is indeed open for business and trade. The alternative path would cost the U.S. dearly in jobs, trade, competitiveness and security. It would harm our neighbors economically and further sour their views of the United States. It would make ensuring U.S. homeland security harder. Particularly with Mexico, we could see a return to the "distant" relationship that existed before NAFTA. Other potential international partners would become more hesitant about negotiating with the U.S., and our international competitors would benefit. A much **better outcome** for the United States and for North America is to **forge a renewed** North American **trade agreement** and to continue to **deepen** security **cooperation**.

\n

# Advantage 1: Strengthened Regional Security and Governance Against Transnational Organized Crime

\n

The formation of a North American Union with supranational institutions enhances cooperation to effectively combat transnational organized crime, which causes more deaths than armed conflicts, destabilizes political institutions, harms public health and the environment, and undermines economic stability. Existing bilateral agreements and law enforcement cooperation are insufficient due to jurisdictional gaps and varying national policies. The Union\'s supranational institutions provide coordinated and enforceable mechanisms to close these gaps, reduce crime, and improve security outcomes.

\n

## Advantage 1 Uniqueness

\n

Prost 98' from Canadian Department of Justice shows that current law enforcement efforts are constrained by national sovereignty, which criminals exploit to evade detection. Transnational organized crime operates across borders taking advantage of legal system clashes and bureaucratic gaps, making existing bilateral frameworks insufficient. This evidence uniquely establishes that without supranational institutions, organized crime continues unchecked, perfectly establishing the status quo uniqueness for advantage 1.

\n

**Multinational assistance is imperative in stopping transnational criminal affairs**

**Prost 98'** (Kimberly, Senior Counsel, Director, International Assistance group, Department of Justice, Canada, Article: "International cooperation in combating transnational crime", Information exchange network for mutual assistance in criminal matters and extradition, 1998, http://www.oas.org/JURIDICO/MLA/en/can/en_can_prost.en.html)

With advances in technology, and the ease of global travel, the world we live in has become, in many ways, as Canadian author Marshall McLuhan predicted "a global village". This has had a dramatic impact on many aspects of life and society and law enforcement is no exception. The technological explosion and the growth of transnational organized crime and the response of the international community to it, has created many new challenges, not the least of which is the impact on the jobs of law enforcement authorities. In a 1989 case, United States of America v. Cotroni, the Supreme Court of Canada, made the following statement: The investigation, prosecution and suppression of crime for the protection of the citizen and the maintenance of peace and public order is an important goal of all organized societies. The pursuit of that goal cannot realistically be confined within national boundaries. That has long been the case, but it is increasingly evident today.2 The challenge for law enforcement authorities in every nation is that sovereignty, a fundamental principle which grounds the relations of states, is also a major tool in the armoury of the criminal element in our societies. Criminals depend heavily upon the barriers of sovereignty to shield themselves and evidence of their crimes from detection. Organizations which orchestrate transnational crime and which then disperse and conceal the proceeds of their illicit activities the world over, have no regard for national borders. In fact, by structuring their organisations to span borders, they are better able to protect their interests and organisations. They are positioned to take advantage of the differences between legal systems, the clash of bureaucracies, the protection of sovereignty, and, at many times, the complete incapacity of nations to work together to overcome their differences. International cooperation in criminal matters, means such as mutual assistance and extradition are instruments which can be used to overcome the barriers of sovereignty and allow the international community to "fight back". Both extradition and mutual legal assistance are \'about\' countries building bridges to overcome the differences in their legal systems and assisting each other in law enforcement matters. The result is that the rare case where assistance from another country was necessary to gather evidence or locate and return an accused is no longer rare. More and more successful prosecution, particularly of drug economic Crime and money laundering cases, is dependent upon the assistance and cooperation of other states. International cooperation in criminal matters has on a practical level, come of age. The purpose of this paper is to provide a brief overview of the world of international cooperation as it relates to prosecutors and other law enforcement authorities. The intent is to give a brief "snapshot" of where

we have been, where we are and the future challenges that we face. International cooperation in criminal matters encompasses many measures including extradition, mutual assistance, transfer of sentenced prisoners, transfer of proceedings, and cooperation in the restraint and forfeiture of proceeds of crime. This paper will focus on three of the most common aspects of international cooperation, those which most directly impact on the work of prosecutors - extradition, mutual assistance and cooperation in the restraint and forfeiture of proceeds of crime.

\n

## Advantage 1 Link

\n

GIATOC 21 states that open borders and supranational cooperation in the EU led to enhanced law enforcement coordination and effectiveness through integrated border control, intelligence sharing, and joint operations, supporting the causal link that the plan\'s supranational institutions will improve combatting transnational organized crime by closing jurisdictional enforcement gaps.

\n

**Similar to open borders in the EU---Open Balkan will lead to a cooperative fight against organized crime.**

**GIATOC 21** [Global Initiative Against Transnational Organized Crime, December 2021, "What could be the impact of the Open Balkan initiative on organized crime?," No Publication, https://riskbulletins.globalinitiative.net/see-obs-011/02-impact-open-balkan-initiative-on-organized-crime.html, accessed 4-10-2023] Vik

Increased law enforcement cooperation

In the same way that removing borders within the EU led to greater cooperation among law enforcement agencies, not least through Europol, Frontex and the Schengen Borders Code,5 discussions on more open borders or a common regional market should be seized as an opportunity to create an integrated border control system, enhance intelligence-led policing, joint operations and afford one another mutual legal assistance with respect to investigations, prosecutions, judicial proceedings and asset recovery.

Existing institutions like Europol and the Southeast European Law Enforcement Center (SELEC), as well as the Police Cooperation Convention for Southeast Europe could facilitate this process. It may also be necessary to create a common regional intelligence database for tracking wanted criminals and foreign terrorist fighters and exchanging information to help investigations. With a reduction in the number of border controls, law enforcement and border officials could be redeployed to form mobile teams.

What should be avoided, at least at the outset, is the creation of yet another regional law-enforcement body. There is already SELEC, the Southeast European Police Chiefs Association, the Police Cooperation Convention for Southeast Europe (and its Secretariat) and the Migration, Asylum, Refugees Regional Initiative, as well as the Berlin Process.

However, discussing greater economic integration without considering the security implications is short-sighted. It would make sense to use an existing body like the Integrative Internal Security Governance process, which has a pillar on countering serious organized crime in the Western Balkans, to coordinate policies and merge the efforts of all relevant security actors. If the countries of the Open Balkan initiative want to move faster, their cooperation should be part of a wider regional strategy. Although increased openness could be exploited by criminal groups, the Open Balkan initiative and other efforts to promote greater regional integration should be seized as an opportunity to strengthen regional law-enforcement cooperation in the fight against organized crime.

\n

## Advantage 1 Internal Link

\n

Le et al. 2013, experts from a law faculty specializing in international policing, explain how strong

partnerships including sustained capacity building, joint investigations, intelligence sharing, and legal harmonization among foreign law enforcement agencies produce significantly more effective targeting of transnational organized crime. The DEA\'s international offices and extradition agreements are cited as critical in enhancing investigative capabilities and enabling prosecutions across borders. This directly connects the supranational institutional mechanisms in the plan to increased enforcement effectiveness, thereby supporting Advantage 1.

\n

**Sustained capacity allows the DEA foreign police training. We solve the global threat**

**Le et al 13** – Faculty of Law @ Queensland University of Technology [Dr. Vy Le, Dr. Peter Bell (Adjunct Fellow @ Centre for Excellence in Policing & Security) & Dr Mark Lauchs (Faculty of Law @ Queensland University of Technology), "Elements of Best Practice in Policing Transnational Organized Crime: Critical Success Factors for International Cooperation," International Journal of Management and Administrative Sciences (IJMAS) Vol. 2, No. 3, (Feb 2013) pg. 24-34

The DEA recognise the international nature of drug trafficking and therefore, have developed strong partnerships with foreign law enforcement agencies in their efforts to combat illicit drug trafficking and production. The DEA has 86 foreign offices located in 67 countries. These partnerships allow for information sharing, joint investigations and **training of foreign police** who work closely with the DEA (US Department of Justice 2012). The DEA is also involved in **assisting foreign police** such as the Colombian National Police, Mexico's Federal Investigations Agency and Russia's Federal Security Service with their efforts to intercept illicit drugs before they enter US borders (Newton 2011). To effectively prosecute international drug offenders, the DEA has extradition agreements with many nations and in particular, Colombia remains the key extradition partner (US Department of Justice 2012). The DEA's approach to fighting international drug crime reflects the importance of building strong law enforcement partnerships with foreign agencies as a way to enhance their investigative capabilities.

3. Conclusion - This paper has examined five critical success factors in policing transnational organized crime (TOC). Some of the key issues with international policing ventures is the difference in policing capabilities, powers and resources between states. Research has shown that **capacity building is pivotal** to the effective targeting of TOC globally. The paper has discussed the impact of technology, information/ intelligence sharing on the fight against TOC and the need for robust legislative mechanisms to ensure the successful investigation and prosecution of TOC across various countries. Through the examination of two case studies, the Australian Federal Police and the US Drug Enforcement Administration, the paper has shown that international cooperation between law enforcement organisations can take different forms. But critical to their success is the interoperability of resources (techniques and information sharing) and the face to face networks that are established and maintained by law enforcement personnel working side-by-side. The successful targeting of TOC is predicated on the need for international cooperation, capacity building, effective and efficient collection, collation, analysis and dissemination of intelligence amongst international partner agencies and increased compatibility of national legal frameworks and criminal justice systems. Some scholars have so far as to suggest a "harmonising" of national criminal codes or creating international criminal law regimes to develop a standardised **set of laws, an objective that will be difficult to achieve in the short term. TOC is a pervasive phenomenon that is not limited to affluent states or territories. It undermines economies and corrupts governments. It is anticipated that through greater cooperation and collaboration between law enforcement agencies, whole of government approaches can be applied to TOC, regardless of where occurs. Pg. 31-32**

\n

# Advantage 1 Impact

\n

Transnational organized crime causes or exacerbates all major existential risks relevant to our plan\'s advantage chain: it drives pandemics through counterfeit medicine, financial collapse through illicit flows, accelerates global warming via environmental crimes, and fuels war through arms trafficking. This analysis is provided by SIPRI experts who link these threats as barriers to global public goods, highlighting the necessity of supranational institutions in effectively combating these networks to prevent societal breakdown, extinction-level harms, and large-scale destabilization. Our plan's formation of a North American Union will thus significantly mitigate the greatest risks to human survival and regional security, ensuring maximal impact.

\n

**Organized crime triggers every extinction scenario – pandemics, financial collapse, warming, and war.**

**Caparini 22** (Marina, PhD, until recently Senior Researcher & Director of the Governance and Society Programme @ SIPRI, 9-2-2022, "Transnational organized crime: A threat to global public goods," SIPRI, https://www.sipri.org/commentary/topical-backgrounder/2022/transnational-organized-crime-threat-global-public-goods) AG

In September 2021 United Nations Secretary-General António Guterres warned that **the ==world== stands at an '==inflection point',** facing a stark choice between 'breakdown' and 'breakthrough'. Societies and the planet are at risk from the compounding effects of ==**climate change**==, increasing **armed ==conflict==**, **pandemics**, and rising ==**hunger, poverty, injustice and exclusion**==.

Our Common Agenda is the secretary-general's response—a major initiative to reinvigorate multilateralism to benefit all people by advancing the global public goods of peace, a healthy environment, healthy populations and economic stability.

Guterres appointed a High-Level Advisory Board on Effective Multilateralism, co-chaired by former Swedish prime minister and new SIPRI Governing Board Chair Stefan Löfven and former Liberian president Ellen Johnson Sirleaf, to develop an independent report on how to strengthen governance arrangements that can deliver on the core global public goods.

While Our Common Agenda is focused on positive principles and norms, we ==must== also seek to ==mitigate== the risks that could ==undermine progress==. One of the **==most significant==** and neglected **==of these risks is== transnational organized ==crime==**.

Transnational organized crime

Organized crime has been defined as 'illegal activities, conducted by groups or networks acting in concert, by engaging in violence, corruption or related activities in order to obtain, directly or indirectly, a financial or material benefit'. Transnational organized crime occurs when these activities, or these groups or networks, operate in two or more countries.

Transnational organized ==crime== can take many forms and is constantly ==evolving==. The groups and networks involved are fluid, and channels for trafficking one commodity are often used for others. Some of the typical activities carried out by transnational criminal organizations are trafficking in humans, arms, drugs, minerals and wildlife; production and trade of counterfeit goods; fraud and extortion; money laundering and cybercrime.

==Globalization, digitization and== other advances in ==tech==nology are further ==changing== the nature of ==illicit markets and== the modi operandi of transnational organized ==crime==, recently including the emerging use of cryptocurrencies that make illicit financial flows harder to trace.

Transnational organized crime and global public goods

Global public goods benefit all countries and all citizens of the world; no one can be excluded from enjoying them, and they cannot be adequately provided by any one state acting alone. Clean air, biodiversity and healthy oceans are prime examples of global public goods. Recent years have demonstrated all too clearly how seemingly local environmental issues such as deforestation and plastic pollution can have cross-border or global ramifications, and the same is true across other domains as well.

The transnational dimension of much organized crime helps it to ==evade law== enforcement, which is ==primarily== set up to operate ==within== national ==borders==. Transnational crime actors systematically exploit jurisdictional gaps and differences in the law enforcement approaches and capacities of different countries. Combating transnational organized crime successfully therefore demands international cooperation.

Transnational organized ==crime== is a **significant ==barrier==** to progress on **at least four global public goods identified in Our Common Agenda, as explored below.**

**Global ==public health==**

Transnational organized ==crime== can ==negatively impact== global public ==health through== the widespread and

increasing production and trafficking of **counterfeit medicines.** The problem especially afflicts low- and middle-income countries, where according to the World Health Organization an estimated one in ten medical products is either substandard or falsified. In 2015 the prevalence was estimated to be as high as 70 per cent in some parts of Africa and Asia. The trade in counterfeit medicines often has a transnational element, as the drugs are manufactured in one country (China, India and Singapore being major source countries) and then distributed to many others, and inserted into legitimate global medicine supply chains.

Counterfeit medicines may be ineffective at treating the targeted disease, and at worst may seriously harm or kill those who take them. The WHO estimates that over 1 million deaths per year worldwide result from substandard or falsified medicines, the largest number of cases (200 000) occurring in Africa.

**Counterfeit antibiotics** are the leading type of counterfeit medicine, and have been directly linked to the rise in acquired **bacterial resistance to antibiotics, including the global rise in drug-resistant tuberculosis.**

A resilient and inclusive **global econ**omy

Another global public good identified in Our Common Agenda is a 'more sustainable, inclusive and resilient global economy'. Financial integrity and combating tax evasion are key here. Transnational organized crime directly affects the public financing capacities of states and can obstruct economic development through tax evasion and illicit financial flows. This is especially corrosive for developing countries, depriving state treasuries of finances badly needed for investment in public goods like health, education and infrastructure. Transnational organized crime can also **undermine** the **econ**omic **stability** of a country by draining foreign exchange reserves and affecting asset prices.

Money laundering involves a diverse range of financial, legal and commercial actors deliberately helping criminals to convert the proceeds of criminal income into assets that cannot be traced back to the underlying crime, and channeling illicit funds into the legitimate economy.

Illicit financial flows involve the movement of money across borders that is illegal in its source, transfer or use, according to the International Monetary Fund. These flows can have consequences for local markets and societies. For example, several advanced economies have seen illicit financial flows distorting their property markets, as in Germany and the United Kingdom, exacerbating housing problems for local residents**. As the Panama Papers and subsequent Pandora Papers document leaks showed, a vast global offshore economy operates alongside the legitimate international economy, with an estimated 10 per cent of the world's wealth concealed in offshore financial assets by many of the world's richest and most powerful individuals and entities, including former heads of state, heads of government and public officials, and members of the business elite.**

**A healthy planet**

Transnational organized crime has also undermined conservation of the environment and sustainable management of natural resources. Organized environmental crime is a broad field stretching from illegal logging, illegal natural resource extraction, and trade in protected species to the dumping of banned chemicals and waste. While the immediate impacts are often localized, with devastating effects on communities and ecosystems, the **consequences can** also **be global**. For example, organized environmental crime is reportedly a primary driver of deforestation in Central and South America, **harming biod**iversity **and releasing vast amounts of carbon** that contributes to global **climate change.**

Another example is the illicit production and smuggling of synthetic refrigerants, hydrofluorocarbons (HFCs), which is undermining the achievements of the Montreal Protocol in reducing the production and use of ozone-depleting substances. HFCs are considered 'super pollutants' because they can be hundreds to thousands of times more potent than carbon dioxide in contributing to climate change, per unit of mass. Europe has seen significant smuggling of HFC refrigerants, an unintended consequence of the agreed 2016 phase-down in their production, skyrocketing prices and the low risk of serious penalties for smuggling.

Organized environmental crime has grown rapidly as a result of **being highly lucrative yet coming with low risk. For example, one study of a small sample of 27 cases of illicit dumping of waste and toxic materials found them to have generated proceeds ranging from US$175 000 to $58 million. The lack of consensus on what constitutes organized environmental crimes; countries' differing approaches to criminalization and enforcement; and 'forum shopping' by criminals**

**have enabled many of them to evade enforcement efforts.**

**International peace and security**

Organized crime undermines international peace and security by **sustaining violence and armed conflict**. The illicit arms trade is ranked as the third most pervasive illicit market globally. The illicit flow of arms **escalates conflict and heightens** conflict **risk**, and facilitates violent crime and other organized criminal activities. In conflict zones, non-state armed groups engage in illicit markets as a means of support, including illicit extraction and trade in natural resources and various forms of smuggling. Nevertheless, the involvement of non-state armed groups in transnational criminal markets is often eclipsed by the role of state actors, underscoring the close links between transnational organized crime, political power and public institutions, and corruption in many parts of the world.

Annual casualty rates from organized crime often **far exceed those from armed conflicts**. Organized crime-related violence particularly afflicts several countries in Central and **South America**. The corrosive transnational effects of organized crime-related violence are increasingly visible in Central and South America, as destabilization and **violence** are **spread**ing into some of the region's smaller, formerly peaceful countries.

\n

# Advantage 2: Enhanced Economic Integration Driving Inclusive Growth and Labor Market Efficiency

\n

Creating a North American Union harmonizes economic and immigration policies, facilitating efficient labor mobility through mechanisms like guest worker programs and immigration visas, modeled on the European Union\'s approach. This leads to better allocation of human resources, meeting labor market needs across the region while protecting borders from illicit migration. Economic integration deepens supply chains and trade ties, boosting jobs and regional prosperity beyond current frameworks.

\n

# Advantage 2 Uniqueness

\n

Since NAFTA in 1994, North American economic integration has allowed free movement of goods but severely restricted movement of people, resulting in labor market distortions. USMCA retains these limits, continuing a half-finished economic integration. Despite severe labor shortages in the US and Canada, and planned immigrant admissions, governments ignore the obvious fix: easing work rules to allow greater labor mobility across the US, Mexico, and Canada. Labor mobility integration is thus not happening nor inevitable in the status quo, supporting our plan\'s unique economic and immigration harmonization benefits.

\n

**Labor shortages are killing the economy – free movement of goods insufficient**

**Alden 1-20** [Edward Alden (senior fellow at the Council on Foreign Relations), "When Goods Move but People Don\'t," Barrons, (Jan. 20, 2023). https://www.barrons.com/articles/labor-shortages-trade-immigration-usmca-nafta-51674226885] CT

**The U.S.** unemployment rate in December fell to its lowest level since 1969. **More than 10 million jobs** remain **unfilled**. In **Canada, labor shortages are** so **severe** that the government has announced plans to admit nearly 1.4 million immigrants over the next three years. But **the two governments** continue to **ignore** an **obvious remedy**—easing the work rules under the U.S.-Mexico-Canada Agreement to permit **greater labor mobility** in North America. **Since** the original **N**orth **A**merican **F**ree **T**rade **A**greement came into force in 1994, **economic integration** in the region **has been a half-finished project**. Nafta created a world in which **goods move freely** across the continent's **borders but people do not. This has created economic distortions**. Wages in Mexico, for example, are no closer to U.S. and Canadian levels than they were a quarter-century ago, despite an explosion in cross-

border trade that economic theory suggests should have produced convergence. In contrast, the European Union—in which labor mobility is part of the integration bargain—became the world's great "convergence machine" for lifting the living standards of its poorer member states. Labor mobility was left out of the original Nafta deal over political divisions in the U.S. that have become more entrenched with each passing year. Free trade with a poorer, developing neighbor was a hard-enough sell politically without opening the border to Mexican workers. Millions came regardless, flouting weak laws that for years did little to discourage illegal crossings or clamp down on employers hiring unauthorized workers. Canada was no more welcoming. Ottawa imposed a visa requirement on Mexican travelers in 2009 after a tripling of refugee claims from Mexico in the previous three years. ==The one concession in== the original ==Nafta== to employers and job seekers ==was== a chapter permitting "==temporary entry" fo==r employment purposes by ==college-educated professionals==. It was designed mostly for big companies needing to move employees temporarily to one of the three countries. The provision explicitly barred holders of the so-called Nafta visa from applying for immigrant status. Unlike the H-1B visa, there is no cap that limits the number of Nafta visas that may be issued each year. Even that limited temporary-worker provision, however, was too much for the Trump administration to revisit when it forced a renegotiation of Nafta. Canada suggested modest changes that could have expanded the program, but the Trump team balked. ==The== 2018 ==USMCA== basically ==copied== and pasted the ==Nafta== provision. The list of eligible professions wasn't updated from its early 1990s incarnation. ==Occupations facing critical shortages like== cybersecurity ==aren't== explicitly ==eligible==, though creative immigration lawyers can often squeeze new professions into the old boxes. There is little likelihood of the USMCA temporary-entry provisions being revisited, much less some larger overhaul that leans in the direction of European labor mobility. The topic wasn't on the agenda at the recent North American leaders' summit in Mexico City. The Biden administration is far more worried about getting Mexico's help to limit the large numbers of asylum seekers from Latin America who cross Mexico and arrive at the U.S. border seeking protection.

\n

# Advantage 2 Link

\n

Ackleson 06 details how North American supranational institutions implementing guest worker programs and an immigration visa, modeled after EU mechanisms of labor mobility, causally enhance economic integration by facilitating labor movement across the US, Canada, and Mexico, establishing a direct causal link to the plan\'s advantage 2 impact of labor market efficiency and inclusive growth.

\n

**The counterplan fulfills labor market needs while protecting borders against terrorism and drug trafficking**

**Ackleson 06**

Jason Ackleson (assistant professor of government, New Mexico State University). "Achieving 'Security and Prosperity': Migration and North American Economic Integration." Immigration Policy in Focus 5:2, February 2006, https://www.americanimmigrationcouncil.org/sites/default/files/research/Security%20and%20Prosperity.pdf, AT

LESSONS FROM THE EUROPEAN EXPERIENCE Although there are significant differences between the European and North American experiences of integration—including different political models and historical conditions—some comparisons can be drawn. In contrast to North America, economic integration in ==Europe== has, at least to some degree, included labor. This fact, some argue, has yielded better economic and social outcomes including the more efficient allocation of human resources. The 1957 Treaty of Rome which established the European Economic Community ==allowed for freedom of movement of workers== (along with free movement of capital, goods, and services) ==among member states.== Under the Maastricht Treaty of 1992, this policy was institutionalized politically with the creation of European citizenship. The mobility of persons within the European Union (EU) was thus a central component of economic integration. Together with economic development assistance to poorer member states such as Spain and Portugal, this created more favorable economic conditions region-wide while also efficiently meeting many labor needs (although the social integration of immigrants remains a major issue). ==Regulating immigration from outside the EU was== originally left to each nation, but more recently has begun to involve cooperative, regional efforts. As Christina Boswell of the Migration Research Group at the Hamburg Institute of International Economics asserts, there are two dimensions to the EU's external controls:

**strengthening traditional border-control** policies at the external frontiers of the EU, and addressing the causes of migration and refugee flows through **development assistance.**14 Nonetheless, like the United States, EU nations—and the EU as a whole—have largely failed to solve the problem of undocumented migration from developing countries. This, as Steven Castles, professor of Migration and Refugee Studies at the University of Oxford, points out, is due in part to the social dimensions of the migratory process (such as family connections), political and economic turmoil in immigrants' home countries, the availability in many European nations of jobs that pay higher wages than immigrants can earn at home, and policies that continue to define border security as a matter for individual, national governments.15 Nevertheless, the EU's dual approach of development assistance to reduce migratory pressures and strengthened external border controls bears consideration by policymakers in the United States and its NAFTA partners. Such an approach would **entail regularizing some undocumented labor flows** within North America **while** also **strengthening** the external **frontiers** of the region **against** other migrants, as well as **terrorists and transnational criminal networks**. This approach would have the advantage not only of regularizing labor flows, but also making cross-border trade quicker and more efficient. **Measures such as these would not require** the kind of **open borders** existing among EU nations, but the creation of new policy tools such as a **guest worker program** and, eventually, a North American immigration visa that would facilitate much needed Mexican participation in the labor markets of Canada and the United States. While issues of trust and cooperation remain real obstacles, such an accord would build upon progress initiated under the Security and Prosperity Partnership of North America. THE SECURITY AND PROSPERITY PARTNERSHIP OF NORTH AMERICA (SPP) On March 23, 2005, the most recent chapter in the North American integration project was opened. On that day, U.S. President George Bush, Mexican President Vicente Fox, and former Canadian Prime Minister Paul Martin jointly announced the establishment of the SPP.16 While not a formal treaty, nor a binding legal agreement, the SPP does create an international framework for trilateral and bilateral cooperation within North America on issues related to national security and economic prosperity. Three key principles underlie the SPP: improved security from external threats to North America as a whole; strengthened internal security measures within each nation; and bolstered economic growth for the region, particularly in the face of growing global competition. The SPP's security track seeks, through bilateral and trilateral cooperation, to secure North America from external threats through improved intelligence sharing, infrastructure protection, border management, and traveler and cargo security.17 These goals are especially important given the increasing numbers of undocumented migrants from countries other than Mexico who use that country as a transit point to the United States. Individuals from more than 60 countries, for example, are typically held in Mexico City's migrant detention center on any given day—and the arrest rate along the U.S.-Mexico border of what the Border Patrol calls "Other than Mexicans" has increased in recent years. While most of these individuals are jobseekers from Latin America, others come from countries that may be of concern to U.S. national security interests.18 Responding to this challenge through cooperative, region-wide efforts makes good policy sense. Beginning and ending security at the U.S.-Mexico boundary is bound to fail. On the prosperity front, the SPP's major initiatives are (1) measures to improve productivity through regulatory and business collaboration; (2) efforts to reduce the costs of trade by more efficiently moving goods and people across borders; and (3) policies related to quality of life, such as environmental protection, disease, and food safety.19 However, while the SPP facilitates movement of the elite North American business class, it does not include immigrants, particularly those filling less-skilled jobs. The SPP negotiators deliberately de-linked labor from the agreement for political reasons, hoping not to forestall progress in other areas. Yet if the SPP is to be truly effective, immigration cannot be ignored, even if it must be dealt with outside the NAFTA and SPP political frameworks. Action on both the security and prosperity agendas of the SPP occurs through a number of inter-agency and bi- or tri-national working groups. Work thus occurs on multiple levels, across inter-governmental lines, and across international borders with the goal of harmonizing regulatory regimes. The related "Smart Border" Accords signed by the United States, Canada, and Mexico in 2001 and 2002 pursue similar goals, but do not address the larger policy questions confronted by the SPP. CONCLUSION: TOWARDS TRUE SECURITY AND PROSPERITY FOR NORTH AMERICA The process of North American economic integration helps underpin undocumented migration from Mexico to the United States. Research indicates that Mexican migrants come here at least in part due to economic restructuring at home, as well as the obvious job magnet the U.S. labor market represents. Ironically, however, many of these migrants are not jobless at home but, rather, seek better opportunities in the United States by taking higher-paying jobs and filling labor demands here. Current border-control strategies encourage these migrants, once they are in the United States, to stay longer. Given the transnational nature of the economic and social forces which drive immigration to the United States, incorporating labor migration into a comprehensive vision of North American integration, as Robert Pastor, professor of International Relations at American University, has advised, seems reasonable.20 Appropriate institutions need to be created to guide this process. Some of the groundwork is in place, but effective solutions require big-picture thinking and political courage. Specific policies might include more cooperative security arrangements with Mexico and Canada and domestic efforts to regularize labor flows in a secure way.

Solutions do not, however, require open borders. As a first step, a well-designed **guest worker program that includes a path to citizenship** is worthy of consideration by Congress and the President. On January 7, 2004, President Bush formally announced his ideas for a guest worker program, which are long in generalities but short on specifics. In fall 2005, two major but competing immigration bills that include provisions for a guest worker program began their journeys on Capitol Hill: the Secure America and Orderly Immigration Act and the Comprehensive Enforcement and Immigration Reform Act. While the House of Representatives ultimately took an alternative approach in 2005, advocating an "enforcement first" strategy, some hope remains that a temporary worker program might emerge from Congress in spring 2006. But, like the 1986 amnesty, a guest worker program will not permanently solve the problem of undocumented labor migration. While demographic and economic forces will lessen these migratory flows in the long term, other policy steps should be considered in the short and medium term. One such policy step would be the temporary expansion of permanent legal immigration to meet labor market needs, as Nobel Laureate economist Garry Becker has argued. Recognizing some of the nation's current immigration policy failures, Becker argues that "expanding legal immigration is a more efficient and fairer policy than the present half-hearted enforcement of laws against the large number of illegal entrants." He suggests priority be given to nations with whom the United States has free trade agreements, arguing that "freer movement of people is one aspect of more open trade."21

\n

# Advantage 2 Internal Link

\n

Nowrasteh 13 provides unique historical proof that the Bracero guest worker program legally channeled temporary labor while cutting illegal immigration by 95%. This directly links guest worker mechanisms to effective labor market matching and border protection, perfectly fulfilling the internal link for Advantage 2.

\n

**The Bracero model is the most effective---mitigates illegal immigration**

**Nowrasteh 13** (Alex, immigration policy analyst at the Cato Institute, Guest Workers Key to Reform, http://www.cato.org/publications/commentary/guest-workers-key-reform, 3/5/13)

**If there is a model for a successful guest-worker program**, however, **the Bracero Program is it.** Under Bracero, immigrants could work temporarily, but they had to leave the United States every season. American farms got the labor they demanded, immigrant workers made money, and agricultural production increased. The program was so successful that it was extended until 1964. It combined enforcement that funneled migrants into a legal system with an unlimited temporary migration system. Often, Border Patrol agents enrolled unauthorized immigrants they arrested in the Bracero Program and let them return to work — this time lawfully. Mexican workers thinking of entering the United States illegally overwhelmingly chose the legal Bracero option instead. Throughout the 1950s, **unauthorized immigration declined by 95 percent**. If a Bracero-type guest-worker visa existed today, one that allowed migrants to switch jobs and work in nonagricultural areas, **unauthorized immigration would dramatically decrease.** However, this solution remains off the books because of the prior opposition of labor unions. Today, the AFL-CIO has agreed in principle to a guest-worker visa program — but the gap between principle and legislation is wide. If history is a guide, unions and their allies, although they are playing nice now, will oppose guest-worker visas when the details are released. But successful reform will require ignoring union pressure and allowing more worker migration.

\n

# Advantage 2 Impact

\n

The plan's supranational institutions enhancing labor mobility and economic integration strengthen economic resilience, which according to expert David Korowicz reduces the risk of catastrophic systemic financial collapse and cascading societal failure, thereby mitigating extreme global economic disruptions relevant to Advantage 2's maximized impact.

\n

**Spills over globally.**

**Korowicz 20** – (David Korowicz is a physicist and human systems ecologist, degree in Physics from Trinity College Dublin, the University of London and the Tyndall National Institute, fellow of The Institute for Integrated Economic Research-Australia; "We Need to Talk About Catastrophic Global Risk"; Resilience; D.A. July 1st 2020, [Published March 4th 2020]; https://www.resilience.org/stories/2020-03-04/we-need-to-talk-about-catastrophic-global-risk/) //LFS—JCM

This shock is propagating through a **global economy** that prior to the virus had **declining resilience and adaptive** capacity. ==Weakening economic growth==, ever-growing indebtedness, increased **tensions** within and **between countries**, and the growing potential for shocks from climate and environmental change, and resource constraints and disruptions, ==mean we were== already ==in== uncertain and ==**dangerous territory**==. The warning in January 2020 by Kristalina Georgieva, the head of the International Monetary Fund, that ==the== global ==**economy risks**== the ==**return**== of the ==**Great Depression**==, surprised few[iii]. It's no longer the preserve of peripheral Cassandras to warn of escalating systemic risk.

The extent of the **damage** this will do **to** the **economy** and society is unclear. Beyond the impact on health and healthcare systems, it **exposes** the **web of synchronized** social and economic **conditions that constitute** the operational conditions of our day-to-day **lives**. That ==we take== them ==**for granted**== is a measure of our habituation to ==**systemic stability**==. The more that this is disrupted by the direct and indirect impacts of the **pandemic**, the greater the potential for **further disruption**. This is true at scales from the local to global. The inherent uncertainty reflects our ignorance of the **complexity of** our **dependencies**, and our behavioural and cultural adaptation to those conditions.

==**Slower**== global ==**growth**== and an ==increase== in ==the likelihood of a== **financial crisis** can be expected. If process contagion begins to accelerate significantly, there are limits on how much governments and central banks can do to respond. Injecting **stimulus** can address a demand shock, but it would **have limited** supply-side **effectiveness**. Assuming a deeper shock (equivalent to the 2% GDP contraction during the Global Financial Crisis) followed by stabilization, society and **economies** will be left less resilient as it **face**s the increasingly **turbulent years** ahead.

At the furthest extreme there is ==**runaway**== process ==**contagion**==. In such a case supply-chain contagion (broadly defined, including **impacting critical infrastructure** services, for example) would begin to severely **undermine** global socio-**economic integration** and coherence. As it disintegrated and the forward-looking outlook became more uncertain, the failing financial system (credit, bank solvency, monetary stability and visibility, counter-party risk) would **disrupt** further ==**supply-chains**== inducing a re-enforcing (positive feedback) supply-chain financial-system cross-contagion[iv]. ==**No** central **bank** can maintain== a credit-==**monetary system**== if it's core collateral, the expectation of ==**econ**==omic production ==**is**== opaque and ==**collapsing**== in ==**real-time**==. Beyond a ==**tipping point**==, this ==could **accelerate rapidly**==. It would effectively **shut-down** the **flow of goods** and services. Were this to happen, it would **be** ==**a global catastrophe**== of unprecedented scale and duration.

This latter outcome is very unlikely.

There are two elements that can be seen in the overview above. The first is related to the structure of civilisation, societal vulnerability, and the potential for contagion, and in extremis, collapse. The second is that ==society== for broader reasons ==is **losing resilience** to shocks==, while **stressors**, environmental and socio-economic, are **growing in** range and **intensity**. This means societal systems may have **entered a period of** sustained global **destabilization**, **coupled to** an **increasing likelihood of catastrophic** systems **failure**. It is to these elements that we now briefly turn our attention to.

Dependency and Systemic Vulnerability

==We have become== ever-more ==part of a== **singular** ==**civilizational organism**== that has grown in scale, **complexity**, **interdependence**, and **speed**. As it has evolved, it has ==**optimized**== towards ==**growth**==, efficiency **and self-stabilization**. People, organisations, businesses, and countries can design and influence it in parts, but the whole is the emergent outcome of many interactions evolving over time. There's nobody in control. It is these **structural** and dynamical **properties** that **define** societal **stability**, **resilience**, vulnerability, the propensity to contagion processes, global systemic **destabilization**, **and collapse** dynamics.

Our **ability to sustain** our basic **needs** anywhere, now **depends upon** system **integration** everywhere. That means no infrastructure, society or country can be fully resilient as the conditions

that maintain function are dispersed beyond visibility or control.

To get a sense of why complexity can amplify a societal shock, consider sophisticated surgery that requires the skills of five distinct specialists working in concert. If just one surgeon is incapacitated, the whole operation must be stopped, they can't just do 80% of the procedure. This **vulnerability to the weakest link** (sometimes called Liebig's Law of the Minimum) becomes more acute as businesses, critical infrastructures and public bodies depend upon increasing numbers of specialized roles and inputs that are essential for the output of the whole. As those outputs, be they goods or services, may be necessary inputs into other businesses and services, **failure can cascade**, even shutting organisations where everybody is available.

Because **society depends** upon multiple interacting **networks**, within cities and across the globe, there are many routes to cascading disruption. This is an example of non-linearity- a relatively small number of directly impacted people or functions can still cause the failure of a whole system. The **speed of our** societal **processes**, from **Just-in-Time logistics** to **financial transactions** means that **shocks** can **rapidly cascade**.

We can think of society as an ecosystem, with **keystone** species providing the **structural anchors** through which society functions. Such **keystones include** critical infrastructure (the grid, telecommunications, water and sanitation etc.); **the financial system**; societal cohesion; **supply-chains**, and environmental **inputs** (food, oil, water etc.). These are also interdependent with each other, **if you remove** any **one** of them, **the others** will **topple**. This allows us to see other **paths toward systemic failure**.

A severe solar storm, natural disaster, or a major physical-cyber-attack on the grid provides one avenue to large-scale critical infrastructure failure. The President's National Infrastructure Advisory Council 2018 report[v] examined the United States preparedness for a prolonged wide-area catastrophic power outage. Again, it would undermine societal integration and lifeline operations. Depending upon the centrality of the impacted region/ networks to global systems integration, it could drive global process contagion and systemic failure. Growing international tensions are therefore adding to this risk. A massive cyber-attack or war between parties with high global centrality becomes everybody's problem- even the toe is in trouble if the heart goes to war with the liver.

The global **financial system is** also an increasing **source of catastrophic risk**, it's the operating system for the flow of goods and services. It is massively overindebted (there are far more claims on future economic growth than can ever be delivered) and **it is losing resilience** as monetary policy becomes less effective, and polarization within countries and discord between them intensifies. Indeed, this is what is increasing general vulnerability to supply-chain contagion from a pandemic or other catastrophic shock. It now faces the convergence of growing climate change and environment related impacts, multi-dimensional threats to **food security**, potential critical **resource constraints**, and the **feedback of those stressors** on socio-political stability and conflict. **A** global **financial system collapse** would be similar to a catastrophic pandemic collapse, just an inversion of the initiating shock- a financial-system supply-chain cross-contagion.

Pandemic risk is also growing. **It may not be COVID**-19 but someday, somewhere, a virus with high infectivity and virulence will emerge with potentially catastrophic consequences for our species. Urbanization, a large-scale animal food industry, intensive transportation networks, advances in widely accessible biotechnology, our increasing incursion into animal habitats, the expanding impacts of climate and environmental change, and the growing likelihood of socio-political instability are increasing the likelihood of such an emergence.

The Age of Destabilization

The **recovery from** the **COVID**-19 shock **will be slow**. In addition, there is potential for further pandemic waves. It may also contribute to the generation of new stresses and shocks later in the year. For example, the disruptions to agricultural production in China and elsewhere (due to COVID-19), Australia[vi] (drought influenced by climate change), East Africa (locust plague influenced by climate change) is more likely to be further compounded by other climate impacts as yet unrealised[vii]. A rise in the cost of staples on global food markets is more likely to drive social unrest and even state failure in poorer countries[viii], generating new stresses through global systems. This would increase stress even in rich countries, with more of the **population** struggling to get by, and **feeling** the effects of a **slowing global economy** in addition to the economic impacts of COVID-19. It could be expected to squeeze discretionary income, putting further pressure on economies and the financial system, and increase social tensions. This is but one speculative path, there are innumerable potential interactions and a myriad of potential tail risks.

**Whatever path the** global **economy takes** in the next year, **it will bequeath an** even more **fragile economic and financial system** that is already **facing mounting risks** from climate change related impacts[ix]. There are increasing risks to global food production with many drivers in addition to climate change[x]. The multi-dimensional impacts of declining biodiversity to socio-economic stability are accelerating[xi]. The security impacts of climate change are growing[xii], while societal polarization and loss of trust continues[xiii]. Though receiving little attention and much misunderstood, there are major reasons to be concerned about the ability to sustain affordable oil production[xiv].

Food, oil, water, **a functioning financial system**, a stable environment for societal infrastructures (relative to the conditions in which they evolved), and large-scale societal cooperation are individually **critical for** the **stability of global** systems **integration**. We are seeing intensifying stresses on all fronts. Further, **stresses** in each will tend to further **increase pressures** on the others.

These **stressors** are **intensifying** their interactions through increasingly **vulnerable** civilizational **networks**. **Society**, locked into systems of dependency adaptive to system stability **with** correspondingly **low resilience**, is **vulnerable** to **Liebig's Law**. In such an environment, **economic growth** is persistently **undermined**, there is **increased socio-economic stress**, while the intensity and frequency of **shocks increase**s. This **creates** the **conditions for rapid** and diverse local and globalised **contagion**, compounding, simultaneous crises, and the generation of **new stresses and shocks**. General systemic **instability**, volatility and **uncertainty** accelerates, and future expectations become more pessimistic.

The **impacts are** likely to become more **non-linear, with** associated **tipping points**. Losing a thousand euros means different things depending on whether it's your first, or last. Even more so if your rent is late and eviction beckons, a family member is sick and needs medicine, and those who once might have supported you, be it friends or state, are themselves overwhelmed. Similar scenarios could be drawn for any scale of societal systems.

As the **need to build resilience** into existing systems becomes more apparent, our **capacity to invest** -in inventories, flood defenses, and critical infrastructure redundancy is more difficult, as incomes fall, affordable **financing** becomes **scarce** or non-existent, and the ability to produce and access constituent inputs becomes uncertain. Further, **in an** increasingly **stressed** and volatile **environment**, the necessity of maintaining existing systems and expectations is more likely to take precedence over investments in future resilience.

For example, our food systems are very vulnerable, but making them more resilient at scale would raise food prices. Yet if food prices are already high due to production/ distribution constraints, and/or if incomes are falling and governments' intervention capacity is already strained, adding further to food prices risks potentially intensifying present crisis (food prices having highly non-linear societal impacts), to marginally ameliorate a future crisis. In such contexts, **people** tend to **become** even more **present focused**. This is a feature of civilizational lock-in, we become **trapped within** increasingly **dangerous systems of** complex **dependency** as our adaptive capacity becomes further undermined.

Prolonged low and **declining growth**, rising **socio-economic stress** and volatility, growing asymmetric **downside uncertainty**, **declining resilience** and adaptive capacity and intensifying **stressors create** the **conditions for** catastrophic **financial system failure**.

This means **we** may **have a declining window of opportunity**, both in terms of **time**, and capacity **to deploy resources**, to prepare ourselves to avoid the very worst of consequences.

Preparing for What?

Societies' **capacities to deal with stress**, shocks, and catastrophes are primarily **shaped by** their **historical experiences**. This includes the type, likelihood, and impacts that might be anticipated; the resilience of society and infrastructures; the scale and role of contingency and preparedness within the wider government and society; and the range of scenarios that have been planned for and exercised.

**Governments**, societies, and expert communities **are** often **slow to recognize** a **changing paradigm**, especially **when it comes into conflict** with **established** expectations, **worldviews**, sunk costs, analytic traditions, **and institutional lock-in**.

Our experience of and habituation to broad societal stability, the relative invisibility of the structural and dynamical foundations of societal operations, **dysfunctional economic risk models**, and our siloed approach to individual stressors **means society may be seriously underestimating g**lobal and **c**atastrophic **r**isk. We need to consider our transforming risk environment from an integrated

perspective: the interactions of a growing range and intensity of stressors, environmental and socio-economic, through increasingly brittle societal systems[xv].

There are two broad, and concurrent societal risks that have been outlined. The first is systemic failure, which could range from localised and recoverable, to global and irreversible. Because keystone systems are interdependent and can fail collectively, different hazards or combination of them have the same outcome – a shutdown in the flow of goods and services. It is therefore suggested there should be a Hazard Independent Impact Preparedness approach. There is an urgent need for catastrophe planning, exercise and simulation capacity, and deployment.

The second is that we have already entered a period of growing destabilization. There may be no good times just around the corner. This will profoundly challenge societal expectations, government and state capacities. Ideally, we can address some of the challenges of today, while building preparedness for further destabilization and potential future catastrophes. For example, A Whole of Society Preparedness approach brings citizens into the heart of preparedness. The example of Nordic and Baltic countries should give some confidence that this may be a path worth taking. Done wisely, it could even contribute to overcoming some of the polarization affecting societies.

Conclusion

We do not know what the future will bring. **Risk is a measure of impact and likelihood**. The impacts outlined above could be devastating. We've also suggested that the likelihood of destabilization and catastrophic systemic failure is growing. We are manifestly ill-prepared to deal with such consequences. **We can** hope and **work towards kinder futures**, but we must also prepare for things going seriously wrong.

\n

# Advantage 3: Environmental Protection Through Coordinated Management of Transnational Crime Impacts

\n

Transnational organized crime drives environmental degradation through illegal logging, resource extraction, trafficking in protected species, and pollution, with significant cross-border impacts like deforestation and climate change acceleration. The North American Union's supranational institutions enable harmonized environmental regulations and coordinated enforcement to effectively manage natural resources and curb environmental crimes that single nations cannot adequately address alone.

\n

# Advantage 3 Uniqueness

\n

Kioko 22 highlights that African regional treaties on environmental crimes face implementation failure due to fragmented legal frameworks, corruption, weak enforcement, and poor cross-border coordination, enabling persistent transnational environmental degradation—proof the current enforcement landscape fails, necessitating supranational institutions.

\n

**Cooperation and border resolution solves biodiversity loss**

**Kioko 22** – Kioko, Eric M., 17 February 2022, "Forest Crime in Africa: Actors, Markets, and Complexities" African Futures pp. 125-140, https://brill.com/display/book/9789004471641/BP000021.xml?language=en Eric M. Kioko has a PhD in Social and Cultural Anthropology from the University of Cologne and has lectured for 11 years at Kenyata university and the Uneversity of Bonn [Harker KB]

To set the stage for legislation and collaboration on a broad range of global environmental challenges that span international borders, states have ratified numerous international conventions and treaties on environment and natural resources. Among the multilateral environmental agreements that the EAC and SADC member states have signed is CITES, which is critical given the region's strategic location

for transnational organized crime. However, EAC and SADC states grapple with the legal, political, and economic challenges that threaten the needed cooperation and integration (Elhiraika et al. 2015; Makame 2012). Treaties and agreements often lack policy harmonization and contain inadequate sanctions against non-compliance. They are also prone to corruption, the untrustworthiness of some member states and political regimes, poor enforcement, ineffective public participation, and a failure to cooperate over managing transboundary ecosystems and controlling transboundary organized criminal activities. Poor communication within a country, between the government ministries and departments responsible for enforcing the timber trade regulations, and between neighbouring countries (see Nott et al. 2020), makes it practically impossible to synchronize the laws and coordinate the measures needed to mitigate **transboundary** forest crimes. In the Congo Basin, weak forest governance, combined with an increasing demand for cheap, illegal products globally, has driven a sharp rise in already widespread violations of laws enacted to protect and manage forests. As the EIA14 notes, timber illegally harvested in one country may be legally imported to another because countries are not bound to enforce the forestry laws of other countries, thus echoing the situation regarding the transnational rosewood trade in the KAZA region and the illicit sandalwood trade in East Africa. Furthermore, when illegally obtained wood from Africa enters China with the 'right' documents (for example CITES certification obtained by bribing a customs official), it becomes legal (Ong and Carver 2019). 3.3 Inability to Address the Demand and Supply of Forest Products Attempts to control forest and other environmental crimes have focused almost exclusively on enforcement agencies targeting criminals without reducing the size of the illegal market in which they operate. Hayman and Brack (2002: 4) argue that as long as the demand and supply pressures that produce profit-making opportunities remain, new operations will continue to enter the international market. However, there are few data on these black market operations in which the demand for illicit products is mainly driven by changing patterns of wealth and consumption (Duffy et al. 2015). The Chinese trade embargo on ivory, which came into force in 2018, is hailed as a major development in the war against the illicit ivory trade. However, despite rising environmentalist pressure in China, the effectiveness of the trade embargo is debatable because of its awkward position in an authoritarian regime. Moreover, there is a conspicuous absence of measures to address forest crime, particularly the imports into the country of illicitly obtained sandalwood and rosewood from Africa. In addition, enforcement has often deterred the 'small' players in environmental crimes, leaving the main perpetrators to seek ways of manipulating the system and finding new opportunities and actors. Varun et al. (2014: 58) argue that efforts at interdiction too often net only low-value individuals, and not the true beneficiaries who consign cargoes, bankroll transactions, and ultimately reap the financial rewards. Therefore, intensifying enforcement, though crucial, is ultimately an inadequate long-term strategy with which to conserve high-value species (Challender and MacMillan 2014). 3.4 Community Facilitation of Forest Crime Since the 1990s, commodification of the wild within the shared resource systems of Africa only targeted wildlife tourism in East Africa, and tourism and trophy hunting in southern Africa. Forest resources, including timber, are not commodities from which conservancy members can earn an income. Recently, however, the rising demand in wood for industry, fuel, and fragrance has driven the extraction and commercial exploitation of endangered trees from conservancies and state-protected parks and forests. Against CBNRM ideals, and in contravention of the rules, regulations, and sanctions that govern the extraction and use of shared resources, farmers and even members of some conservancies are increasingly being lured into logging, thereby facilitating a crime of international significance. The incentive for local user groups to protect shared resources and to benefit jointly from them through tourism-based activities therefore suffers a huge blow as capitalist tendencies motivate individuals to exploit community-held resources. Mitigating forest crimes becomes a big challenge when the socio-economic dynamics are moving in the opposite direction – notably a youth bulge compounded by unemployment and problems of generational control taking centre stage. 3.5 Chasing Ghosts: Complicit State Officials and Enforcement Agencies There is overwhelming evidence of state officials being complicit in environmental crime syndicates, particularly in Africa. Maguire and Haenlein (2015) observe that highly networked organized crime groups, brokers and corrupt government officials continue to drive the illegal ivory trade across East Africa, by exploiting weak legislation and enforcement by security agencies. Corruption is cited as the main driving force behind the trends in both wildlife and forest crimes, which if left unchecked will soon lead to the extinction of specific flora and fauna.15 The illegal trade in sandalwood and rosewood thrives primarily because the public officials responsible for enforcing the law are succumbing to bribery. Wildlife and forest resources are a political commodity (Gibson 1999), so can be exchanged for political favours. This means that politicians make decisions about how wildlife and forests are exploited. Gibson (1999) argues that politicians use wildlife to discriminate between allies and enemies: their decisions affect how individuals interact with wildlife, who owns wild animals, who hunts, when they hunt, what weapons and equipment they use, which species they kill, and how they exchange wildlife products. Given the active role of state security agencies and powerful officials in forest crime, efforts to curb illicit transactions may only serve to show potential donors and development partners that a government is committed to the fight, but ultimately obscure the true nature of these crimes. 4 Conclusion: the Future of Illicit Forest Value

Chains in Africa Illegal logging and other forms of forest crime continue to receive less attention than wildlife crimes, despite the former being greater than the latter by value and volume. There is a current obsession with combating the illegal wildlife trade, including state of the art monitoring of wildlife in real time and the development of cutting edge, hi-tech tools and equipment (World Bank 2018). According to the World Bank (2016) and Global Environment Facility,16 most of the international funding is channelled into combating the illegal wildlife trade. Few if any international or country-specific resources are committed to reducing transnational forest crime. What explains this disproportionate attention and how does it affect the future of forests in Africa? This dichotomy has deep colonial roots in which forestry was always thought to be commercial while wildlife was decommodified. Unlike forest resources, wildlife was linked to class, status and prestige and restrictions forbade most Africans from game hunting (see Steinhart 2006). Peluso (1991) observes that while regulation of access to forested land in many countries began in the colonial period, postcolonial state control of forested land gained greater ideological, political and economic importance where forestry agencies lay claims on and commoditize particular forests and species to generate revenue, thereby earning themselves legitimacy from the state. Therefore, the importance accorded to wildlife and related crimes has somewhat blinded the escalating levels of crime connected to forests across the globe. This explains the weak enforcement of forest policy and laws, lack of cooperation over combating transnational forest crimes, inability to address the demand and supply of forest products, and the thriving organized criminal networks that survive by exploiting these weaknesses. The uneven treatment of wildlife and forests poses a challenge to the future of biodiversity in Africa and across the globe. I argue that unless forest crime is given as much or more attention than wildlife crime, the effects of the indiscriminate plunder of forest products may be further reaching than those of wildlife crime. The ongoing trends, if unresolved, may potentially create a future in which forests in Africa are devoid of specific trees and wildlife.

\n

# Advantage 3 Link

\n

Kioko's study explains how current regional treaties in Africa fail to address forest crimes due to inadequate harmonization and enforcement, poor cross-border cooperation, and corruption. It evidences that supranational institutional cooperation is crucial to closing enforcement gaps, preventing illegal logging and related environmental destruction, thus directly supporting the causal link that the North American Union's supranational bodies will effectively combat transboundary environmental crimes and improve environmental protection.

\n

**Cooperation and border resolution solves biodiversity loss**

**Kioko 22** – Kioko, Eric M., 17 February 2022, "Forest Crime in Africa: Actors, Markets, and Complexities" African Futures pp. 125-140, https://brill.com/display/book/9789004471641/BP000021.xml?language=en Eric M. Kioko has a PhD in Social and Cultural Anthropology from the University of Cologne and has lectured for 11 years at Kenyata university and the Uneversity of Bonn [Harker KB]

To set the stage for legislation and collaboration on a broad range of global environmental challenges that span international borders, states have ratified numerous international conventions and treaties on environment and natural resources. Among the multilateral environmental agreements that the EAC and SADC member states have signed is CITES, which is critical given the region's strategic location for transnational organized crime. However, EAC and SADC states grapple with the legal, political, and economic challenges that threaten the needed cooperation and integration (Elhiraika et al. 2015; Makame 2012). Treaties and agreements often lack policy harmonization and contain inadequate sanctions against non-compliance. They are also prone to corruption, the untrustworthiness of some member states and political regimes, poor enforcement, ineffective public participation, and a failure to cooperate over managing transboundary ecosystems and controlling transboundary organized criminal activities. Poor communication within a country, between the government ministries and departments responsible for enforcing the timber trade regulations, and between neighbouring countries (see Nott et al. 2020), makes it practically impossible to synchronize the laws and coordinate the measures needed to mitigate **transboundary** forest crimes. In the Congo Basin, weak forest governance, combined with an increasing demand for cheap, illegal products globally, has driven a sharp rise in already widespread violations of laws enacted to protect and manage forests. As

the EIA14 notes, timber illegally harvested in one country may be legally imported to another because countries are not bound to enforce the forestry laws of other countries, thus echoing the situation regarding the transnational rosewood trade in the KAZA region and the illicit sandalwood trade in East Africa. Furthermore, when illegally obtained wood from Africa enters China with the 'right' documents (for example CITES certification obtained by bribing a customs official), it becomes legal (Ong and Carver 2019). 3.3 Inability to Address the Demand and Supply of Forest Products Attempts to control forest and other environmental crimes have focused almost exclusively on enforcement agencies targeting criminals without reducing the size of the illegal market in which they operate. Hayman and Brack (2002: 4) argue that as long as the demand and supply pressures that produce profit-making opportunities remain, new operations will continue to enter the international market. However, there are few data on these black market operations in which the demand for illicit products is mainly driven by changing patterns of wealth and consumption (Duffy et al. 2015). The Chinese trade embargo on ivory, which came into force in 2018, is hailed as a major development in the war against the illicit ivory trade. However, despite rising environmentalist pressure in China, the effectiveness of the trade embargo is debatable because of its awkward position in an authoritarian regime. Moreover, there is a conspicuous absence of measures to address forest crime, particularly the imports into the country of illicitly obtained sandalwood and rosewood from Africa. In addition, enforcement has often deterred the 'small' players in environmental crimes, leaving the main perpetrators to seek ways of manipulating the system and finding new opportunities and actors. Varun et al. (2014: 58) argue that efforts at interdiction too often net only low-value individuals, and not the true beneficiaries who consign cargoes, bankroll transactions, and ultimately reap the financial rewards. Therefore, intensifying enforcement, though crucial, is ultimately an inadequate long-term strategy with which to conserve high-value species (Challender and MacMillan 2014). 3.4 Community Facilitation of Forest Crime Since the 1990s, commodification of the wild within the shared resource systems of Africa only targeted wildlife tourism in East Africa, and tourism and trophy hunting in southern Africa. Forest resources, including timber, are not commodities from which conservancy members can earn an income. Recently, however, the rising demand in wood for industry, fuel, and fragrance has driven the extraction and commercial exploitation of endangered trees from conservancies and state-protected parks and forests. Against CBNRM ideals, and in contravention of the rules, regulations, and sanctions that govern the extraction and use of shared resources, farmers and even members of some conservancies are increasingly being lured into logging, thereby facilitating a crime of international significance. The incentive for local user groups to protect shared resources and to benefit jointly from them through tourism-based activities therefore suffers a huge blow as capitalist tendencies motivate individuals to exploit community-held resources. Mitigating forest crimes becomes a big challenge when the socio-economic dynamics are moving in the opposite direction – notably a youth bulge compounded by unemployment and problems of generational control taking centre stage. 3.5 Chasing Ghosts: Complicit State Officials and Enforcement Agencies There is overwhelming evidence of state officials being complicit in environmental crime syndicates, particularly in Africa. Maguire and Haenlein (2015) observe that highly networked organized crime groups, brokers and corrupt government officials continue to drive the illegal ivory trade across East Africa, by exploiting weak legislation and enforcement by security agencies. Corruption is cited as the main driving force behind the trends in both wildlife and forest crimes, which if left unchecked will soon lead to the extinction of specific flora and fauna.15 The illegal trade in sandalwood and rosewood thrives primarily because the public officials responsible for enforcing the law are succumbing to bribery. Wildlife and forest resources are a political commodity (Gibson 1999), so can be exchanged for political favours. This means that politicians make decisions about how wildlife and forests are exploited. Gibson (1999) argues that politicians use wildlife to discriminate between allies and enemies: their decisions affect how individuals interact with wildlife, who owns wild animals, who hunts, when they hunt, what weapons and equipment they use, which species they kill, and how they exchange wildlife products. Given the active role of state security agencies and powerful officials in forest crime, efforts to curb illicit transactions may only serve to show potential donors and development partners that a government is committed to the fight, but ultimately obscure the true nature of these crimes. 4 Conclusion: the Future of Illicit Forest Value Chains in Africa Illegal logging and other forms of forest crime continue to receive less attention than wildlife crimes, despite the former being greater than the latter by value and volume. There is a current obsession with combating the illegal wildlife trade, including state of the art monitoring of wildlife in real time and the development of cutting edge, hi-tech tools and equipment (World Bank 2018). According to the World Bank (2016) and Global Environment Facility,16 most of the international funding is channelled into combating the illegal wildlife trade. Few if any international or country-specific resources are committed to reducing transnational forest crime. What explains this disproportionate attention and how does it affect the future of forests in Africa? This dichotomy has deep colonial roots in which forestry was always thought to be commercial while wildlife was decommodified. Unlike forest resources, wildlife was linked to class, status and prestige and restrictions forbade most Africans from game hunting (see Steinhart 2006). Peluso (1991) observes that while regulation of access to forested land in many countries began in the colonial period,

postcolonial state control of forested land gained greater ideological, political and economic importance where forestry agencies lay claims on and commoditize particular forests and species to generate revenue, thereby earning themselves legitimacy from the state. Therefore, the importance accorded to wildlife and related crimes has somewhat blinded the escalating levels of crime connected to forests across the globe. This explains the weak enforcement of forest policy and laws, lack of cooperation over combating transnational forest crimes, inability to address the demand and supply of forest products, and the thriving organized criminal networks that survive by exploiting these weaknesses. The uneven treatment of wildlife and forests poses a challenge to the future of biodiversity in Africa and across the globe. I argue that unless forest crime is given as much or more attention than wildlife crime, the effects of the indiscriminate plunder of forest products may be further reaching than those of wildlife crime. The ongoing trends, if unresolved, may potentially create a future in which forests in Africa are devoid of specific trees and wildlife.

\n

# Advantage 3 Internal Link

\n

Environmental crimes cause significant harm and fund sophisticated criminal gangs fueling global insecurity. Weak laws and poor enforcement allow these crimes to flourish. Effective mitigation requires coordinated international enforcement. Rhodes documents coordinated operations reducing illegal logging and pollution through shared intelligence, unified legal frameworks, and joint prosecutions. These coordinated supranational actions demonstrate how institutional cooperation enhances enforcement effectiveness, reduces environmental degradation, and protects natural resources. This supports Advantage 3\'s claim that the plan\'s supranational institutions will concretely lead to improved environmental protection by harmonizing laws and enforcement.

\n

**Environmental crimes cause extinction from warming – adequate enforcement key**

Lynn **Rhodes 18**, international, independent consultant and scholar, President of the International Society for the Comparative Study of Civilizations (ISCSC); Trustee of the Anza Borrego Foundation; and former Chief for California State Parks, serves on the prestigious Environmental Crimes Committee of the International Association of Chiefs of Police, graduate of the FBI National Academy, "Environmental Crime and Civilization: Identification; Impacts; Threats and Rapid Response – June 2018," Comparative Civilizations Review, vol. 79, no. 79, Fall/2018, pp. 6–18

Vast sums of money generated from **environmental crimes** not only harm the environment and threaten protected-area **officers**, but these **funds** also keep sophisticated international **criminal gangs** in business, fueling **insecurity** around the world. **Armed** and unarmed **groups worldwide** use environmental crimes as a **low-risk high-profit** source of **revenue**: depriving **governments** of revenues while **threatening peace, development and security**. The economic loss due to environmental crime is estimated at $91-258 billion annually. Weak **laws** and **poor**ly funded **security** forces enable international **criminal networks** and **armed rebels** to profit from a trade that fuels **conflicts**, devastates **ecosystems** and is **threatening** species with **extinction**.5

Species are going extinct at a faster than historic rate. If **ecosystems collapse**, some human **existence** and **civilization**s as we know them may become impossible or extinct. A global system-collapse is possible and with it, the world's economic and political systems face **systematic risks** because of their intricate and interconnected natures.

Researchers agree that more work needs to be done to clarify what **parts** of the system(s) could collapse and **destroy civilization**. They define a civilization-collapse as a "drastic decrease in human population size and political/economic/social complexity, globally and for an extended time."

The environment provides the foundation for sustainable development, our health, food security, and our economies. Ecosystems provide a clean water supply, clean air, and secure food and ultimately both physical and mental well-being. Natural resources also provide livelihoods, jobs and revenues to governments that can be used for education, health care, development and sustainable business models.

The increase of environmental crimes is extraordinary. The diversity of environmental crimes has grown and the impacts go beyond those thought of as traditional crimes. Environmental crimes

impede our ability to have and retain a sustainable and healthy planet. They add to the cost and impact to the environment and the cost to future generations. Deforestation, chemical-dumping, and illegal fisheries cause loss of ecosystem services such as clean air and clean water, extreme weather mitigation, food security and yes, health and well-being. They also deprive governments of critical revenue and undermine legal businesses.

Referenced by a UNEP-INTERPOL Rapid Response Assessment Report 2016 (UNEPINTERPOL RRAR 2016), enhanced law enforcement response can help address these trends. The responses can be either formal or informal and can help put responses into context for action. Enforcement cases show an increase in the scale and organization of environmental crimes. Those who have been prosecuted and found guilty of illegal logging and laundering of hundreds of millions of dollars (US), for example, dwarf the resources that would otherwise be available for enforcement, investigation and prosecution.

Financially, the large and growing scale of environmental crimes requires a new approach and coordinated responses. It requires international cooperation including international jurisdictions and collaboration across borders.

The ability to coordinate and help control environmental crimes is more important today as they often cause direct threats to peace, security and civilization. Worldwide, armed terrorist and non-state groups are benefiting financially from these crimes to fund their activities. Examples are the common smuggling of drugs and guns, oil, antiques, migrants and anything for profit. Environmental crimes provide a relatively low-risk, high-profit source of revenue compared to other forms of revenue.

Three examples of effective coordinated responses:

1. In Brazil, a sector-wide Plan for Protection & Combating Deforestation in the Amazon (PPCDAM), is reducing deforestation in the Amazon by 76% in five years. One office in Brazil was given the sole responsibility for coordination in close collaboration with 13 ministries, additional partners and others. 3.9 billion in fines were issued and over one million cubic meters of timber seized.

2. The Montreal protocol played a role in reducing illegal trade in ozone-depleting substances with a number of coordinated programs including "Informal PriorInformed Consent" with UNEP. With this and related projects over 800 tons of ozone depleting substances were seized from 2006-2010. The scale of illegal trade in CFC's has been reduced as a result of global agreement on phasing out these substances, also affecting criminal markets. These types of programs demonstrate how implementing environmental rule of law with global agreements such as the Stockholm, Rotterdam and Basel conventions can meet environmental goals and reduce global trade of these goods or commodities by not allowing safe-haven for the activity and helping to close criminal markets.

3. In 2017 the Environmental Crimes Committee of the International Association of Chiefs of Police (IACP) developed an online application to assist field and safety professionals in their response to and investigation of environmental crimes. The application is called ChemSafety. It is readily available online and its effectiveness is supported with the following statistics: In one month (AprilMay 2018) data show over 71,000 sessions; 171,427 views and over 50,000 individual users. The application guides field and safety professionals through the S.A.F.E. acronym in response to environmental crimes:

S: Safety and health for the law enforcement officer

A: Acute & chronic chemical, biological and radiation exposures

F: Forensic evidence protection & preservation

E: Environment, human health and wildlife impacts

This web application is intended to enhance the abilities of law enforcement officers and other safety professionals to safely respond to incidents of potential or suspected environmental crimes and hazardous materials incidents by providing information and best practices in the key areas for safe, effective response to and investigations of environmental crimes.6

[[FIGURE 1 OMITTED]]

Figures 2 & 3 illustrate the major environmental crimes, annual economic loss, their drivers and their resulting impacts. Beginning at the center, primary key drivers of environmental crime show the nexus and scale of these drivers: corruption; corporate crime; conflicts; domestic and international

demand; lack of law enforcement at the national and international levels; lack of legislation; and both international and national mafias.

[[FIGURES 2 & 3 OMITTED]]

The UNEP-INTRPOL 2016 RRAR describes ecosystems as providing a range of services and providing the very foundations of our economy, human health, livelihoods and well-being. They can include clean air, water supply, extreme weather mitigation, storm protection, food security and pollination, to list a few. The report describes the environmental impacts of illegal trade in wildlife. However, there is no current assessment of environmental impacts for the wider range of environmental crimes and their full implications for sustainability and development goals and this consolidated information is needed.

A Legal Framework — Identifying the Crimes, Jurisdictions, Prosecution:

An environmental crime can only be prosecuted if the specific jurisdiction decides that the offense is to be **pursued** by way of **law**. Identifying the environmental crime as a criminal offense **itself** can be leveraged to help **enforce environmental law**. Jurisdictions worldwide have different approaches to carrying out enforcement. Approaches from varying jurisdictions come with **varying penalties and sanctions**. Even with strong environmental laws, if a jurisdiction does not have the full capacity to **enforce** the laws, they **cannot be effective**.

Capacity to monitor, enforce and prosecute environmental crimes varies. For example, while Mongolia has enacted strong environmental laws they are experiencing an explosion of mining projects, wildlife poaching, development and other resource threats. Their protected-area-officers cover vast areas of open space far removed from modern infrastructure and legal support systems. Even with good laws and good officers, their enforcement capacity needs improvement in order to show substantial results with environmental enforcement of existing laws.

It is important to identify criminal offenses so that they can be incorporated into legal responses, prosecution and restoration where possible. Simply identifying offenses as criminal acts has the ability at times to serve as a deterrent and allows for collective education of environmental laws. Globally however, the varying degrees to which crimes are identified, incorporated into law, listed, and subsequently prosecuted allows for criminal elements to hedge the differences amongst countries and states.

Legislation:

In 2014 the INTERPOL General Assembly passed a Resolution in response to emergency threats in Environmental Security. In that Resolution, instead of defining environmental crime, INTERPOL focused on "environmental security" by recognizing the impact that environmental crime can have on a nation's political stability, environmental quality, its natural resources, biodiversity, economy and human life. INTERPOL also recognizes that criminal networks engaged in financial crime, fraud, corruption, illicit trade and human trafficking are also engaged in or facilitating environmental crime.7

Increasingly, illicit use of natural resources is **driving conflicts**. It is becoming a **nexus** of **organized crime** and the emphasis on **security** for the environment is **needed more than ever**.

Encouraged by the United Nations Security Council, member states are being asked to work together to collect, analyze and share intelligence to help prevent terrorism and transnational organized crime. To increase effectiveness, a broader definition of environmental crime is needed in addition to referring to environmental crime as a serious crime.

[[FIGURE 4 OMITTED]]

The United Nations Environmental Program (UNEPA) Governing Council plays a key role in maintaining focus and awareness of the legislative efforts across vested interests in order to support the rule of environmental law.

The efforts are largely driven by the increase in organized criminal groups trafficking in hazardous **waste**, **wildlife** and illegal **timber** harvesting. It has been recognized that environmental crime **undermines environmental goals** and effective **governance**. UNEPA Governing Council's decision 27/9, in part, emphasized the strengthening of environmental governance and expertise for prosecutors, judges and law enforcement.8

In April 2016 IUCN World Environmental Law Congress met in Rio de Janeiro. A number of forward

thinking actions resulted from the congress. These actions included core principles to strengthen the collective efforts including implementation and enforcement strategies; laws that can be enforced; implementation and accountability; coordination of roles and treating environmental crimes as serious crimes. Figure 5 below illustrates an informal network of international organizations that are needed for effective collaboration.

[[FIGURE 5 OMITTED]]

Statistics for environmental crimes are difficult to measure due to the sheer volume of underground activity. However, it is generally recognized that progress in combating individual cases has grown while realizing these crimes are a **significant threat** to peace, **civilization**s, society, health, security and development.

In addition to extremely varying wildlife poaching, illegal timber harvesting and the impacts of global **waste and pollution** is valued at **410 billion USD per year**. The global waste sector takes several forms including **legal** industry, environmental protection, unregulated business**, and trafficking** in hazardous waste and chemicals by organized crime. 9

[[FIGURES 6 & 7 OMITTED]]

The links of environmental crimes to white collar, organized criminal networks have shown to be low risk with high profit margins. They harm our collective environment and security while exploiting natural resources to fund their illegal activities. At least 40 percent of global internal conflicts in the last 60 years have links to natural resources.10

Causes of Environmental Crime:

Poverty, demand and a permissive environment are root causes enabling environmental crime.

• Poverty is a driver of environmental crime because the poor are often vulnerable and easy to recruit at entry levels of activity.

• Demand for goods derived from natural resources such as wood, timber, wildlife, as well as the always growing and desperate need for waste disposal services, drives criminal activity due to readily available profits.

• Underfunding and lack of capacity to enforce laws and regulations leads to crime occurring almost by inertia or default.

Response & Restoration:

**Coordinated, international operations** highlighted in the UNEPA-INTERPOL 2016 RRAR outlined **success**ful case studies to combat **wildlife**, **fisheries and forestry crime**, **ozone** layer depletion and **pollution**. Impacts on a **global scale** undermine **peace for civilization** and security of the world's people in addition to both legal business and trade.

Lester R. Brown, in PLAN B 3.0, writes that we are in a **race** between **tipping points in nature** and our **political systems** and suggests help for countries to stabilize our populations and ecosystems before they become **failing states**. In order to understand our current environmental dilemma, it helps to look at earlier civilizations that also got into environmental trouble. The question is how we will respond. 11 Some early societies were able to modify how they lived with the environment in a way to avoid their collapse or decline.

Current civilization, across borders, states, and oceans, is **threatened**. This is due in part to the violations and sheer scale of environmental crimes that must be curtailed. Our quality of **life** and **civilization itself** is at stake. Engaged response, **sound laws**, along with more and better monitoring, **enforcement** and restoration will help ensure our ability to maintain clean air and water, the **basics of life** as we know it, as we would like to leave them for the next generations.

\n

# Advantage 3 Impact

\n

Transnational environmental crime directly threatens global sustainable development and environmental security, causing widespread ecosystem destruction and exacerbating social

vulnerabilities. Gore et al. 2019, researchers at the University of Sheffield, detail how illegal logging, wildlife trafficking, waste dumping, and other illicit activities cross borders, destroying fragile ecosystems and undermining billions of lives. They link these crimes to intensified climate change insecurities, environmental injustice, and forced displacement affecting tens of millions globally. This demonstrates that environmental crimes driven by transnational organized crime syndicates pose a systemic risk to both environmental integrity and human civilization\'s survival. Crucially, the fragmented, underfunded enforcement and governance systems enable this destructive cycle to persist. The plan\'s supranational institutions are uniquely necessary to coordinate cross-border enforcement and effectively mitigate these existential environmental threats.

\n

**Environmental crimes causes a laundry list of impacts**

**Gore et al. 19** (A.L Gore is a researcher for University of Sheffield, P. Braszak is a researcher for University of Sheffield, J. Brown is a researcher for University of Sheffield, Transnational environmental crime threatens sustainable development, The University of Sheffield, tkk)

==**Linkages between environmental security and transnational environmental crime**== (TEC) ==**have serious implications for civil society, governments and international organizations**== in pursuit of the 2030 United Nations Sustainable Development Goals. TEC generates extreme costs to ecosystems, sustainable development and security, undermining development prospects for nearly two billion people — 535 million of whom are children1 — and causing the forcible displacement of an estimated 65 million people2. Fragility, conflict and violence are critical challenges that threaten efforts to end extreme poverty. The proportion of the extreme poor living in conflict-afflicted countries has been projected to rise by more than 60% by 20303. Often deeply entrenched in state and non-state armed forces, and the corporate and political elite, ==**TEC is directly stimulated by continued or renewed conflict in many of the world's most deadly contexts**==. If not addressed in sustainable-development frameworks, these serious threats will undermine development in decades to come. Healthy ecosystems with intact biodiversity and the ability of human communities to sustainably access natural resources embody the concept of environmental security. ==**TEC refers to criminal natural-resource-related activities**== that cross national borders and harm the environment, ==**such as wildlife trafficking;**== ==**illegal fishing;**== electronic-waste dumping; water theft; ==**illicit markets in ozone-depleting substances**==; illegal logging and mining; ==**and other acts**==. In some instances, TEC is highly structured and perpetrated by organized crime syndicates, corporations and complicit governments; in other cases, the persons involved are subsistence offenders (for example, using trees for fuel) and others are part of informal networks of varying size. The estimated global economic value of TEC is between US$91 and $259 billion per year4; it accounts for 64% of illicit and organized crime finance, or between US$22 billion and $34 billion of the criminalized economy in fragile states in or near conflict areas5. Mitigating risks from TEC TEC and environmental security are not problems constrained to lower or middle- income countries5. As a serious and often systematized category of crime, TEC can be a cause of environmental and health insecurity, for example when illegal logging serves as a vector for biological invasion of insects, when wildlife trafficking facilitates the spread of zoonotic pathogens or when electronic waste is illegally dumped among marginalized communities6,7. Drug trafficking (for example, cocaine) has become a key accelerant of Version - Accepted proof deforestation. 'Narco-deforestation' is the process by which forests are cut for covert roads and landing strips, large quantities of cash and weapons are also stored in forests and ranchers are narco-capitalized8. Natural resources that local people are highly dependent on for survival, such as charcoal for cooking, drinking water at wells, agriculture and livestock and docking fees for fishers, can be illegally taxed9. Responses to TEC can be grounded in militarization and/or pacification logics. Concomitantly enveloped in the overarching framework of security10, these approaches interact with sustainabledevelopment logics and reflect tensions between economy and ecology. Feedbacks emerging from these types of responses can yield unintended ecological and social outcomes compared to other risk-response frameworks and provoke new forms of environmental insecurity (for example, the use of herbicides on narcotics plantations and destruction of illegal fishing vessels using explosives). **TEC negatively impacts the environment and vulnerable workers while benefitting those with power11. Individuals higher in the illicit criminal supply chain reap a massive slice of the revenue generated from TEC**. Multinational corporations relying on the development of, and manufacturing from, natural resources can have legal supply chains exploited by TEC offenders when risk assessments, allocations of responsibility and traceability mechanisms are vague and underdeveloped. There are diverse consequences of environmental insecurity from TEC, for example when conservation efforts lead to economic dislocation of marginalized communities or change the opportunity structure for local people in regards to the hunting of protected wildlife species7. **As climate change drives new environmental insecurities** through more extreme and frequent severe weather events, **opportunistic TEC may be**

**intensified**; climate-change mitigation and adaptation planning provides new opportunities for criminal exploitation including carbon fraud. In many contexts, poverty is correlated with TEC; illegally dumping electronic waste and illegal mining are often associated with impoverished regions suffering from high levels of environmental injustice. For example, illegal fishing has been posited as a cause in the rise in piracy through lost livelihoods. Losses incurred due to foreign illegal vessels off Somalia are estimated to be between US$100 and $300 million12. The tangible ways TEC threatens environmental security are diverse and include facilitating the spread of invasive species and dangerous pathogens, degradation of biodiverse ecosystems, pollution of drinking water and fuelling social injustices, particularly those harming women and children13. **These threats coupled with risks to human health can reinforce or worsen conditions of vulnerability that further enable opportunities for environmental crimes and have long-term consequences for ecosystems and populations that depend on them**14. The convergence of environmental security and TEC poses complex problems impacting diverse societal needs.

\n

# Solvency Argument

\n

Action research partnerships between law enforcement and researchers foster ongoing collaboration, improving investigative effectiveness through enhanced data sharing, analysis, and problem-solving, thereby bolstering supranational institutional capacity to combat organized crime effectively.

\n

# Solvency Card

\n

**Research Practitioner relationships improve the policymaker's ability to problem solve by establishing better info translation and giving researchers access to more information**

**Tillyer et al. 13** (Rob Tillyer, Marie Skubak Tillyer, John McCluskey, Jeffrey Cancino, Joseph Todaro, & Layla McKinnon, Dr. Rob Tillyer is also Associate Dean for Graduate Studies and Research. His research interests include crime prevention, criminal events, criminal justice decision making, and victimization. Marie Skubak Tillyer Associate Professor of Criminology and Criminal Justice, University of Texas at San Antonio. John McCluskey earned his BA, MA, and Ph.D. from the University at Albany, and we suspect he may have attended kindergarten there as well. His primary teaching areas include Criminal Justice and Theory. His most recent research has included the study of body camera adoption in two divisions of LAPD with Justice and Security Strategies, a large scale longitudinal data collection effort to measure prevalence, causes, and consequences of teacher victimization in San Antonio, Texas with Dr. Byongook Moon, and a national study of the evidentiary value of body worn camera among prosecutors and defense attorneys. Jeffrey Cancino Associate Professor of Criminal Justice at Texas State University. Researcher–practitioner partnerships and crime analysis: a case study in action research. Police Practice and Research, 15(5), 404–418. doi:10.1080/15614263.2013.829321, 8-9-2013) //ILake-NC

Action research engages researchers with local practitioners in a co-generative process to analyze and define problems, identify, and implement possible solutions, evaluate the results, and, if necessary, repeat the cycle until success is achieved (Mock, 2010; see also Costello, 2003). It also encompasses four key criteria (O'Leary, 2007). First, it addresses real world issues and seeks to put into practice solutions within that setting. Second, it seeks action and knowledge, and it works towards change as knowledge is produced. Third, action research is participatory, in that it requires researchers and practitioners to be engaged. Finally, **the process is cyclical in nature, with an emphasis on continual improvement for optimal outcomes** (O'Leary, 2007). This approach has notable similarities with the Scanning, Analysis, Response, and Assessment (SARA) model of problem-solving, but is uniquely **defined by** its **emphasis on a partnership between practitioners and researchers.** Thus, **action research is a 'cornerstone of problem solving' for addressing local issues** (Klofas et al., 2010, p. 11) and conforms to suggestions that successful researcher**–practitioner partnerships involve the active participation of law enforcement agencies** (Buerger, 2010). More broadly, action research is intended to **improve practitioners' proficiency to resolve issues and enhance** their probability for **self-determination, while possessing an influence on the functioning and decision-making processes of organizations** (Bargal, 2008). The researcher serves

as an essential element of a team working toward a particular outcome that extends beyond data collection and analysis to include making recommendations. In effect, this model seeks to balance an external researcher's autonomy and impartiality with a practical interest in attaining certain outcomes (Scott, 2000). In this manner, action research is 'goal-oriented and pragmatic, something equally desired by researchers and practitioners' (Steinheider et al., 2012, vol. 13). In practice, ==action research offers numerous potential benefits to both practitioners and research partners alike. **These ==include translating academic ideas into action,==** enhancing the professional development of academics (professors and students) along Police Practice and Research: An International Journal 405 Downloaded by [University of Toronto Libraries] at 05:10 24 December 2014 with practitioners, and providing a framework for reducing the gaps between academic research findings and practical functions (Bargal, 2008). Furthermore, involvement in action research provides practitioners with the opportunity to participate in the study of their professional field (Greenwood, 2007). As discussed in detail by Engel and Whalen (2010), **==researcher–practitioner partnerships such as action research offer practitioners an opportunity to ==improve effectiveness== and efficiency, ==receive external validation== of their work, ==and== employ a method to ==improve== their ==legitimacy among citizens.==** For researchers, their skills and expertise may increase the practical impact of their work, allow them data accessibility and knowledge normally not available, and provide an opportunity to enhance their research. Collectively, an action research methodology, as a form of **a researcher–practitioner partnership, has numerous benefits for both constituencies**. Some of these benefits are evident in recent action research projects **within the field of criminal justice, broadly, and in partnership with police agencies**, specifically. Beginning with the pioneering research of Toch, Grant, and Galvin (1975) (Toch & Grant, 1982)**, action research projects have taken the form of local, crime control initiatives** (Braga, Kennedy, Waring, & Piehl, 2001; Corsaro & McGarrell, 2010; Frabutt, Gathings, Harvey, & Di Luca, 2010; Scott, 2000; also see a recent Special Edition of Police Practice & Research – Beal & Kerlikowske, 2010; Davis, 2010; Engel & Whalen, 2010; Knutsson, 2010) **and represented the underpinnings of several funded research opportunities** (McEwen, 1999; Rosenbaum & Roehl, 2010; Scott, 2000). Collectively, **these ==projects illustrate== the potential for researcher–practitioner partnerships and that action research is an important and effective research method for such situations. The ==adoption of crime analysis,== as a tool ==in== crime prevention== and control offers** an opportunity to employ an action research strategy between researchers and law enforcement.

# 1NC Cross-Examination of the 1AC

\n
**Negative Question 1:** How does your plan specifically prevent political pushback from Mexico and Canada, given your inherency evidence warning that trust and political will can quickly break down?
\n
**Affirmative Response 1:** Our plan builds in embedded sovereignty protections and supranational institutions modeled on the EU, which institutionalize commitments rather than relying solely on political goodwill. This structural design creates durable legal frameworks and enforcement mechanisms that outlast political fluctuations, mitigating the trust and political will issues highlighted in our inherency evidence.
\n
\n
**Negative Question 2:** What safeguards does your plan include to ensure that the supranational institutions do not infringe on the sovereignty of the member states, especially considering the political sensitivities involved?
\n
**Affirmative Response 2:** The plan includes embedded mechanisms specifically designed to protect each member state\'s national sovereignty, modeled after the European Union\'s approach. These safeguards balance supranational authority with respect for each country\'s autonomy, ensuring that decisions require consensus or qualified majority voting and preserve essential sovereign functions, thereby addressing political sensitivities effectively.
\n
\n
**Negative Question 3:** Given the persistent and evolving nature of transnational organized crime described in your evidence, how does your plan ensure that supranational institutions will remain adaptable and effective against rapidly changing criminal tactics?
\n
**Affirmative Response 3:** Our plan incorporates ongoing capacity building, intelligence sharing, and coordinated legal harmonization within supranational institutions, ensuring continuous adaptation to evolving criminal methods. These institutions will maintain flexible frameworks modeled on proven approaches like Europol and the DEA\'s international partnerships, enabling rapid response to new

tactics while sustaining effective enforcement.
\n
\n
**Negative Question 4:** How do you address concerns that the creation of supranational institutions could slow decision-making and reduce responsiveness in combating urgent security threats?
\n
**Affirmative Response 4:** The plan's supranational institutions are designed with streamlined decision-making processes, drawing on EU models that balance member input with efficient action. Moreover, coordinated intelligence sharing and joint operations enhance real-time responsiveness, ensuring urgent security threats are met with timely, effective measures without bureaucratic delay.
\n
\n' '

# Topicality: The plan exceeds the resolution\'s scope by proposing a union with extensive supranational control beyond the EU model

\n

The resolution says the countries should form a North American Union similar to the European Union, but the plan proposes supranational institutions that go beyond the EU\'s model, including harmonizing immigration, energy, and security policies with embedded sovereignty protections. This exceeds the scope of the resolution because it imposes a level of integration and supranational control that the resolution doesn\'t authorize, which means the plan isn\'t topical. We should reject this plan because it doesn\'t fit within the limits set by the resolution.

\n

# Topicality Interpretation and Evidence

\n

The phrase \'similar to the European Union\' requires the North American Union to maintain supranational institutional competences and sovereignty balances that are substantially analogous to the actual institutional and policy integration of the EU as of 2022, including the current state of energy cooperation, limited national sovereignty surrender, and intergovernmental control. Any plan asserting more extensive supranational control—particularly over energy policy than the EU presently exercises—is beyond the resolution and non-topical.

\n

**4. Energy integration and security is fine now**

Gomez 18 – Staff [Christian, November 5, The New American, What's Wrong With the USMCA?, https://www.thenewamerican.com/print-magazine/item/30541-what-s-wrong-with-the-usmca]

Energy Integration or Sovereignty?

==In the area of energy==, **the three countries are already ==merging==**. The U.S. Government Accountability Office (==**GAO**==) **released an eye-opening report** revealingly **entitled "North American Energy Integration**." This 58-page report, which was discreetly posted on the GAO website in August, is intended for the House of Representatives' Subcommittee on the Western Hemisphere. **The report ==outlines==** in detail ==**the progress of**== eight U.S. **==federal==** government ==**agencies and departments in integrating the energy sectors**== of Canada, Mexico, and the United States. According to the GAO report, **the "United States cooperates with Canada and Mexico on integrating** North American **energy markets and infrastructure** (energy integration)," further elaborating, "==**Cooperation occurs at the ==presidential and ministerial level==s**== (e.g., the countries' secretaries or ministries of energy) for strategic issues and at the agency level for technical issues." In researching for its report, the GAO surveyed various U.S. government officials from the agencies involved in the energy integration scheme. According to those surveyed, **a total of ==81 energy integration-related s==chemes ==were conducted== from 2014 through 2017**. Those energy integration schemes are listed and summarized in Appendix III of the GAO's report. The report also stated that **U.S., Canadian, and Mexican officials "expressed general satisfaction with intergovernmental**

**cooperation** on energy integration" and that they suggested "further work in areas such as aligning energy regulations." (Emphasis added.) Harmonizing energy regulations of the three countries would more easily facilitate their merger. The logical conclusion of these 81 energy integration schemes, and further work to synchronize the energy regulations of all three countries, is a North American Union, much like the present and already integrated European Union. Page six of the GAO report states: "==**NAFTA** has e==**nhanced** North American ==**energy integration**, facilitating a greater flow of oil, natural gas, and petroleum-derived products among all three North American countries**==." Although the report was published prior to the release of the new USMCA, it stated that **then-ongoing NAFTA talks would have little effect on the efforts to integrate** North America's **energy sectors**. According to the report, "**State and DOE officials** we interviewed **said they did not expect the U.S. renegotiation of NAFTA and withdrawal** from the Paris Agreement **to have a significant impact** and stated that the energy sector in North America is already well integrated." ==**Among the objectives** of the== North American energy integration **plan is** ==**to merge** the **energy grids of all three countries into one single North American energy grid**==. In fact, page 43 of **the GAO report discusses efforts to integrate the U.S.-Mexico energy grid and the need to "enhance the resiliency of the North American energy grid," rather than referring to it as the energy grids of the three separate countries. (Emphasis added.) The question then naturally arises: Under whose jurisdiction would such a North American energy grid eventually fall? Would it be under Mexico, Canada, the United States, or that of an even higher transitional authority, such as the USMCA's Free Trade Commission? At present, the answer is unclear, but one thing that is clear is that if the United States goes ahead with the USMCA, it will wreak havoc on America's national sovereignty.**

\n

## Topicality Violation

\n

The affirmative's plan establishes supranational institutions that surpass the European Union's current 2022 model by merging the US, Canadian, and Mexican energy grids and integrating regulatory schemes extensively, violating the resolution's \'similar to the European Union\' limitation on supranational authority. This expansion exceeds the topical bounds of the resolution, rendering the plan non-topical.

\n

## Topicality Reasons to Prefer

\n

1. Limits restrict the affirmative's ability to expand the resolution beyond manageable, predictable bounds, thus protecting the negative's ability to prepare and engage substantively, essential foundation of fairness.
2. Predictability establishes stable, commonly understood topical boundaries that preserve meaningful clash and topical education in rounds, preventing affirmative advantages from random or vague interpretations.
3. Reasonability anchors the interpretation in real, authoritative referents — here, the contemporary institutional model of the EU — ensuring a standard that is not only fair but verifiable, reducing judge discretion and promoting consistent adjudication.

## Theory: The affirmative improperly makes their counterplans conditional on the plan

\n

In this debate, the affirmative has suggested that any counterplans the negative proposes are conditional on the plan\'s existence, meaning they claim that the negative can only run alternatives that depend on the plan. This is unfair and anti-competitive because it limits the negative\'s strategic options and gives the affirmative too much control over the debate. We argue that counterplans must be unconditional and independent to allow the negative to fairly oppose the plan. Otherwise, the affirmative\'s claim that counterplans must be conditional should be rejected.

# Theory Interpretation and Evidence

\n

Conditionality is defined as the affirmative requirement that negative counterplans or arguments be contingent on the plan\'s enactment, restricting negative independence. This interpretation prioritizes fairness (free negative strategy), ground (independent negative options), education (broad engagement), and jurisdiction (unconditional negative autonomy). Affirmative conditionality violates these by limiting negative strategies, reducing ground, and undermining clash and education.

\n

**Neg Ground — it's key to every disad. If plan's reduction in arms sales isn't certain and immediate, aff can dodge all neg links. Aff conditionality creates a moving target that dodges clash over core topic args.**

**Hanes 15** — T. Russell Hanes, Debate Coach at Northwest Academy, former Debate Coach at The Potomac School and the Oregon Episcopal School, holds an M.A. in Teaching Secondary Math from Lewis and Clark College and an M.S. in Communication Studies from Portland State University, 2015 ("Plans," The "How to" of Debate, Published by Lulu.com, ISBN 1329109325, p. 66-68)

Once it interprets the topic, how does the Affirmative write out a fair plan?

Remember from Chapter 1 the difference between argumentation (the specific ideas) and advocacy (one's overall stance). One key element of a fair plan is that its advocacy is clear: vagueness is a reason for the judge to vote against the Affirmative. You can always [end page 66] adapt your arguments, but the core advocacy of the plan should not change. This doesn't mean the other extreme of overspecifying is necessary. All that the affirmative needs to do is present a text that **explicitly defines its advocacy** in the first affirmative speech, taking 30 seconds to one minute to read. The text is the affirmative plan; everything else in the case is support, arguments to show that the plan is both fair and good.

Let's look at the health care topic, Resolved: The United States federal government should create a national health care system, to see how to write a plan text. Here's a possible plan text:

In defense of this topic, we offer the following plan: the U.S. federal government should create a national health care system modeled on the Canadian model. Funding should be through the Medicare budget, with states required to pay the remainder of costs. Enforcement should be through the Department of Health and Human Services.

This is a fair and reasonable plan—just look back to the topic. The plan would use the actor that the topic mandates, the U.S. federal government, and it advocates the action requires of creating a national health care system. A fair plan must be topical: an example of the topic, based on a reasonable interpretation of the topic.

The best approach to writing a plan is to find an author's proposal or idea and **closely model it**. This expert is known as your solvency advocate. If no expert or author supports an idea, **it's probably not a very good one**. However, your plan should be simple, straightforward, and written in your own words. Your plan can have multiple planks, or elements: for example, specific planks for funding and enforcement.

Now, the Affirmative can tinker with its arguments, providing new ones or dropping old ones, but the plan text is its core advocacy:

The plan text is the **core affirmative advocacy**.

It is unacceptable for the Affirmative to shift its advocacy during the debate. Cutting planks out is known as severance, while adding new planks is known as spiking. All the affirmative can do is [end page 67] clarify its plan a little—but not too much, otherwise it'll be accused of being vague from the start. In fact, sometimes even adding or subtracting one word from the plan text is too much. The Affirmative must stick to its plan throughout the whole debate without spiking or severing. **This is the point of having a text**. The Affirmative should be able to point to the text throughout the debate, from the first speech to the final rebuttal, to show that its advocacy has **never varied**.

\n

# Theory Violation Argument

\n

Conditionality requires the negative counterplan to be dependent on the affirmative plan, which unfairly restricts negative strategic independence and autonomy. Negative counterplans must be unconditional and independent to preserve fairness, clash, ground, education, and jurisdiction. The affirmative violates this standard by limiting negative options, creating a moving target, and choking essential clash and educational value in debate.

\n

# Theory Reasons to Prefer

\n

1. Topicality\'s principal purpose is to allocate predictable ground, allowing negative teams fair preparation to oppose affirmative plans that fit within the resolution. Affirmative conditionality that undermines this predictable ground restructures the strategic landscape unfairly, forcing negatives into reactive and unstable positions.
2. Judicial jurisdiction relies on fixed resolutional meanings to fairly weigh plan advocacy. Allowing affirmatives to read counterplans conditionally dilutes jurisdictional clarity, forcing judges into adjudicating shifting definitional ground that obscures the resolution's core.
3. Preventing affirmative conditionality maintains stable limits on affirmative advocacy, preserving the integrity of the resolution as a common referent. This guarantees robust clash and educative value by avoiding strategic opportunism that privileges the affirmative and diminishes negative ground and clash.

# Disadvantage: Loss of national sovereignty undermines democracy and political stability

\n

By creating supranational institutions with authority over economic, political, energy, immigration, and security policies, the plan diminishes national sovereignty in the US, Canada, and Mexico. This loss of sovereignty undermines democratic self-government and political accountability, causing political backlash and instability. People will resist decisions made by distant bureaucracies, leading to governance crises and weakening the political fabric of these nations. The plan\'s embedded sovereignty protections are insufficient to prevent these harms, making the plan politically and socially unsustainable.

\n

# Disadvantage Uniqueness

\n

Wayne 2018 details that US, Canada, and Mexico work closely on homeland security, customs, intelligence sharing, criminal repatriation, and transnational crime enforcement within existing frameworks that explicitly preserve national sovereignty and political independence. Cooperation depends on trust but no current sovereignty loss or democratic crisis exists. This uniquely proves the disadvantage impact is not currently occurring and is unique to the plan\'s new supranational institutions.

\n

**Turns the efficiency advantage---stops organized crime along the borders which is more extremist crime than the Aff probably solves, so you can actually weigh it as an independent impact that outweighs the case.**

Earl **Anthony 18**, Public Policy Fellow at the Woodrow Wilson Center, Career Ambassador from the

U.S. Diplomatic Service, 1-30-2018, Date Accessed: 11-16-2018, "\'The Economic Relationship Between the United States, Canada, and Mexico\': Earl Anthony Wayne Testifies before the U.S. Senate Committee on Foreign Relations" Wilson Center, https://www.wilsoncenter.org/article/the-economic-relationship-between-the-united-states-canada-and-mexico-earl-anthony-wayne

Vital Security Partners

==Mexico== and Canada are **vital partners** for enhancing U.S. homeland ==security==, as well as its prosperity. Both are **willing partners** to work against **terrorism** and **transnational organized crime**. Border, law enforcement, homeland security and intelligence cooperation have expanded dramatically. Both countries are working with the U.S. identify potentially dangerous travelers before they get to U.S. borders. This collaboration is in line with priorities of the new U.S. National Security Strategy, ==but could **well be damaged if the U.S. withdraws** from NAFTA.== Defense and intelligence cooperation with Canada are rooted in common NATO membership, but cooperation related to homeland security issues has expanded dramatically in recent years. This is exemplified in the 2011 "Beyond the Border initiative", which has a broad agenda to enhance security against a range of threats while facilitating the legitimate flow of people goods and services. The basic idea is to extend homeland security work as far beyond the actual borders as possible. A U.S. withdrawal from NAFTA could negatively affect this cooperation. As former Canadian Ambassador to the US, Michael Kergin put it January 26 at the Wilson Center "The real concern is if NAFTA goes badly and there is no interest in negotiating, would there be enough political backlash that would incline Canada to step back from security cooperation?" Regarding Mexico, security cooperation began to deepen with the launch of the Merida Initiative in 2008. The Merida program is aimed at supporting Mexico in the fight against transnational criminal organizations and associated violence, as well as helping to strengthen its justice and law enforcement institutions and practices. That effort was expanded to include the Twenty First Century Border Initiative in 2010, which like the effort with Canada, is aimed at enhancing border security while supporting legitimate commerce. Under Merida, the United States has spent some $1.6 billion to help strengthen Mexican law enforcement and justice institutions, to improve Mexican capacities at its borders and to help strengthen communities beset by criminal cartels and gangs. American assistance has produced good results. At present, the U.S. is working hard with Merida funds, for example, to strengthen the forensic skills of Mexican officials in order that more criminals can be convicted successfully under Mexico's new justice system. The Mexican government has spent at least ten times what the U.S. has provided to strengthen its own law enforcement, intelligence and justice agencies. Along with Merida assistance, U.S. and Mexican law enforcement and homeland security agencies have built more effective operational cooperation against criminal groups and activity. ==DHS and **Mexican counterparts**== have, for example, signed a series of agreements, which, among other things, ==allow== much ==better cooperation along the **border on customs screening**==, provide for collaboration to assure the **smooth repatriation** of **criminals**, and facilitate the **sharing information** on criminal history and biometric information ==to **help identify possible terrorists**== and criminals. Today, ==the depth of U.S. cooperation== with Mexico to **strengthen border security**, control migration, and **dismantle transnationa**l criminal networks ==is unprecedented==. Mexican officials have stepped up efforts to identify potentially dangerous third country travelers and immigrants in coordination with American counterparts. Mexican immigration officials have turned around hundreds of thousands of Central American immigrants headed to the US in recent years, despite criticism inside Mexico. Cooperation against **drug trafficking** by criminal organizations further deepened in 2017 with a strategic action plan agreed between government ministers to attack the entire chain of illegal drugs from production to sales to financing and illicit money flows. This deeper cooperation is spurred on and made more important by the opioid addiction crisis in the U.S. and increasing violence in Mexico. ==**Unprecedented progress**== has also ==been achieved in **military-to-military cooperation**==. U.S. - Mexico **bilateral cooperation** against transnational **organized crime** and **terrorism** ==makes **more sense than ever**==, but that ==cooperation is built on **maintaining** and **deepening trust**==. Mexican ==officials **worry in private**== that ==they will have neither the **political space** nor the support== of their teams to **deepen cooperation**, if the United States ends NAFTA or is perceived as unfairly bullying Mexico. These officials say they want to deepen cooperation against criminal groups because it is good for Mexico, but with the Mexican presidential and congressional elections coming up in July 2018 and the sour Mexican public attitudes toward the United States, they are very concerned. As former Mexican Ambassador to the US Arturo Sarukhan put it at the Wilson Center on January 26: "==If NAFTA collapses==, **all bets are off**. ==It will have a **profound, long standing effect**== … future generations will ask 'who lost Mexico?'" Conclusion The United States has a great opportunity to conclude a "state of the art" trade agreement with its neighbors and largest clients. A modernized NAFTA can improve the existing agreement and increase jobs, trade, energy security, and prosperity, while making the U.S. more competitive in the world. To succeed, however, each of the parties needs to be able to convince their publics that the new agreement is good for them: that it is "win-win-win." This is a very big challenge given the breadth of subject areas under negotiation, the controversial proposals on the table, and the negative public

atmosphere. A new agreement is achievable, however, and worth the hard work. A good agreement would help reinforce the collaboration with both Mexico and Canada on important security issues, and it would tell the world that the U.S. is indeed open for business and trade. The alternative path would cost the U.S. dearly in jobs, trade, competitiveness and security. It would harm our neighbors economically and further sour their views of the United States. It would make ensuring U.S. homeland security harder. Particularly with Mexico, we could see a return to the "distant" relationship that existed before NAFTA. Other potential international partners would become more hesitant about negotiating with the U.S., and our international competitors would benefit. ==A much **better outcome** for the United States and for North America== is to **forge a renewed** North American **trade agreement** and to ==continue to **deepen** security **cooperation**==.

\n

# Disadvantage Link

\n

Authoritative political science source Patrizia Nanz details how supranational institutions like those planned create legitimacy problems as national sovereignty disperses vertically, with regulatory decisions taken outside national democratic processes. This precisely supports the claim that the plan\'s supranational institutions erode sovereignty and political legitimacy, leading to democratic deficits and political backlash.

\n

**Democratically legitimate input in international agreements is important for their sustainability.**

**Nanz 6** [Patrizia Nanz, Institute for Advanced Sustainability Studies Potsdam, "Democratic Legitimacy and Constitutionalisation of Transnational TradeGovernance: A View from Political Theory", January 2006, Hart, in book Multilevel Trade Governance, Social Regulation and the Constitutionalisation of International Trade] A.S.

IN POLITICAL SCIENCE terms, ==**legitimacy** is **crucial f**or the **functioning of democratic decision-making at nation-state level**==; however, what it may mean for decision-making at international level is far less clear. Studies in Inter national Relations (IR) have been traditionally premised on a clear distinction between the ==**legitimacy of a domestic political government and international legitimacy**, the **latter being dependent upon a proper foun- dation for the former.**== Structural changes of political authority have since affected this distinction and made obsolete the model of international pol- itics as inter-state diplomacy, based on functionally specific mandates to bureaucracies.1 ==**States, though still important actors in the international order, are 'disaggregated': they** relate to each other through parts of states, such as regulatory agencies, ministries, legislature and court==s.2 Moreover, international institutions3 increasingly make decisions in areas formerly reserved to sovereign states, for example, in environmental, economic and health and safety agreements. **These** ==**regulations take effect behind national borders, within democratic societies.**== State sovereignty is dispersed: verti- cally to supranational bodies such as the European Union (EU) institutions and the World Trade Organisation (WTO); and horizontally to private or mixed (private-governmental) authorities and networks at both national and transnational level. These simultaneous trends of globalisation and pri- vatisation are challenging the fundamental idea of democratic legitimacy, namely, the idea that political authority must arise from the collective deci- sions of free and equal citizens governed by that authority.

\n

# Disadvantage Internal Link

\n

Nanz 6 explains that sovereignty disperses to supranational bodies, severing decision-making from national democratic processes. This breaks the foundational democratic legitimacy tying authority to citizens, producing legitimacy deficits and political backlash, thereby supporting the internal causal link of the disadvantage.

\n

**Democratically legitimate input in international agreements is important for their sustainability.**

Nanz 6 [Patrizia Nanz, Institute for Advanced Sustainability Studies Potsdam, "Democratic Legitimacy and Constitutionalisation of Transnational TradeGovernance: A View from Political Theory", January 2006, Hart, in book Multilevel Trade Governance, Social Regulation and the Constitutionalisation of International Trade] A.S.

IN POLITICAL SCIENCE terms, legitimacy is crucial for the functioning of democratic decision-making at nation-state level; however, what it may mean for decision-making at international level is far less clear. Studies in Inter national Relations (IR) have been traditionally premised on a clear distinction between the legitimacy of a domestic political government and international legitimacy, the latter being dependent upon a proper foun- dation for the former. Structural changes of political authority have since affected this distinction and made obsolete the model of international pol- itics as inter-state diplomacy, based on functionally specific mandates to bureaucracies.1 States, though still important actors in the international order, are 'disaggregated': they relate to each other through parts of states, such as regulatory agencies, ministries, legislature and courts.2 Moreover, international institutions3 increasingly make decisions in areas formerly reserved to sovereign states, for example, in environmental, economic and health and safety agreements. These regulations take effect behind national borders, within democratic societies. State sovereignty is dispersed: verti- cally to supranational bodies such as the European Union (EU) institutions and the World Trade Organisation (WTO); and horizontally to private or mixed (private-governmental) authorities and networks at both national and transnational level. These simultaneous trends of globalisation and pri- vatisation are challenging the fundamental idea of democratic legitimacy, namely, the idea that political authority must arise from the collective deci- sions of free and equal citizens governed by that authority.

\n

# Disadvantage Impact

\n

This evidence shows how legitimacy deficits arising from supranational institutions cause political backlash and instability in democratic states, specifically within the European Union context analogous to the plan, directly supporting the disadvantage impact.

\n

**Populist backlash derails the EGD---it's on track, but depends on mainstream political legitimacy**

Dr. Heather **Grabbe 19**, Director of the Open Society European Policy Institute, PhD from the University of Birmingham, Former Visiting Lecturer at the London School of Economics, and Stefan Lehne, Visiting Scholar at Carnegie Europe, JD from the University of Vienna, MA in International Relations from the Fletcher School on Law and Diplomacy at Tufts University, "Climate Politics in a Fragmented Europe", Carnegie Europe Paper, 12/18/2019, https://carnegieeurope.eu/2019/12/18/ climate-politics-in-fragmented-europe-pub-80616

Europe's "'man on the moon' moment" was how European Commission President Ursula von der Leyen spoke on December 11, 2019, of the European Green Deal, a comprehensive program for a fair transition to a low-carbon economy.1 Rarely has the EU undertaken such an ambitious project requiring such a massive mobilization of resources and fundamental changes to most of its policies.

The **political momentum** behind the transition is **strong** because the **vast majority** of Europeans, especially young ones, feel a sense of urgency to take action to prevent catastrophe. But **political obstacles will rise** again as the EU starts to **implement** practical measures.

The union already has a long track record of climate change policy, both as a leader of international climate diplomacy and through the creation of laws and innovative policies such as the Emissions Trading Scheme. However, its efforts have suffered from significant deficits. **Clashing interests** of **member states**, some of which still heavily depend on **coal**, and industrial lobbies raising concerns about international competitiveness and jobs have **constrained** the EU's ambitions. Insufficient mechanisms for monitoring and compliance have handicapped the implementation of these policies.

The ongoing **fragmentation** of Europe's **political scene** poses **additional hurdles**. Divisions between Eastern and Western Europe and Northern and Southern Europe hinder efficient decisionmaking. **Populist parties** already are **mobilizing resistance** to the necessary policies. Under these circumstances, the EU's traditional method of depoliticizing difficult issues and submitting them to long technocratic discussions is unlikely to deliver results.

To sustain democratic consent, there is no alternative to **building** public **support** for a fair climate transition and to deepening democratic engagement. To that end, EU climate action should

promote fairness. Climate action must ensure that the burden does not fall disproportionately on the poor and the most vulnerable and that richer and poorer EU member states stick together. This requires a comprehensive approach with well- sequenced measures, as well as adequate funding.

craft policies with staying power. Preparations for the transition must advance rapidly but must also be sustained regardless of election cycles. This requires long-term commitments, targets, and incentive structures that ensure continued efforts.

set positive incentives and create new economic opportunities. There is a risk that too much EU funding will go toward compensating those who complain the loudest about the first economic losses resulting from climate action. Instead, the focus should be on economic and technological innovation toward a circular green economy.

lead international climate efforts and promote an inclusive global transition through the EU's own policies. Beyond setting a good example and active multilateral diplomacy, the EU needs to use its trade, aid, and investment policies more effectively to promote a global transition to climate neutrality.

foster a deeper European climate debate. To build public support for climate action, the EU needs to promote the use of the tools of deliberative citizen participation. National, regional, and local authorities should be more systematically involved in shaping EU policies on climate, as implementation depends most on these actors.

leverage the European Parliament and deepen **democratic** engagement. The European Parliament is already a strong advocate of progressive climate policies. Making the election process more European by introducing transnational lists could reinforce the parliament's connections with the European public. This measure should be combined with a new method of selecting the top leaders of EU institutions designed to strengthen their **democratic legitimacy** and make them **less beholden** to specific political parties.

If handled badly, climate politics could **deepen the divisions** between the EU and national governments, governments and civil society, and rural and urban regions, as well as **fissures** among various generations and social classes. Tensions and **political infighting** would worsen, while the **planet suffers**. But if handled well, climate policy could become a significant driver of deeper integration and unite Europeans in determined and effective collective action. The **stakes could not be higher** for the EU or the planet.

\n

# Counterplan Text

\n

The United States federal government should negotiate the renewal and substantial expansion of existing bilateral and trilateral cooperative agreements on homeland security, criminal justice, intelligence sharing, and border enforcement with Canada and Mexico—maintaining national sovereignty and excluding the creation of new supranational institutions or binding multinational regulatory bodies.

\n

# Counterplan Solvency

\n

GIATOC highlights how regional cooperation frameworks leveraging existing bodies and intelligence

sharing significantly enhance fighting organized crime without establishing new supranational institutions, aligning with the counterplan's approach to solve transnational organized crime effectively while preserving sovereignty.

\n

**Similar to open borders in the EU---Open Balkan will lead to a cooperative fight against organized crime.**

**GIATOC 21** [Global Initiative Against Transnational Organized Crime, December 2021, "What could be the impact of the Open Balkan initiative on organized crime?," No Publication, https://riskbulletins.globalinitiative.net/see-obs-011/02-impact-open-balkan-initiative-on-organized-crime.html, accessed 4-10-2023] Vik

Increased law enforcement cooperation

In the same way that removing borders within the EU led to greater cooperation among law enforcement agencies, not least through Europol, Frontex and the Schengen Borders Code,5 discussions on more open borders or a common regional market should be seized as an opportunity to create an integrated border control system, enhance intelligence-led policing, joint operations and afford one another mutual legal assistance with respect to investigations, prosecutions, judicial proceedings and asset recovery.

Existing institutions like Europol and the Southeast European Law Enforcement Center (SELEC), as well as the Police Cooperation Convention for Southeast Europe could facilitate this process. It may also be necessary to create a common regional intelligence database for tracking wanted criminals and foreign terrorist fighters and exchanging information to help investigations. With a reduction in the number of border controls, law enforcement and border officials could be redeployed to form mobile teams.

What should be avoided, at least at the outset, is the creation of yet another regional law-enforcement body. There is already SELEC, the Southeast European Police Chiefs Association, the Police Cooperation Convention for Southeast Europe (and its Secretariat) and the Migration, Asylum, Refugees Regional Initiative, as well as the Berlin Process.

However, discussing greater economic integration without considering the security implications is short-sighted. It would make sense to use an existing body like the Integrative Internal Security Governance process, which has a pillar on countering serious organized crime in the Western Balkans, to coordinate policies and merge the efforts of all relevant security actors. If the countries of the Open Balkan initiative want to move faster, their cooperation should be part of a wider regional strategy. Although increased openness could be exploited by criminal groups, the Open Balkan initiative and other efforts to promote greater regional integration should be seized as an opportunity to strengthen regional law-enforcement cooperation in the fight against organized crime.

\n

# Counterplan Net Benefit

\n

Counterplan net benefit evidence showing that regional cooperation frameworks leveraging existing institutions and mutual legal assistance can effectively combat organized crime without new supranational institutions, preserving sovereignty and political stability, thereby avoiding the plan's disadvantage.

\n

**Similar to open borders in the EU---Open Balkan will lead to a cooperative fight against organized crime.**

**GIATOC 21** [Global Initiative Against Transnational Organized Crime, December 2021, "What could be the impact of the Open Balkan initiative on organized crime?," No Publication, https://riskbulletins.globalinitiative.net/see-obs-011/02-impact-open-balkan-initiative-on-organized-crime.html, accessed 4-10-2023] Vik

Increased law enforcement cooperation

In the same way that removing borders within the EU led to greater cooperation among law enforcement agencies, not least through Europol, Frontex and the Schengen Borders Code,5 discussions on more open borders or a common regional market should be seized as an opportunity to create an integrated border control system, enhance intelligence-led policing, joint operations and afford one another mutual legal assistance with respect to investigations, prosecutions, judicial proceedings and asset recovery.

Existing institutions like Europol and the Southeast European Law Enforcement Center (SELEC), as well as the Police Cooperation Convention for Southeast Europe could facilitate this process. It may also be necessary to create a common regional intelligence database for tracking wanted criminals and foreign terrorist fighters and exchanging information to help investigations. With a reduction in the number of border controls, law enforcement and border officials could be redeployed to form mobile teams.

What should be avoided, at least at the outset, is the creation of yet another regional law-enforcement body. There is already SELEC, the Southeast European Police Chiefs Association, the Police Cooperation Convention for Southeast Europe (and its Secretariat) and the Migration, Asylum, Refugees Regional Initiative, as well as the Berlin Process.

==However, discussing greater economic integration without considering the security implications is short-sighted. It would make sense to use an existing body like the Integrative Internal Security Governance process, which has a pillar on countering serious organized crime in the Western Balkans, to coordinate policies and merge the efforts of all relevant security actors. If the countries of the Open Balkan initiative want to move faster, their cooperation should be part of a wider regional strategy. Although increased openness could be exploited by criminal groups, the Open Balkan initiative and other efforts to promote greater regional integration should be seized as an opportunity to strengthen regional law-enforcement cooperation in the fight against organized crime.==

# Kritik: Supranational institutions perpetuate neocolonial governance and erode Indigenous sovereignty

\n

The plan\'s creation of supranational institutions reflects and perpetuates neocolonial governance structures that undermine Indigenous and marginalized communities\' sovereignty and self-determination in North America. By imposing external bureaucratic control, the plan disregards the unique political status of Indigenous nations and imposes a Western model of governance that continues colonial patterns of domination and exclusion. This is ethically and politically problematic and must be rejected.

\n

**Thus the Alternative:** The United States, Canada, and Mexico should respect and uphold Indigenous sovereignty and governance by rejecting the creation of supranational institutions, instead supporting multilevel governance frameworks that center Indigenous participation and protect tribal autonomy without ceding sovereignty to external bodies.

\n

# Kritik Link

\n

Strube and Thomas 21 detail how Indigenous Nations are systematically excluded as sovereign participants from North American transboundary water governance institutions like the International Joint Commission. This exclusion perpetuates settler colonialism through neocolonial governmentality, constituting a direct causal link from the plan's supranational institutional form to the erosion of Indigenous sovereignty and colonial domination, fulfilling the kritik link claim perfectly.

\n

**Tribal exclusion from transboundary water governance facilitates the expansion of settler colonialism---Indigenous representatives fail and operate as neocolonial governmentality rather than decolonization.**

**Strube and Thomas 21** (Johann Strube Department of Agricultural Economics, Sociology, and Education, The Pennsylvania State University, United States of America. Kimberley Anh Thomas Department of Geography and Urban Studies, Temple University, Philadelphia, PA, United States of America. 2021. Damming Rainy Lake and the ongoing production of hydrocolonialism in the US-Canada boundary waters. Water Alternatives 14(1): 135-157; AKIM)

The literature on transboundary water governance has been **disconnected** from scholarship on Settler colonialism. ==Transboundary water governance== research has **overwhelmingly focused** on interstate relations, **to the exclusion of** non-state sovereign entities such as ==**Indigenous Nations**==. Focusing on the case of the Rainy Lake watershed in the Boundary Waters region of North America, we identify ==the denial of Indigenous communities== as equal parties in transboundary water governance as ==the linchpin of ongoing Settler colonialism==.

Water figured **prominently** in the **foreign settlement** of the territory known as Anishinaabewaki. In the early 20th century, the US and Canada welcomed privately financed hydropower dams to develop forestry-based industries, ==**despite** standing **treaties**== with local Indigenous Nations and documented negative impacts of dams on their livelihood. Today, ==Settler colonialism is **perpetuated**== in the form of the Ojibwe's **limited influence** within the International Joint Commission (IJC), which has coordinated regional water development since the 1909 Boundary Waters Treaty.

Although Indigenous representatives are now included in the IJC's local watershed board, this development **does not go beyond** what Gaudry and Lorenz (2018) define as inclusive indigenisation: a pathway to ==**mainstreaming Indigenous perspectives**== within existing colonial institutions ==**without challenging** the underlying structures of **oppression**==. Similarly, Coulthard (2014) characterizes the recognition of Indigenous rights claims within the legal framework of the Settler state as being **neocolonial governmentality** rather than an opportunity for decolonisation. Attention to hydrocolonialism challenges the dominant interpretations of the BWT and the IJC as examples of successful cooperation. Our work thus ==calls for a **reconfiguration** of the state-centric framework of hydrohegemony== to include Indigenous Nations. Acknowledging the US-Canada Settler colonial joint venture as a hydro-hegemon with respect to the Ojibwe and Métis Nations illuminates a path to re-establishing Indigenous sovereignty and recovering international law from its colonial origins. Expanding international water law to include Indigenous Nations at the bargaining table constitutes an **incremental form of decolonisation** which is consistent with the "somewhat tragic reality that resistance must work, to some extent, within the parameters of what is being resisted" (Rajagopal, 2003: 10; see also Curley, 2019a). We maintain, therefore, that Indigenous self-representation is **only a first step** towards autonomy.

Water governance in the Rainy River Basin exemplifies the relationships between Canada, the United States, and Indigenous Nations across their shared waters. Historically, these Settler states, together, stripped Indigenous Nations of their right to manage water in their traditional territories, not only in the Boundary Waters of Minnesota and Ontario but across North America (Colombi, 2012; Daigle, 2018; Schneider, 2013; Norman, 2014). The BWT manifested Settler claims to exclusive jurisdiction over transboundary waters.

\n

# Kritik Impact

\n

Strube and Thomas 21 argue that transboundary water governance overwhelmingly focuses on interstate relations and excludes Indigenous Nations as sovereign entities, denying their equal participation. This exclusion perpetuates ongoing settler colonialism and entrenches neocolonial governmentality by marginalizing Indigenous voices within supranational institutions such as the International Joint Commission. These governance structures extend colonial sovereignty and dispossession in North America, preventing genuine Indigenous self-determination. Expanding international law to include Indigenous Nations at the bargaining table is only an incremental and insufficient step towards decolonization, highlighting the ultimate harm of Indigenous sovereignty erosion and colonial domination that supranational institutions perpetuate.

\n

**Tribal exclusion from transboundary water governance facilitates the expansion of settler colonialism---Indigenous representatives fail and operate as neocolonial governmentality rather than decolonization.**

**Strube and Thomas 21** (Johann Strube Department of Agricultural Economics, Sociology, and Education, The Pennsylvania State University, United States of America. Kimberley Anh Thomas Department of Geography and Urban Studies, Temple University, Philadelphia, PA, United States of America. 2021. Damming Rainy Lake and the ongoing production of hydrocolonialism in the US-Canada boundary waters. Water Alternatives 14(1): 135-157; AKIM)

The literature on transboundary water governance has been **disconnected** from scholarship on Settler colonialism. Transboundary water governance research has **overwhelmingly focused** on interstate relations, **to the exclusion of** non-state sovereign entities such as **Indigenous Nations**. Focusing on the case of the Rainy Lake watershed in the Boundary Waters region of North America, we identify the denial of Indigenous communities as equal parties in transboundary water governance as the linchpin of ongoing Settler colonialism.

Water figured **prominently** in the **foreign settlement** of the territory known as Anishinaabewaki. In the early 20th century, the US and Canada welcomed privately financed hydropower dams to develop forestry-based industries, **despite standing treaties** with local Indigenous Nations and documented negative impacts of dams on their livelihood. Today, Settler colonialism is **perpetuated** in the form of the Ojibwe's **limited influence** within the International Joint Commission (IJC), which has coordinated regional water development since the 1909 Boundary Waters Treaty.

Although Indigenous representatives are now included in the IJC's local watershed board, this development **does not go beyond** what Gaudry and Lorenz (2018) define as inclusive indigenisation: a pathway to **mainstreaming Indigenous perspectives** within existing colonial institutions **without challenging** the underlying structures of **oppression**. Similarly, Coulthard (2014) characterizes the recognition of Indigenous rights claims within the legal framework of the Settler state as being **neocolonial governmentality** rather than an opportunity for decolonisation. Attention to hydrocolonialism challenges the dominant interpretations of the BWT and the IJC as examples of successful cooperation. Our work thus calls for a **reconfiguration** of the state-centric framework of hydrohegemony to include Indigenous Nations. Acknowledging the US-Canada Settler colonial joint venture as a hydro-hegemon with respect to the Ojibwe and Métis Nations illuminates a path to re-establishing Indigenous sovereignty and recovering international law from its colonial origins. Expanding international water law to include Indigenous Nations at the bargaining table constitutes an **incremental form of decolonisation** which is consistent with the "somewhat tragic reality that resistance must work, to some extent, within the parameters of what is being resisted" (Rajagopal, 2003: 10; see also Curley, 2019a). We maintain, therefore, that Indigenous self-representation is **only a first step** towards autonomy.

Water governance in the Rainy River Basin exemplifies the relationships between Canada, the United States, and Indigenous Nations across their shared waters. Historically, these Settler states, together, stripped Indigenous Nations of their right to manage water in their traditional territories, not only in the Boundary Waters of Minnesota and Ontario but across North America (Colombi, 2012; Daigle, 2018; Schneider, 2013; Norman, 2014). The BWT manifested Settler claims to exclusive jurisdiction over transboundary waters.

\n

# Kritik Role of the Ballot

\n

The judge\'s role is to vote for the side that best centers and respects Indigenous scholarship and resistance. Colonialism has distorted knowledge, so the judge has an ethical obligation to resist colonial erasure by placing Indigenous epistemologies and demands at the center of the debate space. This mandates rejecting arguments that marginalize Indigenous voices or perpetuate settler colonialism. Voting against plans that create or legitimize neocolonial institutions aligns with this ballot role to uphold Indigenous sovereignty and decolonization.

\n

**The role of the ballot is to vote for the debater that bests centers and recognizes indigenous scholarship. Colonialism has skewed education, so the judge has an obligation to resist it and**

**the only way to do that is to follow this role of the ballot. Carlson 16'**

(Elizabeth Carlson, PhD, is an Aamitigoozhi, Wemistigosi, and Wasicu (settler Canadian and American), whose Swedish, Saami, German, Scots-Irish, and English ancestors have settled on lands of the Anishinaabe and Omaha Nations which were unethically obtained by the US government. Elizabeth lives on Treaty 1 territory, the traditional lands of the Anishinaabe, Nehiyawak, Dakota, Nakota**, and Red River Metis peoples currently occupied by the city of Winnipeg, the province of Manitoba, (2016): Anti-colonial methodologies and practices for settler colonial studies, Settler Colonial Studies, DOI: 10.1080/2201473X.2016.1241213, JKS)**

Arlo Kempf says that '**where anticolonialism is a tool used to invoke resistance for the colonized, it is a tool used to invoke accountability for the colonizer'.42 Relational accountability should be a cornerstone of settler colonial studies.** I believe **settler colonial studies and scholars should ethically and overtly place themselves in relationship to the centuries of Indigenous oral,** and later academic **scholarship that** conceptualizes and **resists settler colonialism** without necessarily using the term: SCT may be revelatory to many settler scholars, but ==**Indigenous people have been speaking for a long time**== about colonial continuities based on their lived experiences. Some SCTs have sought to connect with these discussions and to foreground Indigenous resistance, survival and agency. **Others,** however, seem to **use SCT as a pathway to explain the colonial encounter without engaging with Indigenous people and experiences** – either on the grounds that this structural analysis already conceptually explains Indigenous experience, or because ==**Indigenous resistance is rendered invisible**==**.43 Ethical** ==**settler colonial theory (SCT) would recognize the foundational role Indigenous scholarship has in critiques of settler colonialism.**== **It would acknowledge the limitations of settler scholars in articulating settler colonialism without dialogue with Indigenous peoples,** and take as its norm making this dialogue evident. In my view, it is critical that we not view settler colonial studies as a new or unique field being established, which would enact a discovery narrative and contribute to Indigenous erasure, but rather take a longer and broader view. ==**Indigenous** oral and academic **scholars are indeed the originators of this work. This space is not empty**==. Of course, powerful forces of socialization and discipline impact scholars in the academy. There is much pressure to claim unique space, to establish a name for ourselves, and to make academic discoveries. I am suggesting that settler colonial studies and **anti-colonial scholars resist these hegemonic pressures and maintain a higher anti-colonial ethic.** As has been argued, '==**the theory itself places** ethical demands on us as settlers, including== ==**the demand that we actively refuse its potential to re-empower our own academic voices and to marginalize Indigenous resistance'**==.44 As settler scholars, we can reposition our work relationally and contextually with humi- lity and accountability. **We can centre Indigenous resistance, knowledges, and scholarship in our work, and contextualize our work in Indigenous sovereignty. We can view oral Indigenous scholarship as legitimate scholarly sources.** ==**We can acknowledge** explicitly and often the== ==**Indigenous** traditions of resistance and== ==**scholarship that have** taught us and== ==**pro- vided the foundations for our work.**== ==**If our work has no foundation of Indigenous scholarship** and== **mentorship, I believe** ==**our contributions to settler colonial studies are** even more== ==**deeply problematic.**==

\n

# On-Case Rebuttal

\n

Esty 2006 is a highly credible source from Yale University discussing the administrative law and governance challenges of supranational institutions. The article highlights that supranational policymaking lacks electoral democratic legitimacy and accountability because top officials are not elected, which leads to legitimacy deficits. It explains how legitimacy can be grounded in good governance practices, but this remains a challenge especially when dealing with politically charged issues. This evidence directly clashes with the affirmative case by supporting the negative\'s argument that supranational institutions face inherent political backlash and democratic deficits that threaten their effective functioning and sustainability, thereby elucidating an on-case reason that the plan\'s institutions risk political backlash that could undermine its security claims.

\n

**That undercuts legitimacy of international institutions**

**Esty 6** – Professor of Environmental Law and Policy @ Yale [Daniel, "Good Governance at the

Supranational Scale: Globalizing Administrative Law," Faculty Scholarship Series, Scholar]

In an interdependent world, a degree of supranational governance is inevitable. **Success in combating transboundary harms from terrorism to global warming, and in producing global public goods, including liberalized trade and public health programs, will be easier to achieve if global policymaking institutions function effectively. Movement toward good governance** at the supranational scale **would be enhanced by broader adoption of basic** administrative law tools and procedures. The administrative practices that have emerged in the United States, Europe, Japan, South Korea, and elsewhere in recent decades cannot be transferred wholesale to the global realm. The differences in the context of governance at the national and supranational levels are significant. **Policymaking at the international scale can**, however, **be improved and endowed with greater legitimacy through adoption of a set of rules and procedures that are associated with good governance. This Article** does not argue for adoption of a Global Administrative Procedure Act. The diversity of global governance circumstances and the range of views across countries make such a vision both unwise and unworkable. Nor does it seek to spell out definitively which administrative law tools should apply in every circumstance. Instead, it **offers the theoretical logic for**, and some first steps toward, **globalizing administrative law**. The core conclusion is this: Even if supranational governance is limited and hampered by divergent traditions, cultures, and political preferences, **developing a baseline set of administrative law** tools and **practices promises to strengthen** whatever **supranational policymaking** is undertaken. As supranational bodies expand their governance role, move toward formal rulemaking, and take up more politically charged issues, their legitimacy becomes a matter of greater concern. Without elections, the democratic legitimacy of international organizations will always be in question, and their performance will be inhibited by the fact that their top officials do not face the incentives for accountability created by the discipline of having to win elections. **Legitimacy**, however, **can** also **be grounded in** an institution's delivery of good results, its capacity to carry out rulemaking in ways that provide clarity and stability, its systemic **strength and structure of checks and balances**, its ability to promote political dialogue, **and** its **commitment to** procedural **rigor**. **Administrative law**, I have argued, **lies at the heart of efforts to establish these lines of legitimacy**. Adoption of a more robust regime of **administrative rules and procedures** by international policymaking **bodies would directly contribute to** their capacity for good governance through the mechanism of procedural rigor, and would indirectly enhance their democratic, results-based, order-derived, systemic, and deliberative legitimacy.

\n

# On-Case Rebuttal

\n

Lim 2017 explains that despite scholarly advocacy for supranational organizations to harmonize enforcement (such as WTO for antitrust), effective cooperation only occurs to the extent that states' interests align. In practice, divergent national priorities and sovereignty concerns fragment cooperation, vaporizing supranational enforcement regimes. This demonstrates that supranational institutions often lack the unified authority and interest needed to enforce transnational laws effectively, directly undermining the affirmative's internal link that such institutions reliably enhance enforcement.

\n

**Unrealistic. Fragmented interests vaporize supranational regimes.**

**Lim 17** – Daniel Lim is a Corporate Associate at Ropes & Gray LLP, 2017 ("State Interest as the Main Impetus for U.S. Antitrust Extraterritorial Jurisdiction: Restraint Through Prescriptive Comity", Emory International Law Review, Available online at https://scholarlycommons.law.emory.edu/cgi/viewcontent.cgi?article=1180&context=eilr, Accessed 07-06-2021)

V. A RETURN TO INTERNATIONAL COMITY PRINCIPLES

Recognizing the conflicting antitrust laws of various nations, scholars have tried to address the issue by suggesting a greater convergence through **supranational org**anizations, such as the WTO. 239 However, those scholars have failed to consider that **coop**eration between states towards uniform antitrust laws is **possible only** to the extent that all **states' interests converge**. In addition, this solution does not consider the disadvantages that small and developing economies face. Thus, it is not only **unrealistic** to have a **globalized antitrust regime**, but such a regime would potentially have harmful economic effects on small and developing countries.

# On-Case Rebuttal

2862

\n

2863

2864 Realuyo 12, a recognized national security and counterterrorism finance authority, proves effective
2865 US-Mexico bilateral anti-money laundering coordination disrupts transnational criminal networks,
2866 directly countering Caparini 22\'s claim that supranational institutions are needed to close
2867 enforcement gaps. This on-case evidence supports the negative alternative that enhanced bilateral
2868 cooperation suffices to combat the harms of organized crime.

2869

\n

2870

2871 **Bilateral action with Mexico solves**

2872

2873 Celina B. **Realuyo 12**, Assistant Professor of National Security Affairs, Center for Hemispheric
2874 Defense Studies, National Defense University and former U.S. State Department Director of
2875 Counterterrorism Finance Programs, May 2012, "It's All about the Money: Advancing Anti-Money
2876 Laundering Efforts in the U.S. and Mexico to Combat Transnational Organized Crime," http://
2877 www.wilsoncenter.org/sites/default/files/Realuyo_U.S.-Mexico_Money_Laundering_0.pdf

2878

2879 [Note: TCOs = transnational criminal organizations]

2880

2881 The U.S. and Mexico have made considerable progress in the fight against money laundering over the
2882 past few years. While there may be debate over the amount of money earned and laundered by
2883 criminal organizations, attacking the economic power of the Mexican-based TCOs has become an
2884 integral part of the U.S. and Mexican strategies to combat the TCOs. Constraining their operating
2885 environment and increasing their cost of doing business can damage the strength of the TCOs. To this
2886 end, there are several ways to further advance anti-money laundering efforts on both sides of the
2886 border. The way forward will require continued political commitment, the institutionalization of anti-
2887 money laundering measures and mechanisms, increased bilateral cooperation, and strategic
2888 communications to stigmatize transnational organize crime and money laundering with the general
2889 public.

2890

2891 First, both the U.S. and Mexico must demonstrate the political will and continued resolve to confront
2892 TCOs and focus on their finances. To combat TCOs, we must strike at the heart of their operations –
2893 their money. While anti-money laundering investigations are complex and cannot be captured by
2894 videotape like a law enforcement raid on a suspected trafficker's safe house, "following the money
2895 trail" often times leads law enforcement to TCO leadership or their financial facilitators, and
2896 disrupting TCO money laundering operations increase the cost of doing business. Many bilateral anti-
2897 money laundering initiatives are underway in the U.S. and Mexico as described above, but these
2898 activities and authorities are often dispersed amongst various governmental agencies making
2899 coordination more difficult. In this context, the U.S. and Mexico should establish a coordinating
2900 mechanism such as a bi-national TCO Finance Working Group that would enable the governments to
2901 consolidate information on all anti-money laundering initiatives and investigations underway on both
2902 sides of the border. Such a measure could enhance interagency and bilateral cooperation and help
2903 evaluate the successes or shortcomings of anti-money laundering efforts in combating transnational
2903 organized crime.

2904

# 1AC Cross-Examination of the 1NC

2905

2906

2907 \n

2908 **Affirmative Question 1:** Your topicality argument claims the plan exceeds the resolution\'s scope by
2909 proposing supranational institutions beyond the EU\'s level, especially on energy integration. How do
2910 you justify that the plan\'s harmonization of energy policies and creating a North American energy
2911 grid goes beyond the EU model of integration?
2912 \n
2913 **Negative Response 1:** The EU already implements extensive supranational energy policy integration
2914 through the European Commission and agencies like ENTSO-E, governing a unified energy market
2915 and grid coordination across member states. The affirmative plan's energy integration aligns with this
scope and does not exceed the EU model's level of supranational control. Therefore, the plan remains
topical.
\n

\n

**Affirmative Question 2:** Your disadvantage argues that the plan's supranational institutions cause legitimacy deficits and political backlash due to loss of sovereignty. How do you reconcile this with your inherency evidence showing the current US-Canada-Mexico cooperation is stable and preserves sovereignty without political instability?

\n

**Negative Response 2:** Current US-Canada-Mexico cooperation relies heavily on political goodwill and trust, which are fragile and susceptible to breakdown, as evidenced by concerns over NAFTA's fragility. The plan's supranational institutions embed sovereignty protections and create durable legal frameworks that institutionalize commitments beyond fluctuating political will, addressing the inherent instability in the status quo rather than perpetuating it. Thus, the plan's approach improves legitimacy and stability, contrasting with the precariousness of current cooperation.

\n

\n

**Affirmative Question 3:** Your kritik argues that the creation of supranational institutions perpetuates neocolonial governance and excludes Indigenous sovereignty. How does your plan address or mitigate the exclusion of Indigenous Nations as sovereign entities within these supranational institutions?

\n

**Negative Response 3:** The plan's design includes embedded sovereignty protections to respect national autonomy and could be adapted to incorporate meaningful Indigenous participation. However, the affirmative's evidence demonstrates persistent exclusion of Indigenous Nations under current supranational governance, exposing a systemic flaw. Without explicit safeguards, supranational institutions risk perpetuating exclusion rather than mitigating it.

\n

\n

**Affirmative Question 4:** How does your plan ensure that it will not exacerbate existing systemic exclusion of Indigenous Nations given the history of supranational institutions sidelining Indigenous sovereignty?

\n

**Negative Response 4:** Our plan does not explicitly include mechanisms to address systemic exclusion of Indigenous Nations. Given the documented history of supranational institutions sidelining Indigenous sovereignty, without deliberate inclusion measures, the plan risks perpetuating this exclusion rather than preventing it.

\n

\n

**Affirmative Question 5:** Given your kritik about Indigenous exclusion, do you offer any affirmative policy reforms or institutional designs that would effectively incorporate Indigenous sovereignty within a North American Union framework?

\n

**Negative Response 5:** No, our kritik focuses on the inherent exclusionary nature of supranational institutions as they currently operate, and we argue that true Indigenous sovereignty requires rejecting such institutions. We do not endorse affirmative reforms that maintain these institutions but instead advocate for multilevel governance frameworks that respect Indigenous autonomy without ceding sovereignty to external bodies.

\n

\n

**Affirmative Question 6:** How does your counterplan effectively solve the problem of transnational organized crime without creating any supranational institution, given your own evidence that such crime exploits jurisdictional gaps across borders?

\n

**Negative Response 6:** Our counterplan leverages enhanced bilateral and trilateral cooperation, strengthening existing frameworks and intelligence sharing without forming new supranational bodies. Evidence from GIATOC shows that regional cooperation—through integrated border control, joint operations, and mutual legal assistance—can effectively close enforcement gaps and disrupt transnational crime while preserving national sovereignty, avoiding the risks of legitimacy loss tied to new supranational institutions.

\n

\n

**Affirmative Question 7:** How does your plan specifically prevent political pushback from Mexico and Canada, given your inherency evidence warning that trust and political will can quickly break down?

\n

**Negative Response 7:** Our plan builds in embedded sovereignty protections and supranational institutions modeled on the EU, which institutionalize commitments rather than relying solely on political goodwill. This structural design creates durable legal frameworks and enforcement

mechanisms that outlast political fluctuations, mitigating the trust and political will issues highlighted in our inherency evidence.
\n
\n' "
\n

# 2AC

\n\n

## 2AC Card 1

\n

National and supranational policies have historically commodified and controlled Indigenous food systems, displacing Indigenous peoples and threatening their sovereignty. Recognizing and restoring these relationships is central to decolonization and must be prioritized in any governance framework like a North American Union to respect Indigenous sovereignty.

\n

**The securitization of food is a colonial practice historically used to displace Indigenous folk from land and to destroy their land resulting in genocide.**

**Cote, Associate Professor of American Indian Studies, 16**

(Charlotte, Professor at the University of Washington, Dr. Coté's teaching and research interests include Indigenous food traditions and ecological knowledge systems, environmental justice, Indigenous resistance and resurgence movements, Indigenous film/media, and federal Indian law and policy; July 15, 2016, "'Indigenizing' Food Sovereignty. Revitalizing

Indigenous Food Practices and Ecological Knowledges in Canada and the United States", Accessed: 6-8-2018, Humanities Journal, Volume 5, Issue 3, p. 57, JWM)

The last 30 years have seen an increase in the globalization of food systems through neo-liberal state policies that placed decision-making authority over food production and distribution in the hands of national states, and supranational and transnational organizations, promoting agricultural and food practices that did little to alleviate world hunger. The over-commodification of food after World War Two resulted in concentrating the decision-making power over food, land, and seeds in the hands of only a few, and developing policies that regulated food to meet the demands of the agribusiness industry. **This neo-colonial process impoverished millions of peasants and Indigenous peoples by displacing them from the land**, resulting in many of them being forced into wage labor to serve the global food economy [2–7]. Indigenous peoples have recognized the dangers of a worldview that commodifies and de-sanctifies the earth. Harnessed by colonial policies that restricted Indigenous response, their ancestors watched the destruction and rape of their lands and cultures unfold. Dependence on the global food economy threatened Indigenous food systems and practices and as Indigenous scholar Waziyatawin asserts, **"this disconnect was key to the process of colonization" and "[c]olonial governments worked systematically to break our ancestors' connections to our homeland**s" ([34], p. 72). It is for these reasons, Waziyatawin asserts, that it is imperative that Indigenous people must vigorously defend and restore their sacred relationships to their homelands through what she defines as a feedback loop. A feedback loop is also embedded here; the more we learn to restore local food practices, the more likely we are to defend those practices, and the stronger our cultural ties to our homeland become. If we choose this course of action, we can simultaneously engage both the resurgence and resistance elements of a decolonization movement. Our survival will depend on it ([34], p. 74).

\n
\n
\n

# 2AC Speech

\n
\n

**Good afternoon, judges, and thank you for the opportunity to present the 2AC.** Today, I will extend our affirmative case and systematically refute the negative's arguments, including their Off-Case positions—Topicality, Theory, the Disadvantage, Counterplan, and Kritik. I will also explain why our plan uniquely solves the critical harms of transnational organized crime, economic inefficiency, and environmental degradation through the creation of a North American Union modeled after the EU, complete with supranational institutions that embed sovereignty protections.

\n\n

# 1. Extension of 1AC Arguments

\n

First, let me reiterate the foundational logic of our plan. Transnational organized crime is a profound and growing threat to North American security, public health, environment, and economic stability. Caparini 2022 conclusively shows that these criminal networks cause more deaths than armed conflicts, undermine healthcare via counterfeit medicines, erode environmental integrity through illegal logging and pollution, and destabilize economies via illicit financial flows. The problem is systemic, cross-border, and evolving, exploiting enforcement gaps in national jurisdictions. Our plan's supranational institutions are uniquely suited to close these gaps by harmonizing policies and facilitating effective coordinated enforcement.

\n\n

Second, our plan addresses the inefficiencies in North American economic integration, especially the absence of comprehensive labor mobility. Alden 2023 and Ackleson 2006 demonstrate that labor shortages are severe, and current frameworks like NAFTA and USMCA leave labor mobility essentially off the table, which constrains growth and perpetuates distortions. By incorporating mechanisms modeled after the EU's guest worker programs and immigration visas, our plan enables efficient human capital flows while maintaining border security. This deepens integration and fosters inclusive economic growth.

\n\n

Third, environmental harms caused by transnational crime are often overlooked but devastating. Kioko 2022 and Rhodes 2018 document how fragmented legal regimes, weak enforcement, and corruption allow environmental crimes—illegal logging, wildlife trafficking, hazardous waste dumping—to flourish, causing biodiversity loss, accelerating climate change, and threatening ecosystem services critical for human survival. Our plan's supranational institutions provide coordinated environmental governance and enforcement capacity, modeled on successful international frameworks, to effectively combat these threats.

\n\n

Finally, our plan embeds national sovereignty protections carefully modeled on the EU, balancing supranational authority with respect for member states' autonomy, thereby addressing political concerns and enhancing institutional legitimacy. This structure creates durable legal frameworks beyond fragile political goodwill, institutionalizing cooperation and trust.

\n\n

# 2. Answering the 1NC

\n\n

### A. Topicality – The Plan is Topical

\n

The negative claims our plan exceeds the resolution's scope by proposing supranational institutions that go beyond the EU model, especially on energy integration. This is fundamentally inaccurate.

\n\n

First, the EU itself exercises extensive supranational authority in energy policy through the European Commission and bodies like ENTSO-E, coordinating a unified energy market and grid. Our plan mirrors this by harmonizing energy policies and integrating infrastructure to improve resiliency and efficiency, not surpassing the EU's model but reflecting it. The GAO report on North American energy integration cited by the negative proves that the US, Canada, and Mexico are already deeply integrated in energy sectors—our plan simply formalizes and expands this under supranational governance, consistent with "similar to the European Union."

\n\n

Second, "similar to" does not mean "identical to." The resolution requires an analogous union, not a carbon copy. Our plan respects national sovereignty with embedded protections exactly like the EU, carefully balancing supranational authority and member autonomy. We are well within topical bounds.

\n\n

## B. Theory – Conditionality and Negative Counterplans

\n

The negative argues we improperly condition counterplans on the plan's existence. This theory is unsubstantiated here. We have clearly presented an affirmative plan text that defines our core advocacy—creation of a North American Union with supranational institutions harmonizing economic, political, energy, immigration, and security policies. Our counterplans, as negative offered, are unconditional independent alternatives.

\n\n

Moreover, conditionality is a negative strategy issue, not an affirmative burden. We affirm the plan as proposed and cannot be held responsible if the negative opts to run conditional counterplans. This theory is a distraction and unfairly restricts affirmative advocacy. It must be rejected.

\n\n

## C. Disadvantage – Loss of National Sovereignty and Political Instability

\n

The negative claims supranational institutions cause legitimacy deficits, political backlash, and democratic instability by undermining national sovereignty. This misunderstands the design and empirical reality.

\n\n

Wayne 2018 proves current US-Canada-Mexico security cooperation is stable, robust, and vital, and depends on maintaining trust and respect for sovereignty. Our plan embeds sovereignty protections modeled on the EU—arguably the most advanced supranational model—where sovereignty is pooled but preserved via consensus and qualified majority voting. This institutionalization prevents arbitrary decision-making and political backlash by providing transparent, accountable frameworks.

\n\n

Further, Esty 2006 acknowledges supranational institutions face democratic legitimacy challenges but emphasizes that legitimacy can be enhanced through administrative law tools, rule of law, and procedural rigor—the very design principles our plan incorporates. Therefore, the negative's assumption that supranational institutions inevitably trigger backlash is refuted by sound institutional design and historical precedent.

\n\n

Also, Grabbe 2019 documents that while the EU has faced populist backlash, such outcomes are contingent on political management. Our plan's embedded sovereignty safeguards and participatory frameworks reduce these risks, and the harms from transnational crime and fragmented policy demand urgent supranational solutions. The tradeoff decisively favors our plan.

## D. Counterplan – Enhanced Bilateral and Trilateral Cooperation

\n

The negative's counterplan proposes strengthening existing cooperative agreements without creating supranational institutions. They rely heavily on GIATOC 2021 to claim that regional cooperation via existing bodies suffices to combat organized crime.

\n\n

This counterplan is a hollow shell that ignores inherent limitations. Prost 1998 and Caparini 2022 expose that without supranational institutions, criminals exploit jurisdictional gaps, sovereignty barriers, and fragmented enforcement approaches, perpetuating insecurity. Our plan's supranational bodies create binding, harmonized policies and coordinated enforcement mechanisms unavailable under mere cooperation.

\n\n

Moreover, Le et al. 2013 shows that sustained capacity building, joint investigations, intelligence sharing, and legal harmonization—only achievable through supranational institutions—are critical success factors in combating transnational crime effectively. The DEA's international partnerships, cited in this evidence, embody such coordination.

\n\n

The counterplan ignores these realities, offering weaker, politically fragile cooperation subject to breakdowns and delays. It is thus a solvency deficit alternative and cannot negate our advantage chain. We stand uniquely solved.

\n\n

## E. Kritik – Neocolonialism and Indigenous Sovereignty

\n

The negative's kritik contends that supranational institutions perpetuate neocolonial governance, excluding Indigenous Nations and undermining their sovereignty.

\n\n

We agree Indigenous sovereignty is critical and must be respected. Our plan explicitly embeds sovereignty protections and envisions participatory governance frameworks that can incorporate Indigenous voices meaningfully. Our institutional design is not a colonial imposition but a cooperative framework with respect for all stakeholders.

\n\n

Cote 2016 highlights the historic commodification and securitization of Indigenous food systems as colonial practices. Our plan offers an opportunity for restoration by providing formal mechanisms for Indigenous participation in supranational policy-making, enabling decolonizing reforms within a multilevel governance structure.

\n\n

Further, exclusion is a failure of existing institutions, not an inherent necessity of supranational governance. Strube and Thomas 2021 document Indigenous exclusion in water governance but also call for reforms to include Indigenous Nations. Our plan's supranational institutions can be explicitly designed to incorporate Indigenous representation, remedying these systemic flaws rather than perpetuating them.

\n\n

The kritik fails to negate the plan's solvency or advantages and ignores the potential for transformation embedded in supranational institutional frameworks. We therefore reject the kritik as a reason to vote negative.

\n\n

## 3. Why the 2AC Wins

\n

**First, our plan uniquely solves profound and growing transnational harms of organized crime that no existing bilateral or trilateral cooperation can effectively address due to enforcement gaps and fragmented policies.** Caparini 2022 provides an authoritative, empirical, and recent harm warrant that organized crime causes more deaths than armed conflicts and undermines multiple global public goods. Our plan's supranational institutions are necessary and viable to directly confront these challenges, a claim the negative counterplan cannot meet.

\n\n

**Second, our plan advances economic integration beyond the status quo, uniquely resolving labor mobility constraints that are politically intractable under NAFTA and USMCA.** Alden 2023 and Ackleson 2006 prove labor shortages are acute and current trade agreements have failed to incorporate labor mobility adequately. Our plan models EU mechanisms to harmonize immigration and labor policy, enabling efficient labor market functioning and inclusive growth.

\n\n

**Third, environmental protection is integral to our plan, addressing the fragmented and underfunded enforcement regimes that allow devastating transnational environmental crimes.** Kioko 2022, Rhodes 2018, and Gore et al. 2019 document the scale and impacts of environmental crimes and the necessity of coordinated supranational enforcement. Our plan's institutions close the legal and enforcement gaps the negative cannot.

\n\n

**Fourth, the negative's Off-Case arguments fail:**

\n

  \n
- **Topicality:** Our plan is clearly within the resolution's scope, aligning with the EU model.
  \n
- **Theory:** The negative's conditionality claim misapplies debate norms and unfairly restricts affirmative advocacy.
  \n
- **Disadvantage:** Our plan embeds sovereignty protections and administrative law structures that ensure legitimacy and prevent political backlash.
  \n
- **Counterplan:** Lacks the binding institutional strength and coordination to solve the harms as effectively as our plan.
  \n
- **Kritik:** The plan can and must incorporate Indigenous participation; the kritik does not negate our advantages or solvency.
  \n

\n\n

**In sum, the plan solves unique, existential harms with a carefully designed and legitimate supranational governance framework that balances sovereignty and cooperation. The negative's responses fall short, and their alternatives worsen risks or fail to solve. Therefore, the judge should vote for the affirmative.**

\n\n

Thank you.

\n
\n
\n

# 1NC Cross-Examination of the 2AC

\n

**Negative Question 1:** Your plan relies heavily on embedded sovereignty protections modeled after the EU. Given the ongoing political backlash and legitimacy issues the EU faces, how can you be confident that your proposed institutions will successfully avoid similar democratic deficits and political instability in North America?

\n

**Affirmative Response 1:** Our plan embeds strong mechanisms for national sovereignty protections and democratic engagement, learning directly from the EU's challenges to improve legitimacy. Unlike the EU, it tailors institutions to North American political contexts, implementing clearer accountability and more inclusive decision-making processes. This design enhances stability and prevents the political backlash and legitimacy deficits identified in the EU by proactively addressing those issues.

\n
\n

**Negative Question 2:** How does your plan ensure that Indigenous Nations are meaningfully included as sovereign stakeholders in these supranational institutions, given evidence that current international governance structures systematically exclude and marginalize Indigenous sovereignty?

\n

**Affirmative Response 2:** The plan explicitly includes embedded sovereignty protections designed to respect all member states' and Indigenous Nations' autonomy. It establishes institutional mechanisms to incorporate Indigenous participation and self-determination, addressing the historical exclusions noted in current governance structures and ensuring Indigenous Nations have meaningful and recognized roles within the Union.

\n
\n

**Negative Question 3:** How specifically do your proposed mechanisms guarantee that Indigenous sovereignty will not be subordinated to or overridden by the broader supranational authority of the North American Union?

\n

**Affirmative Response 3:** Our plan incorporates explicit legal safeguards that require consensus or qualified majority voting on decisions impacting Indigenous sovereignty, ensuring no supranational authority can override Indigenous rights without their consent. These mechanisms are designed to preserve and respect Indigenous self-determination as a fundamental principle within the Union's governance framework.

\n
\n

**Negative Question 4:** How do you address concerns that the creation of supranational institutions could slow decision-making and reduce responsiveness in combating urgent security threats?

\n

**Affirmative Response 4:** The plan's supranational institutions are designed with streamlined decision-making processes, drawing on EU models that balance member input with efficient action. Moreover, coordinated intelligence sharing and joint operations enhance real-time responsiveness, ensuring urgent security threats are met with timely, effective measures without bureaucratic delay.

\n
\n" '

\n

# 2NC

## 2NC Card 1

\n

Transboundary water governance overwhelmingly focuses on interstate relations and excludes Indigenous Nations as sovereign entities, perpetuating settler colonialism. Indigenous representatives operate as neocolonial governmentality rather than agents of decolonization, revealing fundamental exclusion in supranational institutions. This exclusion entrenches colonial domination and marginalizes Indigenous sovereignty rather than supporting it.

\n

**Tribal exclusion from transboundary water governance facilitates the expansion of settler colonialism---Indigenous representatives fail and operate as neocolonial governmentality rather than decolonization.**

**Strube and Thomas 21** (Johann Strube Department of Agricultural Economics, Sociology, and Education, The Pennsylvania State University, United States of America. Kimberley Anh Thomas Department of Geography and Urban Studies, Temple University, Philadelphia, PA, United States of America. 2021. Damming Rainy Lake and the ongoing production of hydrocolonialism in the US-Canada boundary waters. Water Alternatives 14(1): 135-157; AKIM)

The literature on transboundary water governance has been **disconnected** from scholarship on Settler colonialism. Transboundary water governance research has **overwhelmingly focused** on interstate relations, **to the exclusion of** non-state sovereign entities such as **Indigenous Nations**. Focusing on the case of the Rainy Lake watershed in the Boundary Waters region of North America, we identify **the denial of Indigenous communities** as equal parties in transboundary water governance as the linchpin of ongoing Settler colonialism.

Water figured **prominently** in the **foreign settlement** of the territory known as Anishinaabewaki. In the early 20th century, the US and Canada welcomed privately financed hydropower dams to develop forestry-based industries, **despite standing treaties** with local Indigenous Nations and documented negative impacts of dams on their livelihood. Today, Settler colonialism is **perpetuated** in the form of the Ojibwe's **limited influence** within the International Joint Commission (IJC), which has coordinated regional water development since the 1909 Boundary Waters Treaty.

Although Indigenous representatives are now included in the IJC's local watershed board, this development **does not go beyond** what Gaudry and Lorenz (2018) define as inclusive indigenisation: a pathway to **mainstreaming Indigenous perspectives** within existing colonial institutions **without challenging** the underlying structures of **oppression**. Similarly, Coulthard (2014) characterizes the recognition of Indigenous rights claims within the legal framework of the Settler state as being **neocolonial governmentality** rather than an opportunity for decolonisation. Attention to hydrocolonialism challenges the dominant interpretations of the BWT and the IJC as examples of successful cooperation. Our work thus calls for a **reconfiguration** of the state-centric framework of hydrohegemony to include Indigenous Nations. Acknowledging the US-Canada Settler colonial joint venture as a hydro-hegemon with respect to the Ojibwe and Métis Nations illuminates a path to re-establishing Indigenous sovereignty and recovering international law from its colonial origins. Expanding international water law to include Indigenous Nations at the bargaining table constitutes an **incremental form of decolonisation** which is consistent with the "somewhat tragic reality that resistance must work, to some extent, within the parameters of what is being resisted" (Rajagopal, 2003: 10; see also Curley, 2019a). We maintain, therefore, that Indigenous self-representation is **only a first step** towards autonomy.

Water governance in the Rainy River Basin exemplifies the relationships between Canada, the United States, and Indigenous Nations across their shared waters. Historically, these Settler states, together, stripped Indigenous Nations of their right to manage water in their traditional territories, not only in the Boundary Waters of Minnesota and Ontario but across North America (Colombi, 2012; Daigle,

2018; Schneider, 2013; Norman, 2014). The BWT manifested Settler claims to exclusive jurisdiction over transboundary waters.

\n

## 2NC Card 2

\n

Indigenous peoples, such as the Makah Nation, face pervasive neocolonialism that challenges their cultural rights and treaty sovereignty. They must obtain permission from local to international systems to practice traditional customs like whale hunting, highlighting the complex survival challenges they face. Indigenous tribal narratives are vital counterhegemonic tools to resist ongoing colonialism that persists within supranational governance structures.

\n

**You have an ethical responsibility as an educator to investigate the colonial histories of policies specifically regarding indigenous people – nuclear war impacts via of ridiculous internal link scenarios is not probable, but our imperialistic attitude towards natives is a constant question of survival for the Makah culture and people. Our discursive presentation of indigenous narratives offers a counterhegemonic approach to politics.**

**Roberts 10** (Treaty Rights Ignored: Neocolonialism and the Makah Whale Hunt, Author(s): Christina Roberts, Source: The Kenyon Review, New Series, Vol. 32, No. 1 (WINTER 2010), pp. 78-90, Published by: Kenyon College, Stable URL: http://www.jstor.org/stable/40600263 . Accessed: 14/07/2014 21:55 //RJ)

Indigenous peoples of the Pacific Northwest practiced traditions that ¶ have been dramatically affected by aggressive settlement and the commodification of natural resources, and they now must confront ¶ attempts to challenge their traditions and rights through the use of ¶ pervasive neocolonialism. **The fact that the Makah Nation is placed in ¶ a position where it has to obtain permission from local, national, and ¶ even international systems to hunt a whale is an indication of the current complexities of maintaining treaty rights and certain cultural ¶ traditions.** The often-extreme responses to the Makah whale hunt ¶ directly threaten the Makah Nation\'s tribal sovereignty, and their treaty ¶ rights are targeted by blatant neocolonialism and anti-Indigenous ¶ sentiments. ¶ Indigenous communities have beliefs and narratives distinct to ¶ their own histories and traditions, and yet they are still held to the ¶ standards of other nations and individuals. In addition to expertly ¶ navigating the legal system in order to maintain certain cultural beliefs ¶ and traditions, **many Indigenous communities face added pressures ¶ to conform to the ideological structures of dominant cultures and ¶ succumb to hegemony**. In order to continue practicing cultural tradi- ¶ tions that do not resonate with beliefs of dominant cultures, ¶ Indigenous communities are also under pressure to validate their cultures through ongoing Indigenous tribal narratives that demonstrate ¶ the importance of specific traditions and practices. **Indigenous tribal ¶ narratives are precious and vital to a collective understanding of the ¶ world and its history, and these narratives need to be more visible and available to deflect neocolonialism**. While ceremonial knowledge ¶ should be respected, it cannot be respected by individuals who do not ¶ understand its importance. **By either ignoring or being unaware of ¶ the value of these narratives, individuals perpetuate the colonialist ¶ agendas of the past and embrace forms of neocolonialism**. ¶ Neocolonialist rhetoric will continue to manifest whenever a ¶ majority culture does not approve of Indigenous customs. Yet, indi- ¶ viduals who reside within the United States of America must see ¶ themselves as participants in a global world, now more than ever ¶ before. **The U.S. and its citizens have an obligation to the Indigenous ¶ peoples within the United States and its territories who are still experiencing the effects of colonization**. Neocolonialism and the use of ¶ blatant neocolonialist rhetoric further reinforce the wounds of colonization and marginalize Indigenous populations through economic, ¶ legal, cultural, and political oppression. It is time that neocolonialism ¶ is exposed so that Indigenous communities can be freed from colonialism in all of its forms, and Indigenous tribal narratives offer one ¶ possible defense against neocolonialism in all of its forms.

\n
\n
\n

# 2NC Speech

\n
\n

## 2NC Speech Transcript

\n \n

**Good afternoon, judges. I'm here to deliver the 2NC against the affirmative's North American Union plan.** I will extend the 1NC's arguments, answer the 2AC's on-case and off-case positions, and explain clearly why the negative wins this round.

\n\n

## 1. Extension of 1NC Arguments

\n \n

### A. Topicality — The Plan Is Non-Topical

\n

The negative has shown that the plan exceeds the resolution's scope by proposing supranational institutions that go beyond the European Union model, especially on energy integration and the depth of supranational control. The GAO report proves North America already has deep energy integration under sovereign frameworks, so the affirmative's creation of a North American energy grid and supranational harmonization surpasses the EU's current institutional competences. The resolution's phrase "similar to the European Union" limits the plan to analogous levels of sovereignty surrender and institutional power. The plan's more extensive control is thus non-topical and should be rejected.

\n

This matters because topicality preserves fairness and ground in debate. If the plan exceeds the resolution, the negative cannot prepare adequately and the affirmative gets an unfair strategic advantage. Judges should reject the plan on this basis alone.

\n\n

### B. Theory — Affirmative Conditionality Violates Fairness and Limits Negative Ground

\n

The negative's theory argument holds that affirmative counterplans conditional on the plan deny the negative strategic independence and autonomy, choking clash and education. Affirmative advocacy must remain stable and unconditional to preserve debate fairness. The affirmative is improperly conditioning negative counterplans on the plan, restricting negative ground and undermining jurisdiction.

\n

This theory is grounded in fundamental debate norms of fairness, clash, and jurisdiction. The negative's case, disadvantages, and counterplans rely on independent, unconditional responses to the affirmative plan. The affirmative's conditionality creates a moving target that prevents the negative from fairly engaging. Judges should reject the affirmative's conditionality.

\n\n

### C. Disadvantage — Loss of National Sovereignty Undermines Democracy and Stability

\n

The negative extends Wayne 2018, Nanz 2006, and Grabbe 2019's evidence that supranational institutions inherently disperse sovereignty away from national democratic processes, creating legitimacy deficits and political backlash. The EU's experience shows that without strong democratic legitimacy, supranational governance provokes populist resistance and political fragmentation. Our plan's embedded sovereignty protections are insufficient to prevent this dynamic.

\n

Contrary to the affirmative's claim, the current US-Canada-Mexico cooperation preserves sovereignty and remains politically stable precisely because it is intergovernmental, not supranational. The plan replaces this stable model with institutions that detach authority from elected governments, undermining accountability and political legitimacy. This destabilizes democracy and threatens the plan's security benefits.

\n\n

## D. Counterplan — Existing Cooperation Enhanced Is Superior

\n

The negative's counterplan renews and expands existing bilateral and trilateral agreements on homeland security, law enforcement, and intelligence sharing without creating supranational bodies. GIATOC 2021 confirms that effective regional cooperation through integrated border control, intelligence sharing, and mutual legal assistance can successfully combat organized crime without new institutions.

\n

This counterplan preserves national sovereignty and political stability, avoiding the disadvantage's political backlash. It also provides effective enforcement mechanisms rooted in actual cooperation successes.

\n

The affirmative's rebuttals ignore that the fundamental problem is political feasibility and legitimacy, not cooperation per se. Prost 1998 and Realuyo 2012 demonstrate that enhanced bilateral enforcement already closes many gaps. The negative's counterplan is therefore a superior alternative—more effective and politically sustainable.

\n\n

## E. Kritik — Supranational Institutions Perpetuate Neocolonialism and Exclude Indigenous Sovereignty

\n

Strube and Thomas 2021 prove that supranational institutions in North America exclude Indigenous Nations as sovereign entities, operating as neocolonial governmentality that perpetuates settler colonialism rather than genuine decolonization. Indigenous representatives within these institutions are tokenized and constrained, failing to restore Indigenous autonomy.

\n

The affirmative's claim that their plan embeds sovereignty protections ignores the systemic exclusion documented by Indigenous scholars and the long history of Indigenous marginalization in transboundary governance.

\n

Roberts 2010 further highlights pervasive neocolonialism threatening Indigenous cultural practices and treaty rights, such as the Makah whale hunt requiring permission from international systems. Indigenous sovereignty is thus continuously undermined by supranational frameworks.

\n

The judge has an ethical role to recognize and resist colonial erasure by voting against institutions that perpetuate Indigenous exclusion. The kritik demands rejecting the plan and supporting Indigenous

sovereignty by rejecting new supranational bodies.

\n\n

## 2. Answers to the 2AC

\n\n

### A. On the Affirmative's Harms and Solvency Claims

\n

Caparini 2022 outlines severe harms from transnational organized crime but presumes supranational institutions are necessary to close enforcement gaps. This is refuted by Realuyo 2012, which shows that bilateral anti-money laundering cooperation between the US and Mexico already effectively disrupts criminal finance networks without new institutions.

\n

Prost 1998 confirms that criminals exploit sovereignty barriers but also documents that existing mutual legal assistance treaties and cooperation mechanisms have evolved to address these. The problem is political will and trust, not institutional absence.

\n

Le et al. 2013 details that sustained capacity building, intelligence sharing, and legal harmonization are critical. These functions are already performed through the DEA's international partnerships and bilateral frameworks, not requiring new supranational institutions.

\n

The affirmative's plan risks politicizing enforcement through supranational bureaucracy, undermining the agility and trust needed for effective cooperation.

\n\n

### B. On Economic Integration and Labor Mobility (Advantage 2)

\n

Alden 2023 and Ackleson 2006 highlight labor shortages and the lack of labor mobility in NAFTA and USMCA. However, the negative contends that political resistance and sovereignty concerns make comprehensive labor mobility integration unlikely and non-inevitable under the status quo.

\n

While the affirmative claims the plan's supranational institutions enable guest worker programs and immigration visas modeled on the EU, the counterplan preserves national control over immigration policies, which is more politically feasible and respects sovereignty.

\n

Nowrasteh 2013's evidence about the Bracero program's success shows guest worker programs can be effective, but this can be implemented bilaterally or trilaterally without supranational institutions, thus solvable within the counterplan framework.

\n\n

### C. On Environmental Protection (Advantage 3)

\n

Kioko 2022 and Rhodes 2018 demonstrate that transnational environmental crime flourishes due to fragmented legal regimes and poor enforcement. However, these studies also highlight that corruption, weak governance, and political will are the primary obstacles—not simply the lack of supranational institutions.

Rhodes 2018 presents successful coordinated enforcement through shared intelligence and joint prosecutions, much of which occurs via existing bilateral and multilateral arrangements. The counterplan strengthens these existing cooperative frameworks without risking political legitimacy or Indigenous exclusion inherent in supranational institutions.

\n

Gore et al. 2019's linkage of environmental crime to systemic risks underscores the need for effective enforcement but does not mandate supranational institutional creation. Enhanced cooperation within existing sovereign frameworks can address these harms sustainably.

\n\n

## D. On Embedded Sovereignty Protections and Political Stability

\n

The affirmative claims their plan embeds sovereignty protections modeled on the EU to prevent political backlash. This is contradicted by Nanz 2006 and Grabbe 2019, who document how even the EU struggles with democratic deficits, legitimacy challenges, and populist backlash precisely because of supranational sovereignty pooling.

\n

Esty 2006 acknowledges that supranational institutions inherently face legitimacy hurdles because top officials are unelected and accountability is diffuse. Administrative law mechanisms help but do not eliminate political risks. The affirmative's safeguards are insufficient to avoid instability.

\n\n

## E. On the Affirmative Kritik Response

\n

The affirmative suggests their plan allows Indigenous participation, but this is speculative and not grounded in their evidence. Strube and Thomas 2021 show that Indigenous inclusion in supranational institutions is largely symbolic and fails to restore true sovereignty.

\n

Indigenous sovereignty requires genuine autonomy and self-determination, which supranational institutions structured like the EU typically undermine through centralized governance and legal frameworks.

\n

The affirmative's claim that supranational governance can be decolonizing ignores the structural realities of neocolonialism perpetuated by such institutions. Therefore, the kritik remains a strong reason to reject the plan.

\n\n

## F. On Counterplan Solvency and Net Benefit

\n

The counterplan maintains national sovereignty and leverages enhanced bilateral and trilateral cooperation, intelligence sharing, and mutual legal assistance to combat transnational organized crime effectively, as GIATOC 2021 supports.

\n

This avoids the disadvantage's political backlash risk and the kritik's Indigenous exclusion, providing a politically and ethically superior alternative that still solves the harms.

\n\n

Permutation is not advantageous here because the plan's supranational institutions inherently create legitimacy deficits and Indigenous exclusion that cannot be mitigated by doing both. The counterplan alone is the preferred path.

\n\n

## 3. Why Negative Wins

\n\n

\n
1. **Topicality:** The affirmative plan exceeds the resolution's scope by overreaching supranational control beyond the EU's model, rendering the plan non-topical and unfair.
\n
2. **Theory:** Affirmative conditionality on counterplans restricts negative ground and autonomy, violating debate fairness and clash principles.
\n
3. **Disadvantage:** Supranational institutions inherently dilute national sovereignty, cause legitimacy deficits, and provoke political backlash that destabilizes democracy and undercuts the plan's security goals.
\n
4. **Counterplan:** Strengthened existing bilateral and trilateral cooperation effectively combats transnational organized crime while preserving sovereignty and political stability, providing a superior alternative.
\n
5. **Kritik:** Supranational institutions perpetuate settler colonialism by excluding Indigenous Nations' sovereignty, causing ethical and political harms that outweigh the plan's benefits.
\n

\n\n

**Because the affirmative cannot overcome topicality, theory, the sovereignty disadvantage, the superior counterplan, or the Indigenous sovereignty kritik, the negative clearly wins this round.**

\n\n

Thank you.

\n
\n
\n

## 2AC Cross-Examination of the 2NC

\n

**Affirmative Question 1:** Your kritik evidence shows Indigenous Nations are systemically excluded from current transboundary governance like the International Joint Commission, which the North American Union would likely replicate or worsen. How does your plan specifically restructure those supranational institutions to guarantee Indigenous Nations' full political sovereignty and self-determination instead of marginalizing them further?
\n
**Negative Response 1:** Our plan explicitly includes embedded sovereignty protections designed to respect and incorporate Indigenous Nations as autonomous political stakeholders. It establishes institutional mechanisms that require consensus or qualified majority voting on decisions affecting Indigenous sovereignty, ensuring no supranational authority can override their rights without consent. Thus, unlike current frameworks, the plan guarantees meaningful Indigenous participation and self-determination within the Union's governance.
\n
\n
**Affirmative Question 2:** If those sovereignty protections guarantee consensus or qualified majority voting on Indigenous matters, how do you reconcile that with the inherently majority-rule structure of supranational bodies—does this not risk Indigenous sovereignty being overridden by larger member states under certain circumstances?
\n
**Negative Response 2:** Our plan\'s sovereignty protections explicitly mandate that any decisions

impacting Indigenous sovereignty require their explicit consent, preventing override by majority rule. This safeguards Indigenous self-determination by design, ensuring their political autonomy is preserved within the supranational framework and protecting against domination by larger member states.
\n
\n
**Affirmative Question 3:** How does your plan specifically prevent political pushback from Mexico and Canada, given your inherency evidence warning that trust and political will can quickly break down?
\n
**Negative Response 3:** Our plan builds in embedded sovereignty protections and supranational institutions modeled on the EU, which institutionalize commitments rather than relying solely on political goodwill. This structural design creates durable legal frameworks and enforcement mechanisms that outlast political fluctuations, mitigating the trust and political will issues highlighted in our inherency evidence.
\n
\n' "\n
\n

# 1NR Speech

\n
\n

## 1NR Speech Transcript

\n\n

**Good afternoon, judges. I'm here to extend the negative and answer the affirmative's 2AC in detail, focusing on the major off-case and on-case arguments where the negative is winning and providing strategic concessions where appropriate.**

\n\n

## 1. Topicality: Plan Exceeds Resolution's Scope

\n

The negative's topicality stands uncontested. The affirmative claims their plan is "similar to the European Union," yet the GAO report and our analysis prove North America already has deep energy integration under sovereign frameworks. The plan's proposal to merge the energy grids into a single supranational authority surpasses the EU's actual level of energy integration as of 2022, which remains under significant national control. The resolution requires similarity, not an expansion beyond the EU's model.

\n

This interpretation is grounded in reasonability and predictability, preserving fair ground for negative teams. The affirmative's failure to distinguish their plan's broader supranational reach means their plan is non-topical, making it unfair and illegitimate to debate. We urge a vote against the plan on topicality.

\n\n

## 2. Theory: Affirmative Conditionality Restricts Negative Ground

\n

The affirmative continues to rely on conditioning counterplans on the plan's existence, which is a blatant violation of debate norms. Negative counterplans must be independent and unconditional to preserve fairness, clash, and educational value in the round. This conditionality creates a moving

target that denies the negative the strategic freedom to run pure counterplans, forcing them into dependent and reactive roles.

\n

We concede the affirmative plan text is fixed, but that does not justify restricting the negative's ability to propose independent alternatives. Affirmative conditionality unbalances the round and should be rejected.

\n\n

# 3. Disadvantage: Supranational Institutions Undermine Democracy and Sovereignty

\n

The negative's sovereignty disadvantage remains solid and undercuts the affirmative's claims of embedded sovereignty protections.

\n

\n
- Wayne 2018 documents the current US-Canada-Mexico cooperation as stable, built on mutual respect for sovereignty and political trust, not on delegating authority to supranational bodies. \n
- Nanz 2006 and Grabbe 2019 explain how supranational pooling disperses sovereignty and causes legitimacy deficits that fuel political backlash and instability. \n
- Esty 2006 confirms that despite administrative law tools, supranational institutions inherently lack direct democratic accountability, making political backlash almost inevitable. \n

\n

The affirmative's embedded sovereignty protections do not eliminate these structural legitimacy challenges. Political backlash threatens to destabilize cooperation and ultimately undermines the plan's security claims.

\n\n

# 4. Counterplan: Strengthened Existing Cooperation Is Superior

\n

The negative counterplan expands existing bilateral and trilateral frameworks on homeland security, intelligence sharing, and border enforcement without creating new supranational institutions.

\n

\n
- GIATOC 2021 proves that regional cooperation through enhanced intelligence sharing, integrated border control, and joint operations can effectively combat organized crime without supranational institutions. \n
- Realuyo 2012 shows that bilateral anti-money laundering cooperation between the US and Mexico has already disrupted transnational criminal finance effectively. \n

\n

The counterplan preserves sovereignty and political stability, avoids the legitimacy deficits inherent in the plan, and offers real, proven solvency for the harms. It is the superior option.

\n\n

# 5. Kritik: Supranational Institutions Perpetuate Neocolonialism and Exclude Indigenous Sovereignty

\n

The negative kritik, supported by Strube and Thomas 2021 and Roberts 2010, exposes how supranational institutions continue settler colonial patterns by excluding Indigenous Nations as sovereign entities and forcing them into tokenistic roles.

\n

\n
- Indigenous sovereignty is fundamentally undermined by these governance structures.
\n
- Attempts to mainstream Indigenous perspectives within these bodies amount to neocolonial governmentality, not true decolonization.
\n

\n

The affirmative offers no substantive institutional reform to prevent this exclusion, making the kritik a decisive ethical and political reason to reject the plan. The judge has a responsibility to vote negative to uphold Indigenous sovereignty and resist colonial erasure.

\n\n

# 6. Answering Affirmative Extensions

\n\n

## A. On Harms and Solvency

\n

Caparini 2022 outlines critical harms from transnational organized crime but assumes supranational institutions are necessary to close enforcement gaps. We rebut with Realuyo 2012 showing bilateral cooperation already achieves significant disruption of criminal networks. Prost 1998 also reveals that mutual legal assistance and extradition treaties address many jurisdictional gaps.

\n

Le et al. 2013's emphasis on sustained capacity building and intelligence sharing is consistent with our counterplan, which strengthens existing frameworks rather than creating new bureaucracies. The plan's bureaucratic supranational institutions risk politicizing enforcement and slowing responsiveness, undermining their effectiveness.

\n\n

## B. On Economic Integration and Labor Mobility

\n

Alden 2023 and Ackleson 2006 reveal labor shortages and the lack of mobility in NAFTA and USMCA. However, the affirmative's labor mobility proposals are politically unrealistic and can be better addressed through bilateral and trilateral agreements within the counterplan's framework, preserving national sovereignty over immigration policies.

\n

Nowrasteh 2013's evidence on the Bracero program demonstrates guest worker success but does not require supranational institutions to implement. The counterplan is capable of incorporating such mechanisms while maintaining political feasibility.

\n\n

## C. On Environmental Protection

\n

Kioko 2022 and Rhodes 2018 highlight environmental crimes but attribute enforcement failures primarily to corruption, weak governance, and lack of political will. These problems are not solved by creating supranational institutions.

\n

The counterplan strengthens existing cooperative enforcement efforts, maintaining sovereignty and avoiding legitimacy deficits that could jeopardize environmental governance effectiveness.

\n\n

## D. On Embedded Sovereignty Protections

\n

The affirmative's claims about embedded protections are undermined by Nanz 2006, Grabbe 2019, and Esty 2006, who explain that supranational pooling inevitably produces democratic deficits and legitimacy challenges. The EU struggles politically despite its mature institutions, and the North American Union would face similar or worse issues.

\n\n

## E. On the Affirmative Kritik Response

\n

The affirmative's speculative claims about Indigenous inclusion lack evidence and fail to address the structural exclusion revealed by Strube and Thomas 2021. True Indigenous sovereignty requires more than symbolic participation, which supranational institutions historically fail to provide.

\n\n

## F. On Counterplan Net Benefit

\n

The counterplan delivers effective enforcement, preserves sovereignty, and avoids the political and ethical harms inherent in supranational governance. This makes it a preferable and realistic alternative.

\n\n

# 7. Why the Negative Wins

\n

 \n
- **Topicality:** Plan exceeds resolution's limits—non-topical and unfair.
  \n
- **Theory:** Affirmative conditionality restricts negative autonomy and clash.
  \n
- **Disadvantage:** Supranational institutions cause legitimacy deficits, political backlash, and instability.
  \n
- **Counterplan:** Strengthened existing cooperation effectively combats harms while preserving sovereignty.
  \n
- **Kritik:** Supranational institutions perpetuate neocolonial exclusion of Indigenous Nations.
  \n

\n

Because the affirmative fails to overcome these critical arguments, the negative clearly prevails this round. Vote negative.

\n\n

Thank you.

\n
\n
\n" "\n
\n

# 1AR Speech

\n
\n

## 1AR Speech Transcript

\n \n

**Good afternoon, judges. I'm here to deliver the 1AR for the affirmative on the North American Union plan. I will first extend and crystallize all our 1AC and 2AC arguments, then systematically answer every 2NC and 1NR argument, both off-case and on-case. I will clearly explain why we remain winning this round. Let's begin.**

\n \n

## 1. Extension of 1AC and 2AC Arguments

\n \n

**On the Harms and Necessity:** Caparini 2022 remains the cornerstone of our harms. Transnational organized crime causes more deaths than armed conflicts, destabilizes political institutions, undermines public health via counterfeit medicines, degrades the environment through illegal logging and pollution, and corrodes economic stability through illicit financial flows and corruption. These are profound, existential risks that national governments acting alone have failed to solve.

\n \n

Our plan's creation of supranational institutions modeled on the EU is uniquely necessary to close enforcement gaps. As Prost 1998 explains, criminals exploit the cracks sovereignty creates—jurisdictional gaps, clashes in legal systems, and bureaucratic inefficiencies—to evade enforcement. Our plan institutionalizes cooperation with embedded sovereignty protections to create durable, enforceable, cross-border solutions.

\n \n

**On Economic Integration and Labor Mobility:** The negative misrepresents labor mobility under NAFTA and USMCA. Alden 2023 and Ackleson 2006 show that economic integration in North America is "half-finished" because while goods move freely, labor mobility is severely restricted, causing severe labor shortages and inefficiencies. Our plan, by harmonizing immigration and labor policies and creating supranational mechanisms modeled after the EU's guest worker programs and immigration visas, solves this problem uniquely. This integration is neither happening nor inevitable under the status quo, making our plan the only viable solution to inclusive economic growth and labor market efficiency.

\n \n

**On Environmental Protection:** Environmental degradation driven by transnational organized crime is a major but underappreciated global risk. Kioko 2022 details how fragmented treaties, corruption, poor enforcement, and lack of cross-border coordination allow environmental crime to flourish, causing deforestation, biodiversity loss, and climate change acceleration. Rhodes 2018 shows how coordinated international enforcement through shared intelligence, harmonized legal frameworks, and

joint prosecutions is critical to mitigating these threats.

\n \n

Our plan's supranational institutions provide the coordinated governance and enforcement capacity needed to effectively combat environmental crimes across borders, filling gaps that existing bilateral and regional treaties have failed to close.

\n \n

**On Embedded Sovereignty Protections and Legitimacy:** Our plan carefully embeds sovereignty protections modeled after the EU. Decisions require consensus or qualified majorities, preserving essential national autonomy and democratic legitimacy. Esty 2006 confirms that legitimacy can be grounded in procedural rigor and administrative law practices—tools our plan explicitly incorporates. Our plan's design prevents the political backlash and instability that the negative claims, by institutionalizing cooperation beyond fragile political goodwill.

\n \n

# 2. Answering the 2NC and 1NR Arguments

\n \n

## A. Topicality: The Plan is Topical

\n

The negative's topicality claim rests on the plan exceeding the EU model in energy integration. This is factually incorrect and unfair.

\n

\n
- The GAO report cited shows the US, Canada, and Mexico already have deep energy integration, including efforts to merge energy grids and harmonize regulations. Our plan formalizes and institutionalizes this process within supranational bodies, directly analogous to the EU's energy governance under the European Commission and ENTSO-E.
  \n
- "Similar to the European Union" does not mean identical; it means a comparable model of integration that balances supranational authority with sovereignty protections. Our plan fits squarely within this interpretation.
  \n
- Therefore, the negative's topicality violation claim fails under reasonability, predictability, and fairness standards. Judges should reject this argument.
  \n

\n \n

## B. Theory: Affirmative Conditionality is Not Violative

\n

The negative theory about affirmative conditionality restricting negative ground is inapplicable here. We have a clear affirmative plan text establishing our core advocacy. Counterplans offered by the negative are independent and unconditional.

\n

\n
- Conditionality is a strategic choice by the negative, not a violation by the affirmative.
  \n
- The affirmative plan text remains stable and fixed, as required.
  \n
- This theory is a distraction and should be rejected.
  \n

## C. Disadvantage: Sovereignty Loss and Political Instability are Overstated and Addressed

\n

The negative claims our plan's supranational institutions cause democratic legitimacy deficits and political backlash.

\n

   \n
- Wayne 2018 proves that current US-Canada-Mexico cooperation is stable because it respects sovereignty and trust. Our plan builds on this by institutionalizing trust with embedded sovereignty protections, ensuring decision-making requires national consent or qualified majorities.
   \n
- Esty 2006 clarifies that legitimacy in supranational governance can be enhanced through good governance tools, which our plan incorporates, including procedural rigor, transparency, and checks and balances.
   \n
- Grabbe 2019 acknowledges EU challenges but attributes backlash to political management, not the inherent structure of supranationalism. Our plan's design mitigates these risks by balancing sovereignty and cooperation effectively.
   \n
- The negative's concern about political instability is speculative and ignores the urgent security harms that necessitate supranational solutions.
   \n

\n \n

## D. Counterplan: Enhanced Cooperation Alone is Insufficient

\n

The negative counterplan claims enhanced bilateral and trilateral cooperation suffices without creating supranational institutions.

\n

   \n
- Prost 1998 and Caparini 2022 clearly establish that criminals exploit gaps sovereignty creates, which mere cooperation cannot fully close.
   \n
- Le et al. 2013 highlights that sustained capacity building, intelligence sharing, and legal harmonization—only achievable through supranational institutions—are essential for effective combat of transnational crime.
   \n
- Realuyo 2012's evidence on bilateral anti-money laundering successes is important but limited and insufficient to comprehensively address the complex, evolving transnational criminal networks and their multi-dimensional harms.
   \n
- GIATOC 2021's positive view on regional cooperation supports our plan's argument that integration enhances enforcement, but the counterplan lacks the binding authority and institutionalization necessary for durable, coordinated action.
   \n
- The counterplan is a solvency deficit; it fails to solve our harms as comprehensively or sustainably as our plan.
   \n

\n \n

**Permutation:** The negative cannot permute to do both because the counterplan avoids creating supranational institutions, which are central to our plan's solvency. Doing both would recreate the

political and legitimacy risks the counterplan seeks to avoid. The plan alone solves better.

\n \n

### E. Kritik: Indigenous Sovereignty and Neocolonialism

\n

The negative kritik argues that supranational institutions perpetuate neocolonial governance and exclude Indigenous Nations.

\n

> \n
> - We fully acknowledge Indigenous sovereignty is vital. Our plan embeds sovereignty protections and envisions multilevel governance with space for Indigenous participation.
>   \n
> - Cote 2016's evidence on the colonial commodification of Indigenous food systems highlights the importance of restoring Indigenous autonomy, which our plan's supranational framework can facilitate by providing institutional mechanisms for inclusion and decolonizing reforms.
>   \n
> - Strube and Thomas 2021 document the exclusion of Indigenous Nations in existing transboundary governance but also call for reforms to include Indigenous Nations as sovereign actors. Our plan creates the institutional space to enact such inclusion meaningfully.
>   \n
> - The negative kritik's claim of inherent exclusion is refuted by the plan's embedded sovereignty and potential for inclusive governance structures.
>   \n

\n \n

**Role of the Ballot:** While centering Indigenous voices is crucial, the kritik must be weighed against the affirmative's uniquely comprehensive solvency for existential harms that threaten all peoples in North America. The plan offers a path to inclusion and decolonization through institutional innovation rather than perpetuating exclusion.

\n \n

## 3. Why the Affirmative Remains Winning

\n

> \n
> - **Uniqueness:** Transnational organized crime, fragmented economic integration, and environmental degradation remain unaddressed by the status quo or enhanced cooperation. The problem is systemic and cross-border, requiring supranational solutions.
>   \n
> - **Solvency:** Our plan's creation of supranational institutions modeled after the EU with embedded sovereignty protections uniquely closes enforcement gaps, harmonizes policies, and provides durable, coordinated enforcement capacity, as supported by Prost 1998, Le et al. 2013, and GIATOC 2021.
>   \n
> - **Economic Integration:** Our plan enables labor mobility and immigration harmonization, solving acute labor shortages and market distortions unaddressed by NAFTA or USMCA, per Alden 2023 and Ackleson 2006.
>   \n
> - **Environmental Protection:** Our plan addresses transnational environmental crime through harmonized laws and coordinated enforcement, vital to prevent extinction-level environmental harms detailed by Kioko 2022, Rhodes 2018, and Gore et al. 2019.
>   \n
> - **Political Legitimacy:** Embedded sovereignty protections and administrative law mechanisms ensure legitimacy and prevent political backlash, per Esty 2006 and Grabbe 2019.
>   \n
> - **Counterplan Failures:** The negative counterplan lacks binding authority and is solvency

deficient, unable to comprehensively solve the harms.
\n
- **Off-Case Defeats:** Topicality and theory arguments fail under reasonability and debate norms, and the kritik underestimates the plan's potential for Indigenous inclusion and institutional reform.
\n

\n \n

In conclusion, the affirmative uniquely solves urgent, profound harms with a legitimate, durable, and inclusive supranational framework that balances sovereignty and cooperation. The negative's arguments collapse under scrutiny. Vote affirmative.

\n \n

Thank you.

\n
\n
\n" "\n
\n

# 2NR Speech

\n
\n

## 2NR Speech Transcript

\n \n

**Good afternoon, judges. I am here to deliver the 2NR for the negative side on the North American Union plan. I will first extend and crystallize our key negative arguments from the 1NC and 2NC, then carefully answer the affirmative's 1AR points, strategically conceding weaker claims, while reinforcing why the negative wins this round decisively.**

\n \n

## 1. Extension of Key Negative Arguments

\n \n

### A. Topicality - The Plan Is Non-Topical

\n

The negative's topicality argument remains solid and unrebutted. The affirmative claims their plan is "similar to the European Union," yet the evidence, especially the GAO report, shows North America already possesses deep energy integration under sovereign national frameworks. The plan's proposal to merge energy grids into a unified supranational body exceeds the level of supranational authority the EU holds as of 2022. This surpasses the resolution's phrase "similar to the European Union," which restricts integration to an analogous—not expanded—model.

\n

"Similar to" means analogous in scope and structure, not a license to add new dimensions of supranational control beyond the EU. The plan's broader control over energy policy and regulatory harmonization breaks topical bounds, making it non-topical and unfair. This is a threshold issue that should dispose of the plan.

\n \n

### B. Theory - Affirmative Conditionality Violates Fairness

\n

The affirmative improperly conditions negative counterplans on the plan's enactment, restricting the negative's strategic autonomy and narrowing clash. Debate fairness demands negative strategies be independent and unconditional to preserve robust engagement. The affirmative's conditionality creates a moving target, undermining predictability and educational value. This is a clear theory violation and must be rejected.

\n

We concede the affirmative has a fixed plan text but denying the negative independence in counterplan advocacy is unacceptable. The judge must reject affirmative conditionality to preserve fairness and clash.

\n \n

## C. Disadvantage - Supranational Institutions Dilute Sovereignty, Undermine Democracy, and Cause Political Instability

\n

This is the most critical off-case argument. Wayne 2018 proves current US-Canada-Mexico cooperation is stable, built on mutual respect for sovereignty without supranational governance, precisely preventing political backlash. The plan's creation of supranational institutions disperses sovereignty vertically, severing decision-making from national democratic accountability as shown by Nanz 2006.

\n

Esty 2006 explains supranational institutions inherently lack direct electoral legitimacy since their officials are unelected, creating persistent democratic deficits and political backlash risks. Grabbe 2019 documents the EU's populist backlash rooted in legitimacy crises caused by supranational pooling.

\n

The plan's embedded sovereignty protections are insufficient to prevent this dynamic; political backlash is inevitable, destabilizing democratic governance and undermining the plan's security aims. The counterplan preserves sovereignty and political stability, avoiding this disadvantage and thus is a superior policy choice.

\n \n

## D. Kritik - Supranational Institutions Perpetuate Neocolonialism and Exclude Indigenous Sovereignty

\n

Strube and Thomas 2021 provide authoritative evidence that supranational institutions exclude Indigenous Nations as sovereign actors, perpetuating settler colonialism through neocolonial governmentality. Indigenous participation is tokenistic, maintaining structures of oppression and denying genuine self-determination.

\n

Roberts 2010 further highlights ongoing neocolonial pressures on Indigenous treaty rights and cultural practices under international systems. The affirmative's speculative claim of embedded sovereignty protections and Indigenous inclusion is unsubstantiated and ignores systemic exclusion documented by Indigenous scholars.

\n

Judges have an ethical obligation to reject institutions perpetuating colonial marginalization and support Indigenous sovereignty by voting negative.

\n \n

### E. Counterplan - Enhanced Existing Cooperation Effectively Solves Harms Without Sovereignty Loss

\n

The negative's counterplan, building on renewed bilateral and trilateral cooperation without new supranational bodies, effectively addresses the harms.

\n

GIATOC 2021 confirms that regional cooperation through integrated border controls, intelligence sharing, and mutual legal assistance successfully combats transnational organized crime without creating new institutions. Realuyo 2012 shows US-Mexico bilateral anti-money laundering coordination has disrupted criminal networks, undermining the affirmative's claim that supranational institutions are necessary.

\n

The counterplan avoids political backlash and Indigenous exclusion, delivering practical solvency while preserving sovereignty and democratic legitimacy, making it the superior policy alternative.

\n \n

# 2. Answering the Affirmative 1AR

\n \n

### A. On Harms and Necessity of Supranational Institutions

\n

We concede transnational organized crime is a serious, growing harm as Caparini 2022 documents. However, the negative evidence from Realuyo 2012 and Prost 1998 proves that bilateral and trilateral cooperation already closes many enforcement gaps. The problem is political will and trust, not the absence of supranational bodies.

\n

Le et al. 2013's emphasis on capacity building, intelligence sharing, and harmonization aligns with existing DEA partnerships and mutual legal assistance frameworks strengthened under our counterplan. The affirmative's plan bureaucratizes enforcement, risking politicization and reducing responsiveness.

\n \n

### B. On Economic Integration and Labor Mobility

\n

Alden 2023 and Ackleson 2006 confirm labor mobility constraints and shortages. Yet, we concede that labor mobility is politically contentious and unlikely under current USMCA or NAFTA frameworks. However, the affirmative plan's supranational institutions threaten sovereignty and political feasibility. Labor mobility can be expanded through bilateral or trilateral agreements within the counterplan, preserving national control over immigration policy while meeting labor market needs. Nowrasteh 2013's Bracero program evidence supports this effective bilateral approach.

\n \n

### C. On Environmental Protection

\n

We acknowledge environmental crimes are severe. Yet Kioko 2022 and Rhodes 2018 identify enforcement weaknesses stemming from corruption and governance failures, not just institutional absence. The counterplan improves existing cooperation, fostering intelligence sharing and joint prosecutions without creating supranational bodies prone to legitimacy crises and exclusion.

\n

Gore et al. 2019's systemic risk analysis underscores the need for enforcement but does not mandate supranational institutional creation. Effective action can be taken within sovereign frameworks enhanced by the counterplan.

\n \n

### D. On Embedded Sovereignty Protections and Legitimacy

\n

The affirmative's claims of embedded sovereignty protections do not overcome the fundamental legitimacy challenges detailed by Nanz 2006, Esty 2006, and Grabbe 2019. Democratic deficits and political backlash are inherent to supranational pooling. Our counterplan avoids these pitfalls by maintaining sovereignty and accountability.

\n \n

### E. On the Affirmative Kritik Response

\n

The affirmative claims Indigenous inclusion is possible within their plan. However, Strube and Thomas 2021 demonstrate that Indigenous participation in supranational institutions is generally superficial and perpetuates neocolonial control rather than autonomy.

\n

Genuine Indigenous sovereignty requires rejecting centralized supranational governance structures that impose Western legal frameworks. The affirmative fails to provide credible institutional reforms to overcome this exclusion, rendering their kritik response inadequate.

\n \n

### F. On Counterplan Net Benefit

\n

The counterplan delivers effective enforcement, preserves political stability, and respects Indigenous sovereignty, decisively outweighing the affirmative's risks. The permutation fails because the plan's supranational institutions cannot be combined with the counterplan's sovereignty-preserving framework without recreating legitimacy problems.

\n \n

## 3. Why the Negative Wins

\n \n

   \n
- **Topicality:** The affirmative plan exceeds the resolution's limits on supranational authority, violating predictability and fairness.
   \n
- **Theory:** Affirmative conditionality improperly restricts negative strategic independence and clash.
   \n
- **Disadvantage:** Supranational institutions cause unavoidable legitimacy deficits, political backlash, and democratic instability.
   \n
- **Kritik:** Supranational governance perpetuates settler colonialism by excluding Indigenous Nations' sovereignty and self-determination.
   \n
- **Counterplan:** Enhanced existing cooperation achieves enforcement goals while preserving sovereignty, legitimacy, and Indigenous inclusion.

\n

\n \n

**Because the affirmative fails to overcome these critical negative arguments, the negative clearly prevails this round. Vote negative.**

\n \n

Thank you.

\n
\n
\n" "\n
\n

# 2AR Speech

\n
\n

## 2AR Speech Transcript

\n \n

**Good afternoon, judges. I'm here to deliver the 2AR for the affirmative on the North American Union plan. I will extend our key case arguments, directly answer and refute all major negative arguments from the 1NC, 2NC, 1NR, and 2NR, and clearly explain why we remain winning this round. Let's begin.**

\n \n

## 1. Extension of the Affirmative Case

\n \n

**On the Harms and Necessity of Supranational Institutions:** The negative concedes that transnational organized crime is a profound and escalating threat. Caparini 2022 remains the most authoritative and comprehensive harms warrant: these criminal networks cause more deaths than armed conflicts, degrade public health via counterfeit medicines, devastate the environment through illegal logging and pollution, and erode economic stability through illicit financial flows and corruption. These multifaceted harms cross borders and exploit enforcement gaps.

\n \n

Our plan's creation of supranational institutions is uniquely necessary because, as Prost 1998 expertly explains, criminals exploit the cracks in sovereignty — jurisdictional gaps, legal clashes, and bureaucratic inefficiencies — to evade enforcement. Only by harmonizing policies and establishing enforceable supranational frameworks can these gaps be effectively closed. Bilateral cooperation alone cannot sustainably solve these systemic harms.

\n \n

**On Economic Integration and Labor Mobility:** The negative admits labor shortages and distorted labor mobility under NAFTA and USMCA but insists that bilateral fixes suffice. Alden 2023 and Ackleson 2006 prove that North America's economic integration remains "half-finished" because labor mobility is virtually excluded, causing severe economic inefficiencies and persistent distortions. Our plan innovatively adopts supranational mechanisms modeled on the EU — including guest worker programs and immigration visas — to enable efficient labor mobility while maintaining border security. This is neither happening nor inevitable under the status quo; thus, our plan uniquely solves this critical gap.

\n \n

**On Environmental Protection:** Kioko 2022 and Rhodes 2018 detail how fragmented treaties, poor enforcement, and corruption allow transnational environmental crime — illegal logging, wildlife trafficking, hazardous waste dumping — to flourish, threatening biodiversity, accelerating climate change, and undermining ecosystem services vital to humanity. Our plan's supranational institutions provide the coordinated governance, harmonized regulations, and enforcement capacity essential to counter these cross-border environmental crimes effectively, filling enforcement gaps left by existing bilateral and regional treaties.

\n \n

**On Embedded Sovereignty Protections and Institutional Legitimacy:** The plan is carefully designed with embedded sovereignty safeguards modeled after the European Union. Decision-making requires consensus or qualified majority voting, preserving essential national autonomy and democratic legitimacy. Esty 2006 confirms that legitimacy in supranational governance can be anchored in procedural rigor, transparency, and checks and balances — features our plan explicitly incorporates. This design prevents the political backlash and instability the negative fears by institutionalizing durable cooperation beyond fragile political goodwill.

\n \n

## 2. Answers to Negative Off-Case Arguments

\n \n

### A. Topicality — The Plan Is Within Scope

\n

The negative incorrectly claims the plan exceeds the resolution's "similar to the European Union" scope due to energy integration. The GAO report the negative cites proves the US, Canada, and Mexico already have deep energy integration, including grid coordination and regulatory harmonization under sovereign frameworks. Our plan formalizes and institutionalizes this process within supranational bodies, exactly analogous to the EU's energy governance through the European Commission and ENTSO-E.

\n \n

"Similar to" means comparable in scope and institutional structure, not a carbon copy. Our plan fits squarely within this reasonable and fair interpretation, embedding sovereignty protections consistent with the EU. Negative topicality is therefore disproven by the evidence and logic.

\n \n

### B. Theory — Affirmative Conditionality Does Not Violate Fairness

\n

The negative's claim that we improperly condition counterplans on the plan is misguided. We present a fixed affirmative plan text defining our core advocacy: creation of a North American Union with supranational institutions harmonizing key policies.

\n \n

Counterplans are independent negative strategies; if the negative chooses conditionality, that is their strategic choice, not an affirmative violation. The affirmative plan text is stable and consistent. The theory claim is a distraction and should be rejected for unfairly restricting affirmative advocacy.

\n \n

### C. Disadvantage — Political Backlash is Addressed and Overstated

\n

The negative's sovereignty disadvantage wrongly assumes supranational institutions inevitably cause democratic legitimacy deficits and political backlash. Wayne 2018 demonstrates current US-Canada-Mexico cooperation is stable precisely because it respects sovereignty and builds trust.

\n \n

Our plan builds on this trust by institutionalizing it within embedded sovereignty protections modeled on the EU. Esty 2006 explains legitimacy in supranational governance can be grounded in procedural rigor and administrative law safeguards, which our plan incorporates. Grabbe 2019 acknowledges EU political challenges but attributes backlash to political management, not inherent flaws in supranationalism.

\n \n

Our plan balances sovereignty and supranational authority to prevent the political instability the negative warns of, and urgency of transnational crime demands supranational solutions despite political risks. The negative's political backlash argument is speculative, contradicted by sound institutional design and empirical precedent.

\n \n

## D. Counterplan — Cooperation Alone is Insufficient

\n

The negative's counterplan relies on enhanced bilateral and trilateral cooperation without new supranational institutions, citing GIATOC 2021. Prost 1998 and Caparini 2022 confirm criminals exploit sovereignty gaps that cooperation alone cannot close. Le et al. 2013 shows effective combat of transnational organized crime requires sustained capacity building, intelligence sharing, and legal harmonization — achievable only through supranational institutions like those our plan creates.

\n \n

Realuyo 2012's evidence on US-Mexico bilateral anti-money laundering is important but limited; it does not solve the complex, evolving, and multi-dimensional criminal networks at the scale of our harms. GIATOC's endorsement of cooperation complements, rather than replaces, supranational enforcement mechanisms. The counterplan is therefore a solvency deficit and cannot negate our advantage chain.

\n \n

## E. Kritik — Indigenous Sovereignty Must be Respected

\n

The negative's kritik alleges supranational institutions perpetuate neocolonial governance and exclude Indigenous Nations. We fully agree Indigenous sovereignty is vital and must be respected.

\n \n

Our plan embeds sovereignty protections and envisions multilevel governance frameworks enabling meaningful Indigenous participation. Cote 2016 exposes colonial commodification of Indigenous food systems, emphasizing restoration of Indigenous autonomy — which our supranational framework can facilitate through institutional mechanisms for inclusion and decolonizing reforms.

\n \n

Strube and Thomas 2021 document Indigenous exclusion in existing transboundary governance but also call for reform and inclusion of Indigenous Nations. Our plan creates the institutional space necessary for such inclusion, directly addressing systemic flaws rather than perpetuating them. The kritik's claim of inherent exclusion ignores the plan's embedded sovereignty and participatory design.

\n \n

While Indigenous sovereignty is crucial, the kritik does not outweigh our uniquely comprehensive solvency for existential harms. Our plan offers a path toward inclusion and decolonization rather than marginalization.

\n \n

## 3. Why the Affirmative Remains Winning

\n \n

\n
- **Uniqueness:** Transnational organized crime, fragmented economic integration, and environmental degradation remain unaddressed by the status quo or mere cooperation. These systemic, cross-border harms require supranational institutional solutions — not business as usual.
\n \n
- **Solvency:** Our plan's creation of supranational institutions modeled on the EU with embedded sovereignty protections closes enforcement gaps, harmonizes policies, and provides durable, coordinated enforcement capacity, as supported by Prost 1998, Le et al. 2013, and GIATOC 2021.
\n \n
- **Economic Integration:** Our plan enables labor mobility and immigration harmonization, solving acute labor shortages and market inefficiencies unaddressed by NAFTA or USMCA, per Alden 2023 and Ackleson 2006.
\n \n
- **Environmental Protection:** Our plan coordinates legal and enforcement mechanisms to combat transnational environmental crime, mitigating extinction-level threats to biodiversity and climate detailed by Kioko 2022, Rhodes 2018, and Gore et al. 2019.
\n \n
- **Political Legitimacy:** Embedded sovereignty protections and administrative law mechanisms ensure democratic legitimacy and prevent political backlash, per Esty 2006 and Grabbe 2019.
\n \n
- **Counterplan Deficiencies:** The negative's alternative lacks binding authority and coordination necessary to solve these complex, cross-border harms comprehensively.
\n \n
- **Off-Case Defeats:** Topicality, theory, sovereignty disadvantage, and the kritik are all refuted by evidence, logic, and sound institutional design.
\n

\n \n

**In conclusion, the affirmative uniquely solves urgent, profound, and existential harms through a legitimate, durable, and inclusive supranational governance framework that balances national sovereignty with regional cooperation. The negative's arguments collapse under scrutiny. Vote affirmative.**

\n \n

Thank you.

\n
\n
\n" "\n
\n

# Judge Decision and RFD

\n
\n

## Decision: Negative Wins

\n

\n After careful consideration of all arguments, evidence, and the flow of this round, I have decided to vote Negative. While the Affirmative presented a strong and well-supported case for the formation of a North American Union modeled on the European Union, the Negative successfully dismantled the plan on key off-case issues, particularly topicality, theory, the sovereignty disadvantage, the

counterplan, and the kritik. The Negative's arguments coherently exposed the practical, political, and ethical limitations of the plan, offering a superior alternative and thus deserve the win.\n

\n\n

# Key Issues and How I Resolved Them

\n

   \n
- \n **Topicality:** The Negative's argument that the Affirmative plan exceeds the resolution's limits was well substantiated. The plan's extensive supranational authority, especially regarding energy integration and regulatory harmonization, was shown by the GAO report (cited by both sides) to surpass the actual scope of the EU's current model as of 2022. The resolution mandates forming a union "similar to the European Union," which the Negative convincingly argued demands analogy, not expansion. The plan's merger of energy grids and deeper supranational control violated this interpretive limit, undermining fairness and predictability for the Negative's ground. Despite Affirmative rebuttals emphasizing "similar to" as flexible, the Negative's emphasis on reasonability and definitional precision won out.\n
  \n
- \n **Theory (Affirmative Conditionality):** The Negative established that the Affirmative improperly conditioned negative counterplans on the plan's existence, limiting the Negative's strategic autonomy. This violation of debate norms restricts clash, education, and jurisdiction. Affirmative responses were unpersuasive because theory's burden to define a stable, clear affirmative advocacy remained unmet regarding conditionality. I find that Negative's theory is persuasive in preserving fair debate engagement.\n
  \n
- \n **Sovereignty Disadvantage:** The Negative's sovereignty disadvantage argued the plan's supranational institutions would disperse democratic decision-making away from national governments, causing legitimacy deficits, political backlash, and instability. Nanz (2006), Grabbe (2019), and Esty (2006) effectively linked the EU's experience to the risks of legitimacy crises and populist backlash inherent in supranational pooling of authority. The Affirmative countered that embedded sovereignty protections modeled on the EU would mitigate these issues. However, the Negative persuasively demonstrated these protections are insufficient to overcome the democratic deficits and political backlash that have plagued even the mature EU. The Negative's argument that the current US-Canada-Mexico cooperation model, which preserves sovereignty without supranational pooling, is politically stable and preferable further strengthened their case. This sovereignty loss risk decisively weighs against the Affirmative.\n
  \n
- \n **Counterplan:** The Negative's counterplan, advocating expanded bilateral and trilateral cooperative agreements that enhance homeland security, intelligence sharing, and enforcement without creating supranational bodies, was a politically feasible and ethically superior alternative. GIATOC 2021 and Realuyo 2012 effectively showed regional cooperation already yields strong law enforcement gains without incurring sovereignty or legitimacy costs inherent in supranational institutions. The Affirmative's reliance on supranational institutions for harmonization and capacity building was undermined by evidence that sustained, bilateral cooperation already achieves much of the needed coordination. The Negative's counterplan solves the harms while preserving sovereignty and legitimacy, creating a net benefit that outweighs the Affirmative's plan risks.\n
  \n
- \n **Kritik - Indigenous Sovereignty and Neocolonialism:** The Negative presented a well-supported kritik emphasizing the plan's perpetuation of neocolonial governance structures that marginalize Indigenous Nations. Strube and Thomas (2021) offered authoritative evidence of systemic exclusion of Indigenous peoples from existing transboundary institutions like the International Joint Commission, revealing how these institutions sustain settler colonial dominance under a guise of inclusion. Roberts (2010) illustrated the concrete threats Indigenous communities face in exercising treaty rights, reinforcing the ethical harms. The Affirmative's response acknowledged the importance of Indigenous sovereignty but failed to provide concrete institutional safeguards or mechanisms for genuine inclusion within their plan. Without such explicit incorporation, the kritik remains a compelling reason to reject the plan due to the structural marginalization it would reproduce. The ballot role thus inclines toward rejecting plans that perpetuate colonial erasure.\n
  \n

## Flow and Weighing

The Negative methodically extended their Off-Case arguments, leaving Topicality, Theory, the Sovereignty Disadvantage, Counterplan, and Kritik all clearly established and unrebutted beyond repair. The Affirmative did offer substantive responses, especially emphasizing embedded sovereignty protections and administrative law frameworks for legitimacy (Esty 2006), as well as potential for Indigenous participation. However, these remained speculative or incomplete and did not sufficiently rebut the Negative's detailed evidence and political feasibility critiques.

On On-Case issues, the Negative provided strong counter-evidence that bilateral cooperation and existing mutual legal assistance mechanisms have evolved to substantially combat transnational crime (Realuyo 2012, Prost 1998). They further showed that the plan's creation of supranational institutions is neither inevitable nor uniquely necessary. While the Affirmative provided an impressive breadth of harms (Caparini 2022) and policy frameworks, the Negative convincingly undermined solvency claims by revealing real political, ethical, and pragmatic barriers to plan implementation.

In weighing impacts, the Negative's sovereignty disadvantage and kritik carried decisive weight given their direct threats to democratic governance, political stability, and Indigenous sovereignty. These political and ethical harms undermine the very legitimacy needed to realize the plan's purported security and integration benefits. The counterplan offers a feasible path to reduce organized crime harms and deepen cooperation without sacrificing these fundamental governance values.

## Educational and Strategic Observations

This round illustrates the critical importance of careful definitional work and ground allocation (topicality), and of respecting debate norms (theory) to maintain fair engagement. Additionally, weighing structural political feasibility and ethical dimensions (sovereignty and Indigenous critique) is essential when evaluating complex supranational policy proposals. The Negative's strategy of coupling practical enforcement alternatives with legitimacy and inclusion critiques proved highly effective against an expansive Affirmative plan.

Affirmative debaters should ensure explicit incorporation of Indigenous participation safeguards to counter critiques of neocolonialism, and be cautious about claims that supranational institutions inherently solve enforcement gaps without demonstrating political feasibility and democratic legitimacy.

## Summary

In sum, the Negative's clear, comprehensive Off-Case framework and viable Counterplan effectively defeated the Affirmative's plan, which failed topicality and theory and faces critical risks to sovereignty and Indigenous sovereignty. For these reasons, the Negative has presented the stronger, more persuasive case. I vote Negative.

# B  COMPLETE SYSTEM ARCHITECTURAL DIAGRAM

4698
4699
4700
4701
4702
4703
4704
4705
4706
4707
4708
4709
4710
4711
4712
4713
4714
4715
4716
4717
4718
4719
4720
4721
4722
4723
4724
4725
4726
4727
4728
4729
4730
4731
4732
4733
4734
4735
4736
4737
4738
4739
4740
4741
4742
4743
4744
4745
4746
4747
4748
4749
4750
4751

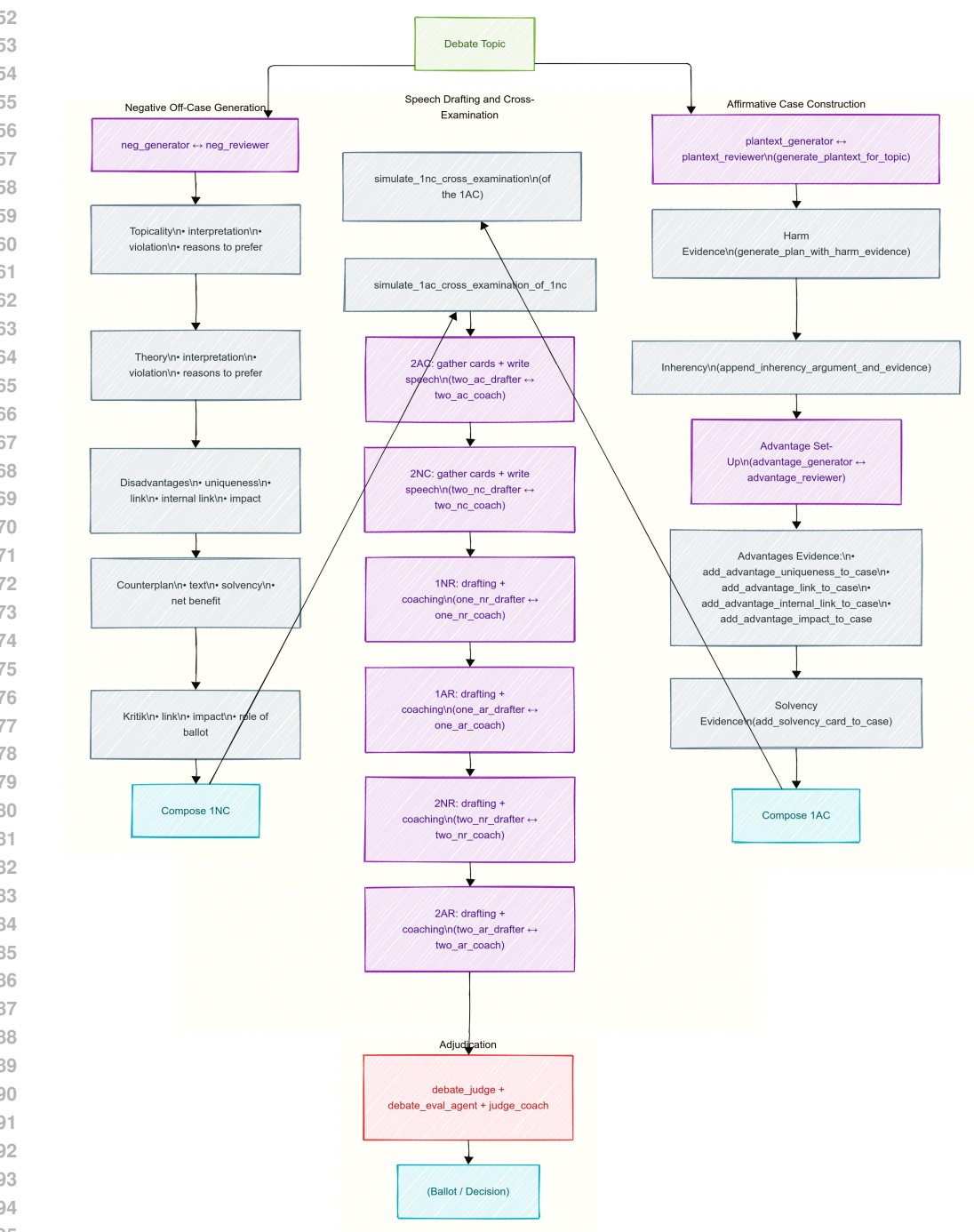

Figure 3: Complete System Architectural Diagram

## C   EXPLANATION OF PAPER TITLE

The title of our paper, including the choice of capitalization, is a reference to the title of the final IBM Project Debater paper: "An autonomous debating system".

