# OpenReview forum: "A superpersuasive autonomous policy debating system"
_ICLR.cc/2026/Conference — ICLR 2026 Conference Withdrawn Submission_

### Official Review · Reviewer_rT5i · 2025-10-30

**Soundness:** 1
**Presentation:** 2
**Contribution:** 1
**Rating:** 2
**Confidence:** 4

**Summary:**

This paper investigates argument modeling and introduces a multi-agent system for policy debate. The proposed framework follows a structured retrieval–synthesis–self-correction workflow to generate debates, leveraging an external knowledge base, OpenDebateEvidence, to provide factual grounding. Through this design, the system produces arguments that are both logically coherent and supported by evidence. The authors evaluate the framework on simulated debates assessed by human judges, focusing on argument quality, factual accuracy, and faithfulness to retrieved evidence.

**Strengths:**

- This paper proposes a multi-agent system for debate modeling with augmentation of external evidence.

**Weaknesses:**

- The paper lacks enough novelty, as its main contribution lies primarily in system construction rather than method innovation. As such, it would be more appropriate as a demonstration paper rather than a full research submission to a top-tier venue like ICLR.
- Important implementation details are missing, including the choice of the base model, prompt design, and human evaluation setup. The absence of these details significantly reduces the transparency and reproducibility of the work.
- The experimental evaluation is insufficient. The authors only report results on 20 simulated debates without comparing against baseline systems or conducting ablation studies to justify the effectiveness of each component. Moreover, the paper provides little to no analysis of the results, which further weakens the empirical support for the proposed framework;
- No statement on ethics, while argumentation or debate system may produce harmful contents.

**Questions:**

None

**Details Of Ethics Concerns:**

This paper studies argument/debate generation and the framework may produce harmful contents. Non ethic statements are included in the paper.

---

### Official Review · Reviewer_JdR4 · 2025-10-31

**Soundness:** 2
**Presentation:** 2
**Contribution:** 1
**Rating:** 2
**Confidence:** 4

**Summary:**

This paper presents an autonomous system designed to compete in full-scale, American competitive policy debates, surpassing prior efforts like IBM Project Debater, which is designed for shorter and simplified debates for lay people. The system uses a hierarchical architecture of LLM agents that collaborate through specialized workflows to perform tasks such as evidence retrieval, argument synthesis, and self-correction, leveraging the OpenDebateEvidence corpus. Demonstrated through a live, ongoing debate performance, the agents generate complete speech transcripts, engage in multi-turn exchanges, and conduct intelligent cross-examinations. Preliminary evaluations show the system consistently produces superior arguments compared to human-authored cases and wins simulated rounds judged by an autonomous adjudicator. Expert debate coaches also favor its outputs over those of human debaters.

Overall, the manuscript seems to have been written hastily. For publication, important details need to be added, the presentation should be improved, etc. as detailed in the weaknesses section below.

**Strengths:**

1. This work tackles an interesting and important topic of persuasive argument generation.
2. The experiment results seem promising.

**Weaknesses:**

1.	The manuscript is missing important details, preventing an adequate evaluation.
-	The use of multiple agents is heavily emphasized, yet they are not explained. The only hint I can find is not in the main text but in the caption for Fig 1, where it states that gpt-4-mini is used. Does that imply an “agent” is simply gpt-4-mini with a different prompt? If so, what are the prompts? How were they selected?
-	The interaction between agents in the pipeline presented in Sec 4.3 is not clearly described.
-	Given practically no information about human authors who competed against the proposed system, it is difficult to judge the significance of the experiment results.
2.	The experiments do not adequately evaluate the system nor provide useful insights.
-	Without any analysis of results, it is hard to know what strengths and weaknesses of the system are.
-	The experiments are only done against human authors, but other LLM-based baselines can show which component of the proposed pipeline are responsible for the performance. For instance, given that policy debate case has a rigid structure, an approach that uses templates would be a good baseline to showcase the superiority of the proposed pipeline for “mastering the intricate structure and esoteric strategies.”
3.	Dense retrievers have been around for years and have shown to be superior to BM25 in general. Yet a simple BM25 is used for retrieval.
4.	The clarity can be improved. More details about American competitive policy debate should be provided for the general audience. It follows a rigid structure, yet the structure is not explained. For instance, the components and structure of an affirmative case can be presented. Also, the figures can be better prepared. Fig 3 is a very important figure, but it was not designed for easy perusal. Fig 1 takes up a lot of space without providing much information. Lastly, the paper can be better formatted. For instance, the unnecessarily large gaps in pg 2 can be removed and the appendix can be formatted so that readers know what to look for and where.
5. The whole body of work on argument generation, which is much more than just IBM project debator, are closely related to this work and should be surveyed.

**Questions:**

-	Spreading seems to be a relatively easy technique for AI systems to master. If it is considered a legal technique, what prevents AI from abusing it?
-	This task seems a lot more rigid in structure than the setup for IBM Project Debator. I see that the need to fluently generate all the components can be more challenging, but doesn’t the rigid structure also make this task easier?

---

### Official Review · Reviewer_knmN · 2025-10-31

**Soundness:** 1
**Presentation:** 2
**Contribution:** 1
**Rating:** 0
**Confidence:** 4

**Summary:**

The authors tackle the problem of American-style policy debating, a challenging and structured competitive debate format that emphasizes detailed high-quality evidence and structure constraints. They propose a multi-agent pipeline that constructs the various components of the policy debates, relying on sparse retrieval over a curated dataset of debate evidence. The paper presents results showing that this agentic pipeline outperforms human-authored debates.

**Strengths:**

The work deals with a timely topic, illustrating the potential (and danger) of producing persuasive content with modern LLMs.

**Weaknesses:**

1. In my view, the paper does not offer any technical novelty - the gist of it is that the authors tailored a multi-agent pipeline to their problem and dataset and used gpt-4.1 mini along with agent frameworks and retrieval with BM25. While a limited technical novelty can be OK for a paper in the applications track, it does certainly increase the burden to deliver strong contributions in other aspects (e.g., comprehensive analyses, useful code, surprising results, valuable resources etc.), which unfortunately I personally thought were lacking as well. Moreover, when claiming that "our core contribution is a novel multi-agent architecture" I would have expected something more than a keyword search with a pipeline of LLM-based nodes that look quite standard (generate -> search -> review quality, as described in Figure 2).
2. There is no analysis, and really very little in terms of results - the entire empirical section consists of human evaluation of 3 system outputs and the evaluation of 20 simulated debates using an LLM judge (with no additional validation). So this raises some doubts about the robustness and significance of the work, but just as importantly in my opinion does not provide much insights to the reader, whether in terms of ML, LLM reasoning, implementation challenges or even debating-specific insights. As I see it, the paper content can be accurately summed up as "we tailored a multi-agent pipeline with retrieval and according to an LLM judge it beats humans". In my view, this is simply not enough as a contribution for a conference paper.
3. The description of the pipeline is composed mainly of a large amount of debating jargon, making it largely incomprehensible to the naive reader.
4. I felt that the paper somewhat misrepresents prior works. Notably, the authors state as an advantage of the present work that it relies on a human curated evidence dataset, whereas prior works synthesized arguments from broad corpora. But importantly, argument mining and synthesis from a large-scale general-purpose corpus is precisely the technical challenge that many of the prior works aimed to tackle; and relying on a debate-specific corpus, where relevant debate arguments and debate evidence are readily accessible, mainly means that the present work tackles a very different, and in certain ways easier, technical problem. Also regarding specifics, stating for example that Slonim et al. "produces a single, monolithic speech with very limited reference to evidence" is incorrect (they produced more than one speech per motion and did employ evidence as a central component).

**Questions:**

* Why does the abstract talk about a "continuously running live spectacle debate performance"? I am not sure how this connects to the rest of the paper content.

Typos:
- l. 108 Our contribution -> Our contributions
- l. 174 growing body literature -> growing body of literature
- l. 178 "hyperpersuasion -> "hyperpersuasion"

---

### Official Review · Reviewer_kAyi · 2025-10-31

**Soundness:** 2
**Presentation:** 1
**Contribution:** 2
**Rating:** 2
**Confidence:** 4

**Summary:**

The paper describes an end-to-end fully autonomic debating system for American-style policy debates which is a significantly more complex competitive debate format than the ones that have been previously considered in this space. The system contains multi-agent workflows for strategic planning and argument generation where multiple agents collaborate and critique each other's outcomes to eventually come up with the best response.

**Strengths:**

Successfully participating in comptetive policy debates requires a very high level of planning, strategizing and reasoning capabilities. Most humans will struggle if asked to participate in such a debate. Developing autonomous systems that can achieve this is extremely ambitious and the authors should be commended for pursuing this goal. The techniques suggested, even though they lack details, seem reasonable as they take inspiration from the cognitive processes that humans go through when participating in a debate.

**Weaknesses:**

The paper lacks details. While there is a full system diagram in the appendix, it is very complex and difficult to understand. The authors write that each component is a multi-agent system where agents collaborate and critique each other, but hardly any details are provided on how this is done. Figure 2 is too high-level and the description in Section 4.3 presents the stages which expert human debaters go through during a competitive debate. It involves a lot of debating terminology but no description of the actual implementation of the multi-agent workflows.

Very few experiments were conducted with no ablations whatsoever. It is possible that the authors developed a very impressive system, but the way the paper is currently written, it is not ready for publication.

**Questions:**

Please address the following comments regarding the comparison to Project Debater:

- From the description in Section 3, one can infer that IBM's Project Debater generates a one-off speech. While it is true that the debate format addressed in Project Debater is much simpler than policy debates, it does include opening, rebuttal, and summary speeches.

- The system described in the paper uses a database of high-quality arguments (‘cards’) created specifically for policy style debates. This bypasses the difficult task of extracting claims and evidence from unstructured corpus, and putting together, which was the main challenge in previous autonomous debating systems such as Project Debater.

---

### Author Response · Authors · 2025-11-12
**We thank the reviewers for the work they've done in reviewing this paper.**

We appreciate the tireless work that the reviewers have put into this paper. While the scores are low, we, the authors, believe that the critique is correct and that we intend to take the criticism to heart and resolve all of these issues.

We will need more time than is offered during this rebuttal period to rework the paper. We also note that our work has been accepted to a workshop at AAAI around creative AI presentations. And we will use the comments from the review phase here, as well as the comments and in-person demonstration feedback to improve this paper. We intend to post a version to arxiv soon, which will be iterated upon as we flesh this out.

---

### Note · Authors · 2025-11-12

I have read and agree with the venue's withdrawal policy on behalf of myself and my co-authors.